# Globally Convergent Newton Methods for Ill-conditioned Generalized Self-concordant Losses

**Ulysse Marteau-Ferey**
INRIA - École Normale Supérieure
PSL Reasearch University
ulysse.marteau-ferey@inria.fr

**Francis Bach**
INRIA - École Normale Supérieure
PSL Reasearch University
francis.bach@inria.fr

**Alessandro Rudi**
INRIA - École Normale Supérieure
PSL Reasearch University
alessandro.rudi@inria.fr

## Abstract

In this paper, we study large-scale convex optimization algorithms based on the Newton method applied to regularized generalized self-concordant losses, which include logistic regression and softmax regression. We first prove that our new simple scheme based on a sequence of problems with decreasing regularization parameters is provably globally convergent, that this convergence is linear with a constant factor which scales only logarithmically with the condition number. In the parametric setting, we obtain an algorithm with the same scaling than regular first-order methods but with an improved behavior, in particular in ill-conditioned problems. Second, in the non-parametric machine learning setting, we provide an explicit algorithm combining the previous scheme with Nyström projection techniques, and prove that it achieves optimal generalization bounds with a time complexity of order $O(n\mathsf{df}_\lambda)$, a memory complexity of order $O(\mathsf{df}_\lambda^2)$ and *no dependence on the condition number*, generalizing the results known for least-squares regression. Here $n$ is the number of observations and $\mathsf{df}_\lambda$ is the associated degrees of freedom. In particular, this is the first large-scale algorithm to solve logistic and softmax regressions in the non-parametric setting with large condition numbers and theoretical guarantees.

## 1 Introduction

Minimization algorithms constitute a crucial algorithmic part of many machine learning methods, with algorithms available for a variety of situations [10]. In this paper, we focus on *finite sum* problems of the form

$$\min_{x \in \mathcal{H}} f_\lambda(x) = f(x) + \frac{\lambda}{2}\|x\|^2, \text{ with } f(x) = \frac{1}{n}\sum_{i=1}^{n} f_i(x),$$

where $\mathcal{H}$ is a Euclidean or a Hilbert space, and each function is convex and smooth. The running-time of minimization algorithms classically depends on the number of functions $n$, the explicit (for Euclidean spaces) or implicit (for Hilbert spaces) dimension $d$ of the search space, and the condition number of the problem, which is upper bounded by $\kappa = L/\lambda$, where $L$ characterizes the smoothness of the functions $f_i$, and $\lambda$ the regularization parameter.

In the last few years, there has been a strong focus on problems with large $n$ and $d$, leading to *first-order* (i.e., gradient-based) stochastic algorithms, culminating in a sequence of linearly convergent

algorithms whose running time is favorable in $n$ and $d$, but scale at best in $\sqrt{\kappa}$ [15, 22, 14, 4]. However, modern problems lead to objective functions with very large condition numbers, i.e., in many learning problems, the regularization parameter that is optimal for test predictive performance may be so small that the scaling above in $\sqrt{\kappa}$ is not practical anymore (see examples in Sect. 5).

These ill-conditioned problems are good candidates for *second-order methods* (i.e., that use the Hessians of the objective functions) such as Newton method. These methods are traditionally discarded within machine learning for several reasons: (1) they are usually adapted to high precision results which are not necessary for generalization to unseen data for machine learning problems [9], (2) computing the Newton step $\Delta_\lambda(x) = \nabla^2 f_\lambda(x)^{-1} \nabla f_\lambda(x)$ requires to form the Hessian and solve the associated linear system, leading to complexity which is at least quadratic in $d$, and thus prohibitive for large $d$, and (3) the global convergence properties are not applicable, unless the function is very special, i.e., self-concordant [24] (which includes only few classical learning problems), so they often are only shown to converge in a small area around the optimal $x$.

In this paper, we argue that the three reasons above for not using Newton method can be circumvented to obtain competitive algorithms: (1) high absolute precisions are indeed not needed for machine learning, but faced with strongly ill-conditioned problems, even a low-precision solution requires second-order schemes; (2) many approximate Newton steps have been designed for approximating the solution of the associated large linear system [1, 27, 25, 8]; (3) we propose a novel second-order method which is globally convergent and which is based on performing approximate Newton methods for a certain class of so-called *generalized self-concordant functions* which includes logistic regression [6]. For these functions, the conditioning of the problem is also characterized by a more *local* quantity: $\kappa_\ell = R^2/\lambda$, where $R$ characterizes the local evolution of Hessians. This leads to second-order algorithms which are competitive with first-order algorithms for well-conditioned problems, while being superior for ill-conditioned problems which are common in practice.

**Contributions.** We make the following contributions:

($a$) We build a global second-order method for the minimization of $f_\lambda$, which relies only on computing approximate Newton steps of the functions $f_\mu, \mu \geq \lambda$. The number of such steps will be of order $O(c \log \kappa_\ell + \log \frac{1}{\epsilon})$ where $\epsilon$ is the desired precision, and $c$ is an explicit constant. In the parametric setting ($\mathcal{H} = \mathbb{R}^d$), $c$ can be as bad as $\sqrt{\kappa_\ell}$ in the worst-case but much smaller in theory and practice. Moreover in the non-parametric/kernel machine learning setting ($\mathcal{H}$ infinite dimensional), $c$ does not depend on the local condition number $\kappa_\ell$.

($b$) Together with the appropriate quadratic solver to compute approximate Newton steps, we obtain an algorithm with the same scaling as regular first-order methods but with an improved behavior, in particular in ill-conditioned problems. Indeed, this algorithm matches the performance of the best quadratic solvers but covers any generalized self-concordant function, up to logarithmic terms.

($c$) In the non-parametric/kernel machine learning setting we provide an explicit algorithm combining the previous scheme with Nyström projections techniques. We prove that it achieves optimal generalization bounds with $O(n\mathsf{df}_\lambda)$ in time and $O(\mathsf{df}_\lambda^2)$ in memory, where $n$ is the number of observations and $\mathsf{df}_\lambda$ is the associated degrees of freedom. In particular, this is the first large-scale algorithm to solve logistic and softmax regression in the non-parametric setting with large condition numbers and theoretical guarantees.

## 1.1 Comparison to related work

We consider two cases for $\mathcal{H}$ and the functions $f_i$ that are common in machine learning: $\mathcal{H} = \mathbb{R}^d$ with linear (in the parameter) models with explicit feature maps, and $\mathcal{H}$ infinite-dimensional, corresponding in machine learning to learning with kernels [32]. Moreover in this section we first consider the quadratic case, for example the squared loss in machine learning (i.e., $f_i(x) = \frac{1}{2}(x^\top z_i - y_i)^2$ for some $z_i \in \mathcal{H}, y_i \in \mathbb{R}$). We first need to introduce the Hessian of the problem, for any $\lambda > 0$, define

$$\mathbf{H}(x) := \nabla^2 f(x), \qquad \mathbf{H}_\lambda(x) := \nabla^2 f_\lambda(x) = \mathbf{H}(x) + \lambda \mathbf{I},$$

in particular we denote by $\mathbf{H}$ (and analogously $\mathbf{H}_\lambda$) the Hessian at optimum (which in case of squared loss corresponds to the covariance matrix of the inputs).

**Quadratic problems and $\mathcal{H} = \mathbb{R}^d$ (ridge regression).** The problem then consists in solving a (ill-conditioned) positive semi-definite symmetric linear system of dimension $d \times d$. Methods based on *randomized linear algebra*, *sketching* and suitable *subsampling* [17, 18, 11] are able to find the solution with precision $\epsilon$ in time that is $O((nd + \min(n,d)^3) \log(L/\lambda\epsilon))$, so essentially independently of the condition number, because of the logarithmic complexity in $\lambda$.

**Quadratic problems and $\mathcal{H}$ infinite-dimensional (kernel ridge regression).** Here the problem corresponds to solving a (ill-conditioned) infinite-dimensional linear system in a reproducing kernel Hilbert space [32]. Since however the sum defining $f$ is finite, the problem can be projected on a subspace of dimension at most $n$ [5], leading to a linear system of dimension $n \times n$. Solving it with the techniques above would lead to a complexity of the order $O(n^2)$, which is not feasible on massive learning problems (e.g., $n \approx 10^7$). Interestingly these problems are usually approximately low-rank, with the rank represented by the so called *effective-dimension* $\mathsf{df}_\lambda$ [13], counting essentially the eigenvalues of the problem larger than $\lambda$,

$$\mathsf{df}_\lambda = \mathrm{Tr}(\mathbf{HH}_\lambda^{-1}). \tag{1}$$

Note that $\mathsf{df}_\lambda$ is bounded by $\min\{n, L/\lambda\}$ and in many cases $\mathsf{df}_\lambda \ll \min(n, L/\lambda)$. Using suitable projection techniques, like *Nyström* [34] or *random features* [26] it is possible to further reduce the problem to dimension $\mathsf{df}_\lambda$, for a total cost to find the solution of $O(n\mathsf{df}_\lambda^2)$. Finally recent methods [29], combining suitable projection methods with refined preconditioning techniques, are able to find the solution with precision compatible with the optimal statistical learning error [13] in time that is $O(n\mathsf{df}_\lambda \log(L/\lambda))$, so being essentially independent of the condition number of the problem.

**Convex problems and explicit features (logistic regression).** When the loss function is *self-concordant* it is possible to leverage the fast techniques for linear systems in approximate Newton algorithms [25] (see more in Sec. 2), to achieve the solution in essentially $O(nd + \min(n,d)^3)$ time, modulo logarithmic terms. However only few loss functions of interest are self-concordant, in particular the widely used logistic and soft-max losses are not self-concordant, but *generalized-self-concordant* [6]. In such cases we need to use (accelerated/stochastic) first order optimization methods to enter in the quadratic convergence region of Newton methods [2], which leads to a solution in $O(dn + d\sqrt{nL/\lambda} + \min(n,d)^3)$ time, which does not present any improvement on a simple accelerated first-order method. Globally convergent second-order methods have also been proposed to solve such problems [21], but the number of Newton steps needed being bounded only by $L/\lambda$, they lead to a solution in $O(L/\lambda \, (nd + \min(n,d)^3))$. With $\lambda$ that could be as small as $10^{-12}$ in modern machine learning problems, this makes both these kind of approaches expensive from a computational viewpoint for ill-conditioned problems. For such problems, with our new global second-order scheme, the algorithm we propose achieves instead a complexity of essentially $O((nd + \min(n,d)^3) \log(R^2/\lambda\epsilon))$ (see Thm. 1).

**Convex problems and $\mathcal{H}$ infinite-dimensional (kernel logistic regression).** Analogously to the case above, it is not possible to use Newton methods profitably as global optimizers on losses that are not self-concordant as we see in Sec. 3. In such cases by combining projecting techniques developped in Sec. 4 and accelerated first-order optimization methods, it is possible to find a solution in $O(n\mathsf{df}_\lambda + \mathsf{df}_\lambda \sqrt{nL/\lambda})$ time. This can still be prohibitive in the very small regularization scenario, since it strongly depends on the condition number $L/\lambda$. In Sec. 4 we suitably combine our optimization algorithm with projection techniques achieving optimal statistical learning error [23] in essentially $O(n\mathsf{df}_\lambda \log(R^2/\lambda))$.

**First-order algorithms for finite sums.** In dimension $d$, accelerated algorithms for strongly-convex smooth (not necessarily self-concordant) finite sums, such as K-SVRG [4], have a running time proportional $O((n + \sqrt{nL/\lambda})d)$. This can be improved with preconditioning to $O((n + \sqrt{dL/\lambda})d)$ for large $n$ [2]. Quasi-Newton methods can also be used [20], but typically without the guarantees that we provide in this paper (which are logarithmic in the condition number in natural scenarios).

## 2   Background: Newton methods and generalized self concordance

In this section we start by recalling the definition of generalized self concordant functions and motivate it with examples. We then recall basic facts about Newton and approximate Newton methods, and

present existing techniques to efficiently compute approximate Newton steps. We start by introducing the definition of generalized self-concordance, that here is an extension of the one in [6].

**Definition 1** (generalized self-concordant (GSC) function). *Let $\mathcal{H}$ be a Hilbert space. We say that $f$ is a generalized self-concordant function on $\mathcal{G} \subset \mathcal{H}$, when $\mathcal{G}$ is a bounded subset of $\mathcal{H}$ and $f$ is a convex and three times differentiable mapping on $\mathcal{H}$ such that*

$$\forall x \in \mathcal{H}, \ \forall h, k \in \mathcal{H}, \ \nabla^{(3)} f(x)[h, k, k] \leq \sup_{g \in \mathcal{G}} |g \cdot h| \ \nabla^2 f(x)[k, k].$$

We will usually denote by $R$ the quantity $\sup_{g \in \mathcal{G}} \|g\| < \infty$ and often omit $\mathcal{G}$ when it is clear from the context (for simplicity think of $\mathcal{G}$ as the ball in $\mathcal{H}$ centered in zero and with radius $R > 0$, then $\sup_{g \in \mathcal{G}} |g \cdot h| = R\|h\|$). The globally convergent second-order scheme we present in Sec. 3 is specific to losses which satisfy this generalized self-concordance property. The following loss functions, which are widely used in machine learning, are generalized-self-concordant, and motivate this work.

**Example 1** (Application to finite-sum minimization). *The following loss functions are generalized self-concordant functions, but not self-concordant:*
*(a) Logistic regression: $f_i(x) = \log(1 + \exp(-y_i w_i^\top x))$, where $x, w_i \in \mathbb{R}^d$ and $y_i \in \{-1, 1\}$.*
*(b) Softmax regression: $f_i(x) = \log \left( \sum_{j=1}^k \exp(x_j^\top w_i) \right) - x_{y_i}^\top w_i$, where now $x \in \mathbb{R}^{d \times k}$ and $y_i \in \{1, \dots, k\}$ and $x_j$ denotes the $j$-th column of $x$.*
*(c) Generalized linear models with bounded features (see details in [7, Sec. 2.1]), which include conditional random fields [33].*
*(d) Robust regression: $f_i(x) = \varphi(y_i - w_i^\top x)$ with $\varphi(u) = \log(e^u + e^{-u})$.*

Note that these losses are not *self-concordant* in the sense of [25]. Moreover, even if the losses $f_i$ are self-concordant, the objective function $f$ is not necessarily self-concordant, making any attempt to prove the self-concordance of the objective function $f$ almost impossible.

**Newton method (NM).** Given $x_0 \in \mathcal{H}$, the Newton method consists in doing the following update:

$$x_{t+1} = x_t - \Delta_\lambda(x_t), \qquad \Delta_\lambda(x_t) := \mathbf{H}_\lambda^{-1}(x_t) \nabla f_\lambda(x_t). \tag{2}$$

The quantity $\Delta_\lambda(x) := \mathbf{H}_\lambda^{-1}(x) \nabla f_\lambda(x)$ is called the Newton step at point $x$, and $x - \Delta_\lambda(x)$ is the minimizer of the second order approximation of $f_\lambda$ around $x$. Newton methods enjoy the following key property: if $x_0$ is close enough to the optimum, the convergence to the optimum is quadratic and the number of iterations required to a given precision is independent of the condition number of the problem [12].

However Newton methods have two main limitations: (a) the region of quadratic convergence can be quite small and reaching the region can be computationally expensive, since it is usually done via first order methods [2] that converge linearly depending on the condition number of the problem, (b) the cost of computing the Hessian can be really expensive when $n, d$ are large, and also (c) the cost of computing $\Delta_\lambda(x_t)$ can be really prohibitive. In the rest of the section we recall some ways to deal with (b) and (c). Our main result of Sec. 3 is to provide globalization scheme for the Newton method to tackle problem (a), which is easily integrable with approximate techniques to deal with (b) ans (c), to make second-order technique competitive.

**Approximate Newton methods (ANM) and approximate solutions to linear systems.** Computing exactly the Newton increment $\Delta_\lambda(x_t)$, which corresponds essentially to the solution of a linear system, can be too expensive when $n, d$ are large. A natural idea is to approximate the Newton iteration, leading to *approximate Newton methods*,

$$x_{t+1} = x_t - \widetilde{\Delta}_\lambda(x_t), \qquad \widetilde{\Delta}_\lambda \approx \Delta_\lambda(x_t). \tag{3}$$

In this paper, more generally we consider any technique to compute $\widetilde{\Delta}_\lambda(x_t)$ that provides a *relative approximation* [16] of $\Delta_\lambda(x_t)$ defined as follows.

**Definition 2** (relative approximation). *Let $\rho < 1$, let $\mathbf{A}$ be an invertible positive definite Hermitian operator on $\mathcal{H}$ and $b$ in $\mathcal{H}$. We denote by $\mathrm{LinApprox}(\mathbf{A}, b, \rho)$ the set of all $\rho$-relative approximations of $z^* = \mathbf{A}^{-1} b$, i.e., $\mathrm{LinApprox}(\mathbf{A}, b, \rho) = \{z \in \mathcal{H} \mid \|z - z^*\|_\mathbf{A} \leq \rho\|z^*\|_\mathbf{A}\}$.*

**Sketching and subsampling for approximate Newton methods.** Many techniques for approximating linear systems have been used to compute $\widetilde{\Delta}_\lambda$, in particular *sketching* of the Hessian matrix via fast transforms and *subsampling* (see [25, 8, 2] and references therein). Assuming for simplicity that $f_i = \ell_i(w_i^\top x)$, with $\ell_i : \mathbb{R} \to \mathbb{R}$ and $w_i \in \mathcal{H}$, it holds:

$$\mathbf{H}(x) = \frac{1}{n} \sum_{i=1}^{n} \ell_i^{(2)}(w_i^\top x) w_i w_i^\top = V_x^\top V_x, \tag{4}$$

with $V_x \in \mathbb{R}^{n \times d} = D_x W$, where $D_x \in \mathbb{R}^{n \times n}$ is a diagonal matrix defined as $(D_x)_{ii} = (\ell_i^{(2)}(w_i^\top x))^{1/2}$ and $W \in \mathbb{R}^{n \times d}$ defined as $W = (w_1, \ldots, w_n)^\top$.

Both sketching and subsampling methods approximate $z^* = \mathbf{H}_\lambda(x)^{-1} \nabla f_\lambda(x)$ with $\tilde{z} = \widetilde{\mathbf{H}}_\lambda(x)^{-1} \nabla f_\lambda(x)$, in particular, in the case of subsampling $\widetilde{\mathbf{H}}(x) = \sum_{j=1}^{Q} p_j w_{i_j} w_{i_j}^\top$ where $Q \ll \min(n, d)$, $(p_j)_{j=1}^n$ are suitable weights and $(i_j)_{j=1}^Q$ are indices selected at random from $\{1, \ldots, n\}$ with suitable probabilities. Sketching methods instead use $\widetilde{\mathbf{H}}(x) = \widetilde{V}_x^\top \widetilde{V}_x$, with $\widetilde{V}_x = \Omega V_x$ with $\Omega \in \mathbb{R}^{Q \times n}$ a structured matrix such that computing $\widetilde{V}_x$ has a cost in the order of $O(nd \log n)$; to this end usually $\Omega$ is based on fast Fourier or Hadamard transforms [25]. Note that essentially all the techniques used in approximate Newton methods guarantee relative approximation. In particular the following results can be found in the literature (see Lemmas 28 and 29 in Appendix I and [25], Lemma 2 for more details).

**Lemma 1.** *Let $x, b \in \mathcal{H}$ and assume that $\ell_i^{(2)} \leq a$ for $a > 0$. With probability $1 - \delta$ the following methods output an element in $\mathrm{LinApprox}(\mathbf{H}_\lambda(x), b, \rho)$, in $O(Q^2 d + Q^3 + c)$ time, $O(Q^2 + d)$ space:*
*(a) Subsampling with uniform sampling (see [27, 28]), where $Q = O(\rho^{-2} a / \lambda \log \frac{1}{\lambda \delta})$ and $c = O(1)$.*
*(b) Subsampling with approximate leverage scores [27, 3, 28]), where $Q = O(\rho^{-2} \bar{\mathsf{df}}_\lambda \log 1/\lambda \delta)$, $c = O(\min(n, a/\lambda) \bar{\mathsf{df}}_\lambda^2)$ and $\bar{\mathsf{df}}_\lambda = \mathrm{Tr}(W^\top W (W^\top W + \lambda/a I)^{-1})$ [30]. Note that $\bar{\mathsf{df}}_\lambda \leq \min(n, d)$.*
*(c) Sketching with fast Hadamard transform [25], where $Q = O(\rho^{-2} \bar{\mathsf{df}}_\lambda \log a/\lambda \delta)$, $c = O(nd \log n)$.*

## 3 Globally convergent scheme for ANM algorithms on GSC functions

The algorithm is based on the observation that when $f_\lambda$ is generalized self concordant, there exists a region where $t$ steps of ANM converge as fast as $2^{-t}$. Our idea is to start from a very large regularization parameter $\lambda_0$, such that we are sure that $x_0$ is in the convergence region and perform some steps of ANM such that the solution enters in the convergence region of $f_{\lambda_1}$, with $\lambda_1 = q\lambda_0$ with $q < 1$, and to iterate this procedure until we enter the convergence region of $f_\lambda$. First we define the region of interest and characterize the behavior of NM and ANM in the region, then we analyze the globalization scheme.

**Preliminary results: the Dikin ellipsoid.** We consider the following region that we prove to be contained in the region of quadratic convergence for the Newton method and that will be useful to build the globalization scheme. Let $c, R > 0$ and $f_\lambda$ be generalized self-concordant with coefficient $R$, we call *Dikin ellipsoid* and denote by $\mathsf{D}_\lambda(\mathsf{c})$ the region

$$\mathsf{D}_\lambda(\mathsf{c}) := \left\{ x \mid \nu_\lambda(x) \leq \mathsf{c}\sqrt{\lambda}/R \right\}, \quad \text{with} \quad \nu_\lambda(x) := \|\nabla f_\lambda(x)\|_{\mathbf{H}_\lambda^{-1}(x)},$$

where $\nu_\lambda(x)$ is usually called the *Newton decrement* and $\|x\|_\mathbf{A}$ stands for $\|\mathbf{A}^{1/2} x\|$.

**Lemma 2.** *Let $\lambda > 0, \mathsf{c} \leq 1/7$, let $f_\lambda$ be generalized self-concordant and $x \in \mathsf{D}_\lambda(\mathsf{c})$. Then it holds: $\frac{1}{4} \nu_\lambda(x)^2 \leq f_\lambda(x) - f_\lambda(x^\star) \leq \nu_\lambda(x)^2$. Moreover Newton method starting from $x_0$ has quadratic convergence, i.e., let $x_t$ be obtained via $t \in \mathbb{N}$ steps of Newton method in Eq. (2), then $\nu_\lambda(x_t) \leq 2^{-(2^t-1)} \nu_\lambda(x_0)$. Finally, approximate Newton methods starting from $x_0$ have a linear convergence rate, i.e., let $x_t$ given by Eq. (3), with $\widetilde{\Delta}_t \in \mathrm{LinApprox}(\mathbf{H}_\lambda(x_t), \nabla f_\lambda(x_t), \rho)$ and $\rho \leq 1/7$, then $\nu_\lambda(x_t) \leq 2^{-t} \nu_\lambda(x_0)$.*

This result is proved in Lemma 11 in Appendix B.3. The crucial aspect of the result above is that when $x_0 \in \mathsf{D}_\lambda(\mathsf{c})$, the convergence of the approximate Newton method is linear and does not depend on the condition number of the problem. However $\mathsf{D}_\lambda(\mathsf{c})$ itself can be very small depending on $\sqrt{\lambda}/R$. In the next subsection we see how to enter in $\mathsf{D}_\lambda(\mathsf{c})$ in an efficient way.

**Entering the Dikin ellipsoid using a second-order scheme.** The lemma above shows that $\mathsf{D}_\lambda(\mathsf{c})$ is a good region where to use the approximate Newton algorithm on GSC functions. However the region itself is quite small, since it depends on $\sqrt{\lambda}/R$. Some other globalization schemes arrive to regions of interest by first-order methods or back-tracking schemes [2, 1]. However such approaches require a number of steps that is usually proportional to $\sqrt{L/\lambda}$ making them non-beneficial in machine learning contexts. Here instead we consider the following simple scheme where $\mathtt{ANM}_\rho(f_\lambda, x, t)$ is the result of a $\rho$-relative approximate Newton method performing $t$ steps of optimization starting from $x$.

The main ingredient to guarantee the scheme to work is the following lemma (see Lemma 13 in Appendix C.1 for a proof).

**Lemma 3.** *Let $\mu > 0$, $\mathsf{c} < 1$ and $x \in \mathcal{H}$. Let $s = 1 + R\|x\|/\mathsf{c}$, then for $q \in [1 - 2/(3s), 1)$*
$$\mathsf{D}_\mu(\mathsf{c}/3) \subseteq \mathsf{D}_{q\mu}(\mathsf{c}).$$

Now we are ready to show that we can guarantee the loop invariant $x_k \in \mathsf{D}_{\mu_k}(\mathsf{c})$. Indeed assume that $x_{k-1} \in \mathsf{D}_{\mu_{k-1}}(\mathsf{c})$. Then $\nu_{\mu_{k-1}}(x_{k-1}) \leq \mathsf{c}\sqrt{\mu_{k-1}}/R$. By taking $t = 2, \rho = 1/7$, and performing $x_k = \mathtt{ANM}_\rho(f_{\mu_{k-1}}, x_{k-1}, t)$, by Lemma 2, $\nu_{\mu_{k-1}}(x_k) \leq 1/4\nu_{\mu_{k-1}}(x_{k-1}) \leq \mathsf{c}/4\ \sqrt{\mu_{k-1}}/R$, i.e., $x_k \in \mathsf{D}_{\mu_{k-1}}(\mathsf{c}/4)$. If $q_k$ is large enough, this implies that $x_k \in \mathsf{D}_{q_k\mu_{k-1}}(\mathsf{c}) = \mathsf{D}_{\mu_k}(\mathsf{c})$, by Lemma 3. Now we are ready to state our main theorem of this section.

---

**Proposed Globalization Scheme**

*Phase I: Getting in the Dikin ellispoid of $f_\lambda$*

Start with $x_0 \in \mathcal{H}, \mu_0 > 0, t, T \in \mathbb{N}$ and $(q_k)_{k \in \mathbb{N}} \in (0, 1]$.
For $k \in \mathbb{N}$
$\quad x_{k+1} \leftarrow \mathtt{ANM}_\rho(f_{\mu_k}, x_k, t)$
$\quad \mu_{k+1} \leftarrow q_{k+1}\mu_k$
Stop when $\mu_{k+1} < \lambda$ and set $x_{last} \leftarrow x_k$.

*Phase II: reach a certain precision starting from inside the Dikin ellipsoid*

Return $\widehat{x} \leftarrow \mathtt{ANM}_\rho(f_\lambda, x_{last}, T)$

---

**Fully adaptive method.** The scheme presented above converges with the following parameters.

**Theorem 1.** *Let $\epsilon > 0$. Set $\mu_0 = 7R\|\nabla f(0)\|$, $x_0 = 0$, and perform the globalization scheme above for $\rho \leq 1/7, t = 2$, and $q_k = \frac{1/3 + 7R\|x_k\|}{1 + 7R\|x_k\|}$, $T = \lceil \log_2 \sqrt{1 \vee (\lambda\epsilon^{-1}/R^2)} \rceil$. Then denoting by $K$ the number of steps performed in the Phase I, it holds:*
$$f_\lambda(\widehat{x}) - f_\lambda(x_\lambda^\star) \leq \epsilon, \qquad K \leq \lfloor (3 + 11R\|x_\lambda^\star\|) \log(7R\|\nabla f(0)\|/\lambda) \rfloor.$$

Note that the theorem above (proven in Appendix C.3) guarantees a solution with error $\epsilon$ with $K$ steps of ANM each performing 2 iterations of approximate linear system solving, plus a final step of ANM which performs $T$ iterations of approximate linear system solving. In case of $f_i(x) = \ell_i(w_i^\top x)$, with $\ell_i : \mathbb{R} \to \mathbb{R}, w_i \in \mathcal{H}$ with $\ell_i^{(2)} \leq a$, for $a > 0$, the final runtime cost of the proposed scheme to achieve precision $\epsilon$, when combined with of the methods for approximate linear system solving from Lemma 1 (i.e. sketching), is $O(Q^2 + d)$ in memory and

$$O\left((nd\log n + dQ^2 + Q^3)\left(R\|x_\lambda^\star\|\log\frac{R}{\lambda} + \log\frac{\lambda}{R\epsilon}\right)\right) \text{ in time}, \quad Q = O\left(\bar{\mathsf{df}}_\lambda \log\frac{1}{\lambda\delta}\right),$$

where $\bar{\mathsf{df}}_\lambda$, defined in Lemma 1, measures the *effective dimension* of the correlation matrix $W^\top W$ with $W = (w_1, \ldots, w_n)^\top \in \mathbb{R}^{n \times d}$, corresponding essentially to the number of eigenvalues of $W^\top W$ larger than $\lambda/a$. In particular note that $\bar{\mathsf{df}}_\lambda \leq \min(n, d, \text{rank}(W), ab^2/\lambda)$, with $b := \max_i \|w_i\|$, and usually way smaller than such quantities.

**Remark 1.** *The proposed method does not depend on the condition number of the problem $L/\lambda$, but on the term $R\|x_\lambda^\star\|$ which can be in the order of $R/\sqrt{\lambda}$ in the worst case, but usually way smaller. For example, it is possible to prove that this term is bounded by an absolute constant not depending on $\lambda$, if at least one minimum for $f$ exists. In the appendix (see Proposition 7), we show a variant of this adaptive method which can leverage the regularity of the solution with respect to the Hessian, i.e., depending on the smaller quantity $R\sqrt{\lambda}\|x_\lambda^\star\|_{\mathbf{H}_\lambda^{-1}(x_\lambda^\star)}$ instead of $R\|x_\lambda^\star\|$.*

Finally note that it is possible to use $q_k = q$ fixed for all the iterations and way smaller than the one in Thm. 1, depending on some regularity properties of $\mathbf{H}$ (see Proposition 8 in Appendix C.2).

# 4 Application to the non-parametric setting: Kernel methods

In supervised learning the goal is to predict well on future data, given the observed training dataset. Let $\mathcal{X}$ be the input space and $\mathcal{Y} \subseteq \mathbb{R}^p$ be the output space. We consider a probability distribution $P$ over $\mathcal{X} \times \mathcal{Y}$ generating the data and the goal is to estimate $g^* : \mathcal{X} \to \mathcal{Y}$ solving the problem

$$g^* = \arg\min_{g:\mathcal{X}\to\mathcal{Y}} \mathcal{L}(g), \quad \mathcal{L}(g) = \mathbb{E}[\ell(g(x), y)], \tag{5}$$

for a given loss function $\ell : \mathcal{Y} \times \mathcal{Y} \to \mathbb{R}$. Note that $P$ is not known, and accessible only via the dataset $(x_i, y_i)_{i=1}^n$, with $n \in \mathbb{N}$, independently sampled from $P$. A prototypical estimator for $g^*$ is the regularized minimizer of the empirical risk $\widehat{\mathcal{L}}(g) = \frac{1}{n}\sum_{i=1}^n \ell(g(x_i), y_i)$ over a suitable space of functions $\mathcal{G}$. Given $\phi : \mathcal{X} \to \mathcal{H}$ a common choice is to select $\mathcal{G}$ as the set of linear functions of $\phi(x)$, that is, $\mathcal{G} = \{w^\top \phi(\cdot) \mid w \in \mathcal{H}\}$. Then the regularized minimizer of $\widehat{\mathcal{L}}$, denoted by $\widehat{g}_\lambda$, corresponds to

$$\widehat{g}_\lambda(x) = \widehat{w}_\lambda^\top \phi(x), \quad \widehat{w}_\lambda = \arg\min_{w\in\mathcal{H}} \frac{1}{n}\sum_{i=1}^n f_i(w) + \lambda\|w\|^2, \quad f_i(w) = \ell(w^\top\phi(x_i), y_i). \tag{6}$$

Learning theory guarantees how fast $\widehat{g}_\lambda$ converges to the best possible estimator $g^*$ with respect to the number of observed examples, in terms of the so called *excess risk* $\mathcal{L}(\widehat{g}_\lambda) - \mathcal{L}(g^*)$. The following theorem recovers the minimax optimal learning rates for squared loss and extend them to any generalized self-concordant loss function.

*Note on* $\mathsf{df}_\lambda$. In this section, we always denote with $\mathsf{df}_\lambda$ the effective dimension of the problem in Eq. (5). When the loss belongs to the family of generalized linear models (see Example 1) and if the model is well-specified, then $\mathsf{df}_\lambda$ is defined exactly as in Eq. (1) otherwise we need a more refined definition (see [23] or Eq. (30) in Appendix D).

**Theorem 2** (from [23], Thm. 4). *Let $\lambda > 0, \delta \in (0, 1]$. Let $\ell$ be generalized self-concordant with parameter $R > 0$ and $\sup_{x\in X}\|\phi(x)\| \le C < \infty$. Assume that there exists $g^*$ minimizing $\mathcal{L}$. Then there exists $c_0$ not depending on $n, \lambda, \delta, \mathsf{df}_\lambda, C, g^*$, such that if $\sqrt{\mathsf{df}_\lambda/n}, \mathsf{b}_\lambda \le \lambda^{1/2}/R$, and $n \ge C/\lambda\log(\delta^{-1}C/\lambda)$ the following holds with probability $1 - \delta$:*

$$\mathcal{L}(\widehat{g}_\lambda) - \mathcal{L}(g^*) \le c_0\Big(\frac{\mathsf{df}_\lambda}{n} + \mathsf{b}_\lambda^2\Big)\log(1/\delta), \qquad \mathsf{b}_\lambda := \lambda\|g^*\|_{\mathbf{H}_\lambda^{-1}(g^*)}. \tag{7}$$

Under standard regularity assumptions of the learning problems [23], i.e., (a) the *capacity condition* $\sigma_j(\mathbf{H}(g^*)) \le Cj^{-\alpha}$, for $\alpha \ge 1, C > 0$ (i.e., a decay of eigenvalues $\sigma_j(\mathbf{H}(g^*))$ of the Hessian at the optimum), and (b) the *source condition* $g^* = \mathbf{H}(g^*)^r v$, with $v \in \mathcal{H}$ and $r > 0$ (i.e., the control of the optimal $g^*$ for a specific Hessian-dependent norm), $\mathsf{df}_\lambda \le C'\lambda^{-1/\alpha}$ and $\mathsf{b}_\lambda^2 \le C''\lambda^{1+2r}$, leading to the following optimal learning rate,

$$\mathcal{L}(\widehat{g}_\lambda) - \mathcal{L}(g^*) \le c_1 n^{-\frac{1+2r\alpha}{1+\alpha+2r\alpha}}\log(1/\delta), \quad \text{when} \quad \lambda = n^{-\frac{\alpha}{1+\alpha+2r\alpha}}. \tag{8}$$

Now we propose an algorithmic scheme to compute efficiently an approximation of $\widehat{g}_\lambda$ that achieves the same optimal learning rates. First we need to introduce the technique we are going to use.

**Nyström projection.** It consists in suitably selecting $\{\bar{x}_1, \dots, \bar{x}_M\} \subset \{x_1, \dots, x_n\}$, with $M \ll n$ and computing $\bar{g}_{M,\lambda}$, i.e., the solution of Eq. (6) over $\mathcal{H}_M = \mathrm{span}\{\phi(\bar{x}_1), \dots, \phi(\bar{x}_M)\}$ instead of $\mathcal{H}$. In this case the problem can be reformulated as a problem in $\mathbb{R}^M$ as

$$\bar{g}_{M,\lambda} = \bar{\alpha}_{M,\lambda}^\top \mathbf{T}^{-1}v(x), \qquad \bar{\alpha}_{M,\lambda} = \arg\min_{\alpha\in\mathbb{R}^M} \bar{f}_\lambda(\alpha), \qquad \bar{f}(\alpha) = \frac{1}{n}\sum_{i=1}^n \bar{f}_i(\alpha) + \lambda\|\alpha\|^2, \tag{9}$$

where $\bar{f}_i(\alpha) = \ell(v(x_i)^\top\mathbf{T}^{-1}\alpha, y_i)$ and $v(x) \in \mathbb{R}^M$, $v(x) = (k(x, \bar{x}_1), \dots, k(x, \bar{x}_M))$ with $k(x, x') = \phi(x)^\top\phi(x')$ the associated positive-definite kernel [32], while $\mathbf{T}$ is the upper triangular matrix such that $\mathbf{K} = \mathbf{T}^\top\mathbf{T}$, with $\mathbf{K} \in \mathbb{R}^{M\times M}$ with $\mathbf{K}_{ij} = k(\bar{x}_i, \bar{x}_j)$. In the next theorem we characterize the sufficient $M$ to achieve minimax optimal rates, for two standard techniques of choosing the Nyström points $\{\bar{x}_1, \dots, \bar{x}_M\}$.

**Theorem 3** (Optimal rates for learning with Nyström). *Let $\lambda > 0, \delta \in (0, 1]$. Assume the conditions of Thm. 2. Then the excess risk of $\bar{g}_{M,\lambda}$ is bounded with prob. $1 - 2\delta$ as in Eq. (7) (with $c_1' \propto c_1$), when*

(1) *Uniform Nyström method [28, 29] is used and $M \ge C_1/\lambda \log(C_2/\lambda\delta)$.*
(2) *Approximate leverage score method [3, 28, 29] is used and $M \ge C_3 \mathsf{df}_\lambda \log(C_4/\lambda\delta)$.*
*Here $C, C_1, C_2, C_4$ do not depend on $\lambda, n, M, \mathsf{df}_\lambda, \delta$.*

Thm. 3 generalizes results for learning with Nyström and squared loss [28], to GSC losses. It is proved in Thm. 6, in Appendix D.4. As in [28], Thm. 3 shows that Nyström is a valid technique for dimensionality reduction. Indeed it is essentially possible to project the learning problem on a subspace $\mathcal{H}_M$ of dimension $M = O(c/\lambda)$ or even as small as $M = O(\mathsf{df}_\lambda)$ and still achieve the optimal rates of Thm. 2. Now we are ready to introduce our algorithm.

**Proposed algorithm.** The algorithm conceptually consists in (a) performing a projection step with Nyström, and (b) solving the resulting optimization problem with the globalization scheme proposed in Sec. 3 based on ANM in Eq. (3). In particular, we want to avoid to apply explicitly $\mathbf{T}^{-1}$ to each $v(x_i)$ in Eq. (9), which would require $O(nM^2)$ time. Then we will use the following approximation technique based only on matrix vector products, so we can just apply $\mathbf{T}^{-1}$ to $\alpha$ at each iteration, with a total cost proportional only to $O(nM + M^2)$ per iteration. Given $\alpha, \nabla \bar{f}_\lambda(\alpha)$, we approximate $z^* = \bar{\mathbf{H}}_\lambda(\alpha)^{-1} \nabla \bar{f}_\lambda(\alpha)$, where $\bar{\mathbf{H}}_\lambda$ is the Hessian of $\bar{f}_\lambda(\alpha)$, with $\tilde{z}$ defined as

$$\tilde{z} = \texttt{prec-conj-grad}_t(\bar{\mathbf{H}}_\lambda(\alpha), \nabla \bar{f}_\lambda(\alpha)),$$

where $\texttt{prec-conj-grad}_t$ corresponds to performing $t$ steps of preconditioned conjugate gradient [19] with preconditioner computed using a subsampling approach for the Hessian among the ones presented in Sec. 2, in the paragraph starting with Eq. (4). The pseudocode for the whole procedure is presented in Alg. 1, Appendix E. This technique of approximate linear system solving has been studied in [29] in the context of empirical risk minimization for squared loss.

**Lemma 4** ([29]). *Let $\lambda > 0, \alpha, b \in \mathbb{R}^M$. The previous method, applied with $t = O(\log 1/\rho)$, outputs an element of $\mathrm{LinApprox}(\bar{\mathbf{H}}_\lambda(\alpha), b, \rho)$, with probability $1 - \delta$ with complexity $O((nM + M^2 Q + M^3 + c)t)$ in time and $O(M^2 + n)$ in space, with $Q = O(C_1/\lambda \log(C_1/\lambda\delta)), c = O(1)$ if uniform sub-sampling is used or $Q = O(C_2 \mathsf{df}_\lambda \log(C_1/\lambda\delta)), c = O(\mathsf{df}_\lambda^2 \min(n, \frac{1}{\lambda}))$ if sub-sampling with leverage scores is used [30].*

A more complete version of this lemma is shown in Proposition 12 in Appendix D.5.1. We conclude this section with a result proving the learning properties of the proposed algorithm.

**Theorem 4** (Optimal rates for the proposed algorithms). *Let $\lambda > 0$ and $\epsilon < \lambda/R^2$. Under the hypotheses of Thm. 3, if we set $M$ as in Thm. 3, $Q$ as in Lemma 4 and setting the globalization scheme as in Thm. 1, then the proposed algorithm (Alg. 1, Appendix E) finishes in a finite number of newton steps $N_{ns} = O(R\|g^*\| \log(C/\lambda) + \log(C/\epsilon))$ and returns a predictor $g_{Q,M,\lambda}$ of the form $g_{Q,M,\lambda} = \alpha^\top \mathbf{T}^{-1} v(x)$. With probability at least $1 - \delta$, this predictor satisfies:*

$$\mathcal{L}(g_{Q,M,\lambda}) - \mathcal{L}(g^*) \le c_0 \Big( \frac{\mathsf{df}_\lambda}{n} + \mathsf{b}_\lambda^2 + \epsilon \Big) \log(1/\delta), \qquad \mathsf{b}_\lambda := \lambda \|g^*\|_{\mathbf{H}_\lambda^{-1}(g^*)}. \tag{10}$$

The theorem above (see Proposition 14, Appendix D.6 for exacts quantifications) shows that the proposed algorithm is able to achieve the same learning rates of plain empirical risk minimization as in Thm. 2. The total complexity of the procedure, including the cost of computing the preconditioner, the selection of the Nyström points via approximate leverage scores and also the computation of the leverage scores [30] is then

$$O\left(R\|g^*\| \log(R^2/\lambda) \left( n\, \mathsf{df}_\lambda \log(C\lambda^{-1}\delta^{-1})\, c_X + + \mathsf{df}_\lambda^3 \log^3(C\lambda^{-1}\delta^{-1}) + \min(n, C/\lambda)\, \mathsf{df}_\lambda^2 \right) \right)$$

in time and $O(\mathsf{df}_\lambda^2 \log^2(C\lambda^{-1}\delta^{-1}))$ in space, where $c_X$ is the cost of computing the inner product $k(x, x')$ (in the kernel setting assumed when the input space $X$ is $X = \mathbb{R}^p$ it is $c = O(p)$). As noted in [30], under the standard regularity assumptions on the learning problem seen above, $\mathsf{df}_\lambda^2 \le \mathsf{df}_\lambda/\lambda \le n$ when the optimal $\lambda$ is chosen. So the total computational complexity is

$$O\left(R \log(R^2/\lambda)\, \log^3(C\lambda^{-1}\delta^{-1})\, \|g^*\| \cdot n \cdot \mathsf{df}_\lambda \cdot c_X\right) \text{ in time,} \quad O(\mathsf{df}_\lambda^2 \cdot \log^2(C\lambda^{-1}\delta^{-1})) \text{ in space.}$$

First note, the fact that due to the statistical properties of the problem the complexity does not depend even implicitly on $\sqrt{C/\lambda}$, but only on $\log(C/\lambda)$, so the algorithm runs in essentially $O(n\mathsf{df}_\lambda)$, compared to $O(\mathsf{df}_\lambda \sqrt{nC/\lambda})$ of the accelerated first-order methods we develop in Appendix F and the $O(n\mathsf{df}_\lambda \sqrt{C/\lambda})$ of other Newton schemes (see Sec. 1.1). To our knowledge, this is the first algorithm to achieve optimal statistical learning rates for generalized self-concordant losses and with complexity only $\widetilde{O}(n\mathsf{df}_\lambda)$. This generalizes similar results for squared loss [29, 30].

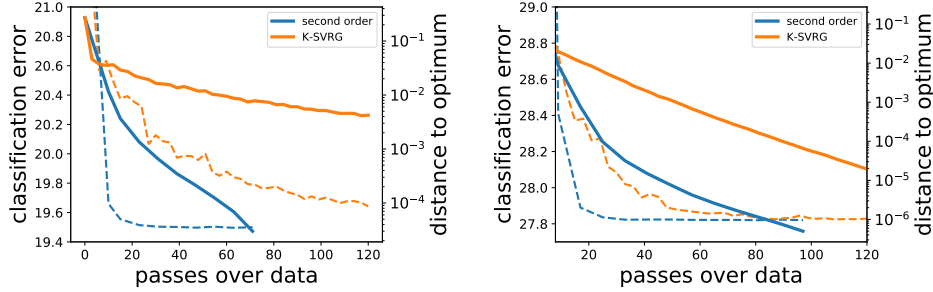

Figure 1: Training loss and test error as as function of the number of passes on the data for our algorithm vs. K-SVRG. on the **(left)** Susy and **(right)** Higgs data sets.

# 5 Experiments

The code necessary to reproduce the following experiments is available on GitHub at `https://github.com/umarteau/Newton-Method-for-GSC-losses-`.

We compared the performances of our algorithm for kernel logistic regression on two large scale classification datasets ($n \approx 10^7$), Higgs and Susy, pre-processed as in [29]. We implemented the algorithm in pytorch and performed the computations on 1 Tesla P100-PCIE-16GB GPU. For Susy ($n = 5 \times 10^6, p = 18$): we used Gaussian kernel with $k(x, x') = e^{-\|x-x'\|^2/(2\sigma^2)}$, with $\sigma = 5$, which we obtained through a grid search (in [29], $\sigma = 4$ is taken for the ridge regression); $M = 10^4$ Nyström centers and a subsampling $Q = M$ for the preconditioner, both obtained with uniform sampling. Analogously for Higgs ($n = 1.1 \times 10^7, p = 28$): , we used a Gaussian kernel with $\sigma = 5$ and $M = 2.5 \times 10^4$ and $Q = M$, using again uniform sampling. To find reasonable $\lambda$ for supervised learning applications, we cross-validated $\lambda$ finding the minimum test error at $\lambda = 10^{-10}$ for Susy and $\lambda = 10^{-9}$ for Higgs (see Figs. 2 and 3 in Appendix F) for such values our algorithm and the competitor achieve an error of 19.5% on the test set for Susy, comparable to the state of the art (19.6% [29]) and analogously for Higgs (see Appendix F). We then used such $\lambda$'s as regularization parameters and compared our algorithm with a well known accelerated stochastic gradient technique *Katyusha SVRG* (K-SVRG) [4], tailored to our problem using mini batches. In Fig. 1 we show the convergence of the training loss and classification error with respect to the number of passes on the data, of our algorithm compared to K-SVRG. It is possible to note our algorithm is order of magnitude faster in achieving convergence, validating empirically the fact that the proposed algorithm scales as $O(n\mathsf{df}_\lambda)$ in learning settings, while accelerated first order methods go as $O((n + \sqrt{nL/\lambda})\mathsf{df}_\lambda)$. Moreover, as mentioned in the introduction, this highlights the fact that precise optimization is necessary to achieve a good performance in terms of test error. Finally, note that since a pass on the data is much more expensive for K-SVRG than for our second order method (see Appendix F for details), the difference in computing time between the second order scheme and K-SVRG is even more in favour of our second order method (see Figs. 4 and 5 in Appendix F).

### Acknowledgments

We acknowledge support from the European Research Council (grant SEQUOIA 724063).

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
