[Supplementary Material]

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

# Organization of the Appendix

## A. Main results on generalized self-concordant functions

Notations, definitions and basic results concerning generalized self-concordant functions.

## B. Results on approximate Newton methods

In this section, the interaction between the notion of Dikin ellipsoid, approximate Newton methods and generalized self-concordant functions is studied. The results needed in the main paper are all concentrated in Appendix B.3. In particular the results in Lemma 2 are proven in a more general form in Lemma 11.

## C. Proof of bounds for the globalization scheme

In this section, we leverage the results of the previous two sections to analyze the globalization scheme.

### C.1. Main technical lemmas
We start by proving the result on the inclusion of Dikin ellipsoids (Lemma 3).

### C.2. Proof of main theorems
In particular, a general version of Thm. 1 is proven. Moreover Remark 1 is proven in Proposition 7, while the fixed scheme to choose $(q_k)_{k \in \mathbb{N}}$ is proven in Proposition 8.

### C.3. Proof of Thm. 1
Finally, we prove the properties of the globalization schemes presented in Thm. 1.

## D. Non-parametric learning with generalized self-concordant functions

In this section, some basic results about non-parametric learning with generalized self-concordant functions are recalled and the main results of Sec. 4 are proven.

### D.1. General setting and assumptions, statistical result for regularized ERM.
More details about the generalization properties of empirical risk minimization as well as the optimal rates in Thm. 2 are recalled.

### D.2. Reducing the dimension: projecting on a subspace using Nyström sub-sampling.

### D.3. Sub-sampling techniques.
The basics of uniform sub-sampling and sub-sampling with approximate leverage scores are recalled.

### D.4. Selecting the $M$ Nyström points
Thm. 3 is proven in a more general version in Thm. 6.

### D.5 Performing the globalization scheme to approximate $\beta_{M,\lambda}$
A general scheme is proposed to solve the projected problem approximately using the globalization scheme.

#### D.5.1. Performing approximate Newton steps
We start by describing the way of computing approximate Newton steps. A generalized version of Lemma 4 is proven in Proposition 12.

#### D.5.2. Applying the globalization scheme to control $\widehat{\nu}_{M,\lambda}(\beta)$
We then completely analyse the approximating of $\beta_{M,\lambda}$ from an optimization point of view (see Proposition 13).

### D.6. Final algorithm and results
Finally, the proof of Thm. 4 is provided, using the results of the previous subsections.

## E. Algorithm

In this section, the pseudocode for the algorithm presented in Sec. 4 and analyzed in Thm. 7 is provided.

## F. Experiments

In this section, more details about the experiments are provided.

## G. Solving a projected problem to reduce dimension

In this section, more details about the problem of randomized projections are provided.

## A   Main results on generalized self-concordant functions

In this section, we start by introducing a few notations. We define the key notion of generalized self-concordance in Appendix A.1, and present the main results concerning generalized self-concordant functions. In Appendix A.2, we describe how generalized self-concordance behaves with respect to an expectation or to certain relaxations.

**Notations**   Let $\lambda \geq 0$ and $\mathbf{A}$ be a bounded positive semidefinite Hermitian operator on $\mathcal{H}$. We denote with $\mathbf{I}$ the identity operator, and

$$\|x\|_{\mathbf{A}} := \|\mathbf{A}^{1/2}x\|, \tag{11}$$
$$\mathbf{A}_{\lambda} := \mathbf{A} + \lambda\mathbf{I}. \tag{12}$$

Let $f$ be a twice differentiable convex function on a Hilbert space $\mathcal{H}$. We adopt the following notation for the Hessian of $f$:

$$\forall x \in \mathcal{H}, \ \mathbf{H}_f(x) := \nabla^2 f(x) \in \mathcal{L}(\mathcal{H}).$$

For any $\lambda > 0$, we define the $\lambda$-regularization of $f$:

$$f_{\lambda} := f + \frac{\lambda}{2}\|\cdot\|^2.$$

$f_{\lambda}$ is $\lambda$-strongly convex and has a unique minimizer which we denote with $x_{\star}^{f,\lambda}$. Moreover, define

$$\forall x \in \mathcal{H}, \ \mathbf{H}_{f,\lambda}(x) := \nabla^2 f_{\lambda}(x) = \mathbf{H}_f(x) + \lambda\mathbf{I}, \qquad \nu_{f,\lambda}(x) := \|\nabla f_{\lambda}(x)\|_{\mathbf{H}_{f,\lambda}^{-1}(x)}.$$

The quantity $\nu_{f,\lambda}(x)$ is called the **Newton decrement** at point $x$ and will play a significant role.

When the function $f$ is clear from the context, we will omit the subscripts with $f$ and use $\mathbf{H}, \mathbf{H}_{\lambda}, \nu_{\lambda}....$

### A.1   Definitions and results on generalized self-concordant functions

In this section, we introduce the main definitions and results for self-concordant functions. These results are mainly the same as in appendix B of [23].

**Definition 3** (generalized self-concordant function)**.** *Let $\mathcal{H}$ be a Hilbert space. Formally, a generalized self-concordant function on $\mathcal{H}$ is a couple $(f, \mathcal{G})$ where:*

*i  $\mathcal{G}$ is a bounded subset of $\mathcal{H}$; we will usually denote $\|\mathcal{G}\|$ or $R$ the quantity $\sup_{g \in \mathcal{G}} \|g\| < \infty$;*

*ii  $f$ is a convex and three times differentiable mapping on $\mathcal{H}$ such that*

$$\forall x \in \mathcal{H}, \ \forall h, k \in \mathcal{H}, \ \nabla^{(3)} f(x)[h, k, k] \leq \sup_{g \in \mathcal{G}} |g \cdot h| \, \nabla^2 f(x)[k, k].$$

To make notations lighter, we will often omit $\mathcal{G}$ from the notations and simply say that $f$ stands both for the mapping and the couple $(f, \mathcal{G})$.

**Definition 4** (Definitions). *Let $f$ be a generalized self-concordant function. We define the following quantities.*

- *$\forall h \in \mathcal{H},\ \mathsf{t}_f(h) := \sup_{g \in \mathcal{G}} |h \cdot g|$;*

- *$\forall x \in \mathcal{H},\ \forall \lambda > 0,\ \mathsf{r}_{f,\lambda}(x) := \frac{1}{\sup_{g \in \mathcal{G}} \|g\|_{\mathbf{H}_{f,\lambda}^{-1}(x)}}$;*

- *$\forall \mathsf{c} \geq 0,\ \forall \lambda > 0,\ \mathsf{D}_{f,\lambda}(\mathsf{c}) := \{x\ :\ \nu_{f,\lambda}(x) \leq \mathsf{cr}_{f,\lambda}(x)\}$.*

*We also define the following functions:*

$$\psi(t) = \frac{e^t - t - 1}{t^2}, \quad \underline{\phi}(t) = \frac{1 - e^{-t}}{t}, \quad \overline{\phi}(t) = \frac{e^t - 1}{t}. \tag{13}$$

Note that $\psi, \overline{\phi}$ are increasing functions and that $\underline{\phi}$ is a decreasing function. Moreover, $\frac{\overline{\phi}(t)}{\underline{\phi}(t)} = e^t$. Once again, if $f$ is clear, we will often omit the reference to $f$ in the quantities above, keeping only $\mathsf{t}, \mathsf{r}_\lambda, \mathsf{D}_\lambda$...

We condense results obtained in [23] under a slightly different form. The proofs, however, are exactly the same.

While in [23], only the regularized case is dealt with, the proof techniques are exactly the same to obtain Proposition 1. Proposition 2 is proved explicitly in Proposition 4 of [23] and Lemma 5 is proved in Proposition 5.
Omitting the subscript $f$, we get the following results.

**Proposition 1** (Bounds for the non-regularized function $f$). *Let $f$ be a generalized self-concordant function. Then the following bounds hold (we omit $f$ in the subscripts):*

$$\forall x \in \mathcal{H},\ \forall h \in \mathcal{H},\ e^{-\mathsf{t}(h)} \mathbf{H}(x) \preceq \mathbf{H}(x + h) \preceq e^{\mathsf{t}(h)} \mathbf{H}(x), \tag{14}$$

$$\forall x, h \in \mathcal{H},\ \forall \lambda > 0,\ \|\nabla f(x + h) - \nabla f(x)\|_{\mathbf{H}_\lambda^{-1}(x)} \leq \overline{\phi}(\mathsf{t}(h))\|h\|_{\mathbf{H}_\lambda(x)}, \tag{15}$$

$$\forall x, h \in \mathcal{H},\ \psi(-\mathsf{t}(h))\|h\|_{\mathbf{H}(x)}^2 \leq f(x + h) - f(x) - \nabla f(x).h \leq \psi(\mathsf{t}(h))\|h\|_{\mathbf{H}(x)}^2. \tag{16}$$

We get the analoguous bounds in the regularized case.

**Proposition 2** (Bounds for the regularized function $f_\lambda$). *Let $f$ be a generalized self-concordant function and $\lambda > 0$ be a regularizer. Then the following bounds hold:*

$$\forall x, h \in \mathcal{H},\ e^{-\mathsf{t}(h)} \mathbf{H}_\lambda(x) \preceq \mathbf{H}_\lambda(x + h) \preceq e^{\mathsf{t}(h)} \mathbf{H}_\lambda(x), \tag{17}$$

$$\forall x, h \in \mathcal{H},\ \underline{\phi}(\mathsf{t}(h))\|h\|_{\mathbf{H}_\lambda(x)} \leq \|\nabla f_\lambda(x + h) - \nabla f_\lambda(x)\|_{\mathbf{H}_\lambda^{-1}(x)} \leq \overline{\phi}(\mathsf{t}(h))\|h\|_{\mathbf{H}_\lambda(x)}, \tag{18}$$

$$\forall x, h \in \mathcal{H},\ \psi(-\mathsf{t}(h))\|h\|_{\mathbf{H}_\lambda(x)}^2 \leq f_\lambda(x + h) - f_\lambda(x) - \nabla f_\lambda(x).h \leq \psi(\mathsf{t}(h))\|h\|_{\mathbf{H}_\lambda(x)}^2. \tag{19}$$

**Corollary 1.** *Let $f$ be a $\mathcal{G}$ generalized self-concordant function and $\lambda > 0$ be a regularizer, and $x_\lambda^\star$ the unique minimizer of $f_\lambda$. Then the following bounds hold for any $x \in \mathcal{H}$:*

$$\underline{\phi}(\mathsf{t}(x - x_\lambda^\star))\|x - x_\lambda^\star\|_{\mathbf{H}_\lambda(x)} \leq \underbrace{\|\nabla f_\lambda(x)\|_{\mathbf{H}_\lambda^{-1}(x)}}_{\nu_\lambda(x)} \leq \overline{\phi}(\mathsf{t}(x - x_\lambda^\star))\|x - x_\lambda^\star\|_{\mathbf{H}_\lambda(x)}, \tag{20}$$

$$\psi(-\mathsf{t}(x - x_\lambda^\star))\|x - x_\lambda^\star\|_{\mathbf{H}_\lambda(x_\lambda^\star)}^2 \leq f_\lambda(x) - f_\lambda(x_\lambda^\star) \leq \psi(\mathsf{t}(x - x_\lambda^\star))\|x - x_\lambda^\star\|_{\mathbf{H}_\lambda(x_\lambda^\star)}^2. \tag{21}$$

Moreover, the following localization lemma holds.

**Lemma 5** (localization). *Let $\lambda > 0$ be fixed. If $\frac{\nu_\lambda(x)}{r_\lambda(x)} < 1$, then*

$$\mathsf{t}(x - x_\lambda^\star) \leq -\log\left(1 - \frac{\nu_\lambda(x)}{r_\lambda(x)}\right). \tag{22}$$

*In particular, this shows:*

$$\forall \mathsf{c} < 1, \ \forall \lambda > 0, \ x \in \mathsf{D}_\lambda(\mathsf{c}) \implies \mathsf{t}(x - x_\lambda^\star) \leq -\log(1 - \mathsf{c}).$$

We now state a Lemma which shows that the difference to the optimum in function values is equivalent to the squared newton decrement in a small Dikin ellipsoid. We will use this result in the main paper.

**Lemma 6** (Equivalence of norms). *Let $\lambda > 0$ and $x \in \mathsf{D}_\lambda(\frac{1}{7})$. Then the following holds:*

$$\frac{1}{4}\nu_\lambda(x)^2 \leq f_\lambda(x) - f_\lambda(x_\lambda^\star) \leq \nu_\lambda(x)^2.$$

*Proof.* Apply Lemma 5 knowing $x \in \mathsf{D}_\lambda(\frac{1}{7})$ to get $\mathsf{t}(x - x_\lambda^\star) \leq \log(7/6)$. Then apply Eq. (19) and Eq. (18) to get:

$$
\begin{aligned}
f_\lambda(x) - f_\lambda(x_\lambda^\star) &\leq \psi(\mathsf{t}(x - x_\lambda^\star))\|x - x_\lambda^\star\|_{\mathbf{H}_\lambda(x_\lambda^\star)}^2 \\
&\leq e^{\mathsf{t}(x - x_\lambda^\star)}\psi(\mathsf{t}(x - x_\lambda^\star))\|x - x_\lambda^\star\|_{\mathbf{H}_\lambda(x)}^2 \\
&\leq \frac{e^{\mathsf{t}(x - x_\lambda^\star)}\psi(\mathsf{t}(x - x_\lambda^\star))}{\phi(\mathsf{t}(x - x_\lambda^\star))^2}\nu_\lambda(x)^2.
\end{aligned}
$$

Replacing with the bound above, we get

$$\forall \lambda > 0, \ \forall x \in \mathsf{D}_\lambda(\frac{1}{7}), \ f_\lambda(x) - f_\lambda(x_\lambda^\star) \leq \nu_\lambda(x)^2.$$

For the lower bound, proceed in exactly the same way. $\qquad\square$

## A.2 Comparison between generalized self-concordant functions

The following result is straightforward.

**Lemma 7** (Comparison between generalized self-concordant functions). *Let $\mathcal{G}_1 \subset \mathcal{G}_2 \subset \mathcal{H}$ be two bounded subsets. If $(f, \mathcal{G}_1)$ is generalized self-concordant, then $(f, \mathcal{G}_2)$ is also generalized self-concordant. Moreover,*

$$\forall x \in \mathcal{H}, \ \forall \lambda > 0, \ \mathsf{r}_{(f,\mathcal{G}_1),\lambda}(x) \geq \mathsf{r}_{(f,\mathcal{G}_2),\lambda}(x).$$

In particular, we will often use the following fact. If $(f, \mathcal{G})$ is generalized self-concordant, and $\mathcal{G}$ is bounded by $R$, then $(f, B_\mathcal{H}(R))$ is also generalized self-concordant. Moreover,

$$\mathsf{r}_{(f,B_\mathcal{H}(R)),\lambda}(x) = \frac{\sqrt{\lambda + \lambda_{\min}(\mathbf{H}_f(x))}}{R} \geq \frac{\sqrt{\lambda}}{R}.$$

We now state a result which shows that, given a family of generalized self-concordant functions, the expectancy of that family is also generalized self-concordant. This can be seen as a reformulation of Proposition 2 of [23].

**Proposition 3** (Expectation). *Let $\mathcal{Z}$ be a polish space equipped with its Borel sigma-algebra, and $\mathcal{H}$ be a Hilbert space. Let $((f_z, \mathcal{G}_z))_{z \in \mathcal{Z}}$ be a family of generalized self-concordant functions such that the mapping $(z, x) \mapsto f_z(x)$ is measurable.*

*Assume we are given a random variable $Z$ on $\mathcal{Z}$, whose support we denote with $\mathrm{supp}(Z)$, such that*

- *the random variables $\|f_Z(0)\|, \|\nabla f_Z(0)\|, \mathrm{Tr}(\nabla^2 f_Z(0))$ are are bounded;*

- *$\mathcal{G} := \bigcup_{z \in \mathrm{supp}(Z)} \mathcal{G}_z$ is a bounded subset of $\mathcal{H}$.*

*Then the mapping $f : x \in \mathcal{H} \mapsto \mathbb{E}\left[f_Z(x)\right]$ is well defined, $(f, \mathcal{G})$ is generalized self-concordant, and we can differentiate under the expectation.*

**Corollary 2.** *Let $n \in \mathbb{N}$ and $(f_i, \mathcal{G}_i)_{1 \leq i \leq n}$ be a family of generalized self-concordant functions. Define*

$$f(x) = \frac{1}{n} \sum_{i=1}^{n} f_i(x), \ \mathcal{G} = \bigcup_{i=1}^{n} \mathcal{G}_i.$$

*Then $(f, \mathcal{G})$ is generalized self-concordant.*

# B  Results on approximate Newton methods

In this section, we assume we are given a generalized self-concordant function $f$ in the sense of Appendix A. As $f$ will be fixed throughout this part, we will omit it from the notations. Recall the definitions from Definition 4:

$$\nu_\lambda(x) := \|\nabla f_\lambda(x)\|_{\mathbf{H}_\lambda^{-1}(x)}, \quad \frac{1}{\mathsf{r}_\lambda(x)} := \sup_{g \in \mathcal{G}} \|g\|_{\mathbf{H}_\lambda^{-1}(x)}, \quad \mathsf{D}_\lambda(\mathsf{c}) := \left\{ x \ : \ \frac{\nu_\lambda(x)}{\mathsf{r}_\lambda(x)} \leq \mathsf{c} \right\}.$$

Define the following quantities:

- the true Newton step at point $x$ for the $\lambda$-regularized problem:

$$\Delta_\lambda(x) := \mathbf{H}_\lambda^{-1}(x) \nabla f_\lambda(x).$$

- the renormalized Newton decrement $\widetilde{\nu}_\lambda(x)$:

$$\widetilde{\nu}_\lambda(x) := \frac{\nu_\lambda(x)}{\mathsf{r}_\lambda(x)}.$$

Moreover, note that a direct application of Eq. (17) yields the following equation which relates the radii at different points:

$$\forall \lambda > 0, \ \forall x \in \mathcal{H}, \ \forall h \in \mathcal{H}, \ e^{-\mathsf{t}(h)} \mathsf{r}_\lambda(x) \leq \mathsf{r}_\lambda(x+h) \leq e^{\mathsf{t}(h)} \mathsf{r}_\lambda(x). \tag{23}$$

In this appendix, we develop a complete analysis of so-called approximate Newton methods in the case of generalized self-concordant losses. By "approximate Newton method", we mean that instead of performing the classical update $x_{t+1} = x_t - \Delta_\lambda(x_t)$, we perform an update of the form $x_{t+1} = x_t - \widetilde{\Delta}_t$ where $\widetilde{\Delta}_t$ is an approximation of the real Newton step. We will characterize this approximation by measuring its distance to the real Newton step using two parameters $\rho$ and $\epsilon_0$:

$$\|\widetilde{\Delta}_t - \Delta_\lambda(x_t)\| \leq \rho \nu_\lambda(x_t) + \epsilon_0.$$

We start by presenting a few technical results in Appendix B.1. We continue by proving that an approximate Newton method has linear convergence guarantees in the right Dikin ellipsoid in Appendix B.2. In Appendix B.3, we adapt these results to a certain way of computing approximate Newton steps, which will be the one we use in the core of the paper. In Appendix B.4, we mention ways to reduce the computational burden of these methods by showing that since all Hessians are equivalent in Dikin ellipsoids, one can actually sketch the Hessian at one given point in that ellipsoid instead of re-sketching it at each Newton step. For the sake of simplicity, this is not mentioned in the core paper, but works very well in practice.

## B.1  Main technical results

We start with a technical decomposition of the Newton decrement at point $x - \widetilde{\Delta}$ for a given $\widetilde{\Delta} \in \mathcal{H}$.

**Lemma 8** (Technical decomposition). *Let $\lambda > 0$, $x \in \mathcal{H}$ be fixed. Assume we perform a step of the form $x - \widetilde{\Delta}$ for a certain $\widetilde{\Delta} \in \mathcal{H}$. Define*

$$\delta := \|\widetilde{\Delta} - \Delta_\lambda(x)\|_{\mathbf{H}_\lambda(x)}, \qquad \widetilde{\delta} := \frac{\delta}{\mathsf{r}_\lambda(x)}.$$

*The following holds:*

$$\widetilde{\nu}_\lambda(x - \widetilde{\Delta}) \leq e^{\widetilde{\nu}_\lambda(x) + \widetilde{\delta}} \left[ \psi(\widetilde{\nu}_\lambda(x) + \widetilde{\delta})(\widetilde{\nu}_\lambda(x) + \widetilde{\delta})^2 + \widetilde{\delta} \right]; \tag{24}$$

$$\nu_\lambda(x - \widetilde{\Delta}_\lambda(x)) \leq e^{\widetilde{\nu}_\lambda(x) + \widetilde{\delta}} \left[ \psi(\widetilde{\nu}_\lambda(x) + \widetilde{\delta})(\widetilde{\nu}_\lambda(x) + \widetilde{\delta})(\nu_\lambda(x) + \delta) + \delta \right]. \tag{25}$$

*Proof.* Note that by definition, $\nabla f_\lambda(x) = \mathbf{H}_\lambda(x) \Delta_\lambda(x)$. Hence

$$\|\nabla f^\lambda(x - \widetilde{\Delta})\|_{\mathbf{H}_\lambda^{-1}(x)} = \|\nabla f^\lambda(x - \widetilde{\Delta}) - \nabla f^\lambda(x) + \mathbf{H}_\lambda(x)\Delta_\lambda(x)\|_{\mathbf{H}_\lambda^{-1}(x)}$$

$$\leq \|\nabla f^\lambda(x - \widetilde{\Delta}) - \nabla f^\lambda(x) + \mathbf{H}_\lambda(x)\widetilde{\Delta}\|_{\mathbf{H}_\lambda^{-1}(x)}$$

$$+ \|\mathbf{H}_\lambda(x)(\Delta_\lambda(x) - \widetilde{\Delta})\|_{\mathbf{H}_\lambda^{-1}(x)}$$

$$= \|\int_0^1 [\mathbf{H}_\lambda(x - s\widetilde{\Delta}) - \mathbf{H}_\lambda(x)]\widetilde{\Delta}ds\|_{\mathbf{H}_\lambda^{-1}(x)} + \delta$$

$$\leq \int_0^1 \|\mathbf{H}_\lambda^{-1/2}(x)\mathbf{H}_\lambda(x - s\widetilde{\Delta})\mathbf{H}_\lambda^{-1/2}(x) - \mathbf{I}\|ds \, \|\widetilde{\Delta}\|_{\mathbf{H}_\lambda(x)} + \delta.$$

Now using Eq. (17), one has $\|\mathbf{H}_\lambda^{-1/2}(x)\mathbf{H}_\lambda(x - s\widetilde{\Delta})\mathbf{H}_\lambda^{-1/2}(x) - \mathbf{I}\| \leq e^{s\mathsf{t}(\widetilde{\Delta})} - 1$, whose integral on $s$ is $\psi(\mathsf{t}(\widetilde{\Delta}))\mathsf{t}(\widetilde{\Delta})$ where $\psi$ is defined in Definition 4. Morever, bounding

$$\|\widetilde{\Delta}\|_{\mathbf{H}_\lambda(x)} \leq \|\widetilde{\Delta} - \Delta_\lambda(x)\|_{\mathbf{H}_\lambda(x)} + \|\Delta_\lambda(x)\|_{\mathbf{H}_\lambda(x)} = \delta + \nu_\lambda(x),$$

it holds

$$\|\nabla f^\lambda(x - \widetilde{\Delta})\|_{\mathbf{H}_\lambda^{-1}(x)} \leq \psi(\mathsf{t}(\widetilde{\Delta}))\mathsf{t}(\widetilde{\Delta}) \, (\nu_\lambda(x) + \delta) + \delta.$$

**1.** Now note that using Eq. (17), it holds: $\nu_\lambda(x - \widetilde{\Delta}) \leq e^{\mathsf{t}(\widetilde{\Delta})/2}\|\nabla f^\lambda(x - \widetilde{\Delta})\|_{\mathbf{H}_\lambda^{-1}(x)}$ and hence:

$$\nu_\lambda(x - \widetilde{\Delta}) \leq e^{\mathsf{t}(\widetilde{\Delta})/2} \left( \psi(\mathsf{t}(\widetilde{\Delta}))\mathsf{t}(\widetilde{\Delta}) \, (\nu_\lambda(x) + \delta) + \delta \right). \tag{26}$$

**2.** Moreover, using Eq. (23),

$$\widetilde{\nu}_\lambda(x - \widetilde{\Delta}) \leq e^{\mathsf{t}(\widetilde{\Delta})} \left( \psi(\mathsf{t}(\widetilde{\Delta}))\mathsf{t}(\widetilde{\Delta}) \, (\widetilde{\nu}_\lambda(x) + \widetilde{\delta}) + \widetilde{\delta} \right). \tag{27}$$

Noting that

$$\mathsf{t}(\widetilde{\Delta}) \leq \frac{\|\widetilde{\Delta}\|_{\mathbf{H}_\lambda(x)}}{\mathsf{r}_\lambda(x)} \leq \widetilde{\nu}_\lambda(x) + \widetilde{\delta},$$

and bounding Eq. (26) simply by taking $e^{\mathsf{t}(\widetilde{\Delta})/2} \leq e^{\mathsf{t}(\widetilde{\Delta})}$, we get the two bounds in the lemma.

$\square$

We now place ourselves in the case where we are given an approximation of the Newton step of the following form. Assume $\lambda$ and $x$ are fixed, and that we approximate $\Delta_\lambda(x)$ with $\widetilde{\Delta}$ such that there exists $\rho \geq 0$ and $\epsilon_0 \geq 0$ such that it holds:

$$\|\widetilde{\Delta} - \Delta_\lambda(x)\|_{\mathbf{H}_\lambda(x)} \leq \rho\nu_\lambda(x) + \epsilon_0.$$

We define/prove the three different following regimes.

**Lemma 9** (3 regimes). *Let $x \in \mathsf{D}_\lambda\left(\frac{1}{7}\right)$ and $\lambda > 0$ be fixed. Let*

$$0 \leq \rho \leq \frac{1}{7}, \ \epsilon_0 \geq 0 \ s.t. \ \tilde{\varepsilon}_0 := \frac{\epsilon_0}{\mathsf{r}_\lambda(x)} \leq \frac{1}{21}.$$

*Let $\widetilde{\Delta}$ be an approximation of the Newton steps satisfying $\|\widetilde{\Delta} - \Delta_\lambda(x)\|_{\mathbf{H}_\lambda(x)} \leq \rho\nu_\lambda(x) + \epsilon_0$. The three following regimes appear.*

- *If $\widetilde{\nu}_\lambda(x) \geq \rho$ and $\widetilde{\nu}_\lambda(x)^2 \geq \tilde{\varepsilon}_0$, then we are in the **quadratic regime**, i.e.*

$$\frac{10\widetilde{\nu}_\lambda(x - \widetilde{\Delta}_\lambda(x))}{3} \leq \left( \frac{10\widetilde{\nu}_\lambda(x)}{3} \right)^2, \ \nu_\lambda(x - \widetilde{\Delta}_\lambda(x)) \leq \frac{10}{3}\widetilde{\nu}_\lambda(x)\nu_\lambda(x).$$

- *If $\rho \geq \widetilde{\nu}_\lambda(x)$ and $\rho\widetilde{\nu}_\lambda(x) \geq \widetilde{\epsilon}_0$, then we are in the **linear regime**, i.e.*

$$\frac{10}{3}\widetilde{\nu}_\lambda(x - \widetilde{\Delta}_\lambda(x)) \leq \left(\frac{10\rho}{3}\right)\left(\frac{10}{3}\widetilde{\nu}_\lambda(x)\right), \ \nu_\lambda(x - \widetilde{\Delta}_\lambda(x)) \leq \frac{10}{3}\widetilde{\nu}_\lambda(x)\nu_\lambda(x).$$

- *If $\widetilde{\epsilon}_0 \geq \widetilde{\nu}_\lambda(x)^2, \rho\,\widetilde{\nu}_\lambda(x)$, then the **maximal precision** of the approximation is reached, and it holds:*

$$\widetilde{\nu}_\lambda(x - \widetilde{\Delta}_\lambda(x)) \leq 3\widetilde{\epsilon}_0 \leq \frac{1}{7}, \ \nu_\lambda(x - \widetilde{\Delta}_\lambda(x)) \leq 3\epsilon_0.$$

*Proof.* Using the previous lemma,

$$\widetilde{\nu}_\lambda(x - \widetilde{\Delta}_\lambda(x)) \leq e^{(1+\rho)\widetilde{\nu}_\lambda(x)+\widetilde{\epsilon}_0}\left[\psi((1+\rho)\widetilde{\nu}_\lambda(x) + \widetilde{\epsilon}_0)((1+\rho)\widetilde{\nu}_\lambda(x) + \widetilde{\epsilon}_0)^2 + \rho\widetilde{\nu}_\lambda(x) + \widetilde{\epsilon}_0\right]$$
$$\leq \square_1(\widetilde{\nu}_\lambda(x), \rho, \widetilde{\epsilon}_0)\,\widetilde{\nu}_\lambda(x)^2 + \square_2(\widetilde{\nu}_\lambda(x), \rho, \widetilde{\epsilon}_0)\,\rho\widetilde{\nu}_\lambda(x) + \square_3(\widetilde{\nu}_\lambda(x), \rho, \widetilde{\epsilon}_0)\,\widetilde{\epsilon}_0,$$

and

$$\nu_\lambda(x - \widetilde{\Delta}_\lambda(x)) \leq \square_1(\widetilde{\nu}_\lambda(x), \rho, \widetilde{\epsilon}_0)\,\widetilde{\nu}_\lambda(x)\nu_\lambda(x) + \square_2(\widetilde{\nu}_\lambda(x), \rho, \widetilde{\epsilon}_0)\,\rho\nu_\lambda(x) + \square_3(\widetilde{\nu}_\lambda(x), \rho, \widetilde{\epsilon}_0)\,\epsilon_0,$$

where the following defintions are used:

$$\square_1(\widetilde{\nu}, \rho, \widetilde{\epsilon}_0) := e^{(1+\rho)\widetilde{\nu}+\widetilde{\epsilon}_0}\psi((1+\rho)\widetilde{\nu} + \widetilde{\epsilon}_0)(1+\rho)^2,$$
$$\square_2(\widetilde{\nu}, \rho, \widetilde{\epsilon}_0) := e^{(1+\rho)\widetilde{\nu}+\widetilde{\epsilon}_0},$$
$$\square_3(\widetilde{\nu}, \rho, \widetilde{\epsilon}_0) := e^{(1+\rho)\widetilde{\nu}+\widetilde{\epsilon}_0}\left[2\psi((1+\rho)\widetilde{\nu} + \widetilde{\epsilon}_0)(1+\rho)\widetilde{\nu} + 1\right].$$

Now assume $\widetilde{\epsilon}_0 \leq \frac{1}{21}, \widetilde{\nu}_\lambda(x), \rho \leq \frac{1}{7}$. Replacing these values in the functions above bounds $\square_1, \square_2$ and $\square_3$, and using the case distinction, we get the result. $\qquad\square$

## B.2 General analysis of an approximate Newton method

The following proposition describes the behavior of an approximate newton method where $\rho$ and $\epsilon_0$ are fixed a priori.

**Proposition 4** (General approximate Newton scheme results)**.** *Let $\mathsf{c} \leq \frac{1}{7}$ be fixed and $x_0 \in \mathsf{D}_\lambda(\mathsf{c})$ be a given starting point.*
*Let $\rho \leq \frac{1}{7}$ and $\epsilon_0$ such that $\epsilon_0 \leq \frac{\mathsf{c}}{4}\,\mathsf{r}_\lambda(x_0)$.*
*Define the following approximate Newton scheme:*

$$\forall t \geq 0, \ x_{t+1} = x_t - \widetilde{\Delta}_t, \qquad \|\widetilde{\Delta}_t - \Delta_\lambda(x_t)\|_{\mathbf{H}_\lambda(x_t)} \leq \rho\nu_\lambda(x_t) + \epsilon_0.$$

*The following guarantees hold.*

- $\forall t \geq 0, \ x_t \in \mathsf{D}_\lambda(\mathsf{c})$.

- *Let $t_c = \left\lfloor \log_2 \log_2 \frac{3}{10\rho} \right\rfloor + 1$.*

$$\forall t \leq t_c, \ \frac{10\widetilde{\nu}_\lambda(x_t)}{3} \leq \max\left(\frac{12\epsilon_0}{\mathsf{r}_\lambda(x_0)}, 2^{-2^t}\right),$$

$$\forall t \geq t_c, \ \frac{10\widetilde{\nu}_\lambda(x_t)}{3} \leq \max\left(\frac{12\epsilon_0}{\mathsf{r}_\lambda(x_0)}, \left(\frac{10\rho}{3}\right)^{t-t_c+1}\right).$$

- *We can bound the relative decrease for both the Newton decrement and the renormalized Newton decrement:*

$$\forall t \leq t_c, \qquad \nu_\lambda(x_t) \leq \max\left(3\epsilon_0, \left(\frac{1}{2}\right)^{2^t-1}\nu_\lambda(x_0)\right),$$

$$\widetilde{\nu}_\lambda(x_t) \leq \max\left(\frac{18\epsilon_0}{5 r_\lambda(x_0)}, \left(\frac{1}{2}\right)^{2^t-1}\widetilde{\nu}_\lambda(x_0)\right).$$

$$\forall t \geq t_c, \qquad \nu_\lambda(x_t) \leq \max\left(3\epsilon_0, \left(\frac{10\rho}{3}\right)^{t-t_c+1}\nu_\lambda(x_0)\right),$$

$$\widetilde{\nu}_\lambda(x_t) \leq \max\left(\frac{18\epsilon_0}{5 r_\lambda(x_0)}, \left(\frac{10\rho}{3}\right)^{t-t_c+1}\widetilde{\nu}_\lambda(x_0)\right).$$

*Proof.* Start by noting, using Eq. (23),

$$\forall x \in \mathsf{D}_\lambda\left(\frac{1}{7}\right), \quad \varepsilon \leq \frac{r_\lambda(x)}{21}, \quad \frac{6}{7}r_\lambda(x_0) \leq r_\lambda(x) \leq \frac{7}{6}r_\lambda(x_0). \tag{28}$$

In particular, this holds for any $x \in \mathsf{D}_\lambda(\mathsf{c})$, $\mathsf{c} \leq \frac{1}{7}$. Thus,

$$\forall \mathsf{c} \leq \frac{1}{7}, \ \forall x_0 \in \mathsf{D}_\lambda(\mathsf{c}), \quad \frac{\epsilon_0}{r_\lambda(x_0)} \leq \frac{\mathsf{c}}{4} \implies \forall x \in \mathsf{D}_\lambda(\mathsf{c}), \quad \frac{\epsilon_0}{r_\lambda(x)} \leq \frac{\mathsf{c}}{3}.$$

**1.** Proving the first point is simple by induction. Indeed, assume $\widetilde{\nu}_\lambda(x_t) \leq \mathsf{c}$. We can apply Lemma 9 since the conditions on $\varepsilon$ and $\rho$ guarantee that the conditions of this lemma are satisfied.

If we are in either the linear or quadratic regime, the fact that $\frac{10\rho}{3}, \frac{10\widetilde{\nu}_\lambda(x_t)}{3} \leq \frac{10}{21}$ show that $\widetilde{\nu}_\lambda(x_{t+1}) \leq \frac{10}{21}\widetilde{\nu}_\lambda(x_t) \leq \mathsf{c}$.

If we are in the last case, $\widetilde{\nu}_\lambda(x_{t+1}) \leq \frac{3\epsilon_0}{r_\lambda(x_t)} \leq \mathsf{c}$.

**2.** Let us prove the second bullet point by induction. Start by assuming the property holds at $t$. By the previous point, the hypothesis of Lemma 9 are satisfied at $x_t$ with $\rho$ and $\varepsilon$. Assume we are in the limiting case; we easily show that in this case,

$$\frac{10\widetilde{\nu}_\lambda(x_{t+1})}{3} \leq \frac{10}{3} 3\frac{\epsilon_0}{r_\lambda(x_t)} \leq \frac{35\epsilon_0}{3 r_\lambda(x_0)}.$$

Here, the last inequality comes from Eq. (28). If we are not in the limiting case, let us distinguish between the two following cases.

If $t \leq t_c - 1$,

$$\frac{10\widetilde{\nu}_\lambda(x_{t+1})}{3} \leq \frac{10\widetilde{\nu}_\lambda(x_t)}{3}\max\left(\frac{10\widetilde{\nu}_\lambda(x_t)}{3}, \frac{10\rho}{3}\right)$$

$$\leq \max\left(\frac{35\epsilon_0}{3 r_\lambda(x_0)}, \frac{10\widetilde{\nu}_\lambda(x_t)}{3}\max\left(\left(\frac{1}{2}\right)^{2^t}, \frac{10\rho}{3}\right)\right),$$

where the last inequality comes from using the induction hypothesis and the fact that $\frac{10\widetilde{\nu}_\lambda(x_t)}{3} \leq 1$.

Using once again the induction hypotheses and the fact that $t \leq \left\lfloor \log_2 \log_2 \frac{3}{10\rho} \right\rfloor$ which implies $\frac{10\rho}{3} \leq \left(\frac{1}{2}\right)^{2^t}$, we finally get

$$\frac{10\widetilde{\nu}_\lambda(x_{t+1})}{3} \leq \max\left(\frac{35\epsilon_0}{3 r_\lambda(x_0)}, \left(\frac{1}{2}\right)^{2^{t+1}}\right).$$

The fact that the second property holds for $t = t_c$ is trivial Now consider the case where $t \geq t_c$. Using the same technique as before but noting that in this case

$$\frac{10\widetilde{\nu}_\lambda(x_t)}{3} \leq \max\left(\frac{35\epsilon_0}{3r_\lambda(x_0)}, \left(\frac{10\rho}{3}\right)^{t-t_c+1}\right) \leq \max\left(\frac{35\epsilon_0}{3r_\lambda(x_0)}, \frac{10\rho}{3}\right),$$

We easily use Lemma 9 to reach the desired conclusion.

**3.** Let $t < t_c$. Then using Lemma 9:

$$\forall s \leq t, \; \nu_\lambda(x_{s+1}) \leq \max\left(3\epsilon_0, \max(\frac{10\rho}{3}, \frac{10\widetilde{\nu}_\lambda(x_s)}{3})\nu_\lambda(x_s)\right).$$

Using the fact that for any $s \leq t$, $\frac{10\widetilde{\nu}_\lambda(x_s)}{3} \leq \max(\frac{35\epsilon_0}{3r_\lambda(x_0)}, \left(\frac{1}{2}\right)^{2^s})$:

$$\forall s \leq t, \; \nu_\lambda(x_{s+1}) \leq \max\left(3\epsilon_0, \frac{35\epsilon_0}{3}\frac{\nu_\lambda(x_s)}{r_\lambda(x_0)}, \max(\frac{10\rho}{3}, \left(\frac{1}{2}\right)^{2^s})\nu_\lambda(x_s)\right).$$

Now using the fact that for any $s \leq t$, $\widetilde{\nu}_\lambda(x_s) \leq \frac{1}{7}$, we see that $\frac{\nu_\lambda(x_s)}{r_\lambda(x_0)} \leq \frac{7}{6}\widetilde{\nu}_\lambda(x_s) \leq \frac{1}{6}$ and hence $\frac{35\epsilon_0}{3}\frac{\nu_\lambda(x_s)}{r_\lambda(x_0)} \leq 3\epsilon_0$. Moreover, since $s \leq t < t_c$, $\max(\frac{10\rho}{3}, \left(\frac{1}{2}\right)^{2^s}) = \left(\frac{1}{2}\right)^{2^s}$. Thus:

$$\forall s \leq t, \; \nu_\lambda(x_{s+1}) \leq \max\left(3\epsilon_0, \left(\frac{1}{2}\right)^{2^s}\nu_\lambda(x_s)\right).$$

Combining these results yields:

$$\nu_\lambda(x_{t+1}) \leq \max\left(3\epsilon_0, \left(\frac{1}{2}\right)^{2^{t+1}-1}\nu_\lambda(x_0)\right).$$

This shows the first equation, that is:

$$\forall t \leq t_c, \; \nu_\lambda(x_t) \leq \max\left(3\epsilon_0, \left(\frac{1}{2}\right)^{2^t-1}\nu_\lambda(x_0)\right).$$

The case for $t \geq t_c$ is completely analogous. We can also reproduce the same proof to get the same bounds for $\widetilde{\nu}$, since the bounds in Lemma 9 are the same for both.

$\square$

## B.3 Main results in the paper

In the main paper, we mention two types of Newton method. First, we present a result of convergence on the full Newton method:

**Lemma 10** (Quadratic convergence of the full Newton method). *Let* $c \leq \frac{1}{7}$ *and* $x_0 \in D_\lambda(c)$. *Define*

$$x_{t+1} = x_t - \Delta_\lambda(x_t).$$

*Then this scheme converges quadratically, i.e.:*

$$\forall t \in \mathbb{N}, \; \frac{\nu_\lambda(x_t)}{\nu_\lambda(x_0)}, \frac{\widetilde{\nu}_\lambda(x_t)}{\widetilde{\nu}_\lambda(x_0)} \leq 2^{-(2^t-1)}.$$

*Thus :*

- $\forall t \in \mathbb{N}, \; x_t \in D_\lambda(c)$.

- *For any* $\tilde{c} \leq c$ *then* $\forall t \geq \left\lceil \log_2\left(1 + \log_2 \frac{c}{\tilde{c}}\right)\right\rceil, \; x_t \in D_\lambda(\tilde{c})$.

- *For any* $\varepsilon > 0$, $\forall t \geq \left\lceil \log_2\left(1 + \log_2 \frac{\nu_\lambda(x_0)}{\sqrt{\varepsilon}}\right)\right\rceil, \; \nu_\lambda(x_t) \leq \sqrt{\varepsilon}, \; f_\lambda(x) - f_\lambda(x_\lambda^\star) \leq \varepsilon$.

- *If we perform the Newton method and return the first $x_t$ such that $\nu_\lambda(x_t) \leq \sqrt{\varepsilon}$, then the number of Newton steps computations is at most $1 + \left\lceil \log_2 \left( 1 + \log_2 \frac{\nu_\lambda(x_0)}{\sqrt{\varepsilon}} \right) \right\rceil$.*

*Proof.* A full Newton method is an approximate Newton method where $\rho, \epsilon_0 = 0$. Thus apply Proposition 4; note that in this case $t_c = +\infty$. The last point shows that if $\mathsf{c} \leq \frac{1}{7}$, and if we perform the Newton method with a full Newton step, then

$$\forall t \geq 0, \ \widetilde{\nu}_\lambda(x_t) \leq 2^{-(2^t-1)}\nu_\lambda(x_0), \ \widetilde{\nu}_\lambda(x_t) \leq 2^{-(2^t-1)}\nu_\lambda(x_0).$$

This shows the quadratic convergence, and the first two points directly follow. For the third point, the result for $\nu_\lambda(x_t)$ directly follows from the previous equation, and the one on function values is a direct consequence of Lemma 6 and the fact that $x_t \in \mathsf{D}_\lambda(1/7)$.

For the last point, note that $\nu_t(x_t) = \nabla f_\lambda(x_t) \cdot \Delta_\lambda(x_t)$ is accessible. Moreover, the bound on $t$ is given in the point before, and since one has to compute $\Delta_\lambda(x_s)$ for $0 \leq s \leq t$, there are at most $t + 1$ computations. $\qquad\square$

In the main paper, we compute approximate Newton steps by considering methods which naturally yield only a relative error $\rho$ and no absolute error $\epsilon_0$. Indeed, we take the following notation.

**Approximate solutions to linear problems.** Let $\mathbf{A}$ be a positive definite Hermitian operator on $\mathcal{H}$, $b$ in $\mathcal{H}$, and a wanted relative precision $\rho$.

We say that $x$ is a $\rho$-relative approximation to the linear problem $\mathbf{A}x = b$ and write $x \in \mathrm{LinApprox}(\mathbf{A}, b, \rho)$ if the following holds:

$$\|\mathbf{A}^{-1}b - x\|_\mathbf{A} \leq \rho\|b\|_{\mathbf{A}^{-1}} = \rho\|\mathbf{A}^{-1}b\|_\mathbf{A}.$$

Note that if $x \in \mathrm{LinApprox}(\mathbf{A}, b, \rho)$ for $\rho < 1$, then

$$(1-\rho)\|b\|_{\mathbf{A}^{-1}} \leq x \cdot b \leq (1+\rho)\|b\|_{\mathbf{A}^{-1}}.$$

The following lemma shows that if, instead of computing the exact Newton step, we compute a relative approximation of the Newton step belonging to $\mathrm{LinApprox}(\mathbf{H}_\lambda(x), \nabla f_\lambda(x), \rho)$ for a given $\rho < 1$, then one has linear convergence. Moreover, we show that we can still perform a method which automatically stops.

**Proposition 5** (relative approximate Newton method). *Let $\lambda > 0$, $\rho \leq \frac{1}{7}$, $\mathsf{c} \leq \frac{1}{7}$ and a starting point $x_0 \in \mathsf{D}_\lambda(\mathsf{c})$. Assume we perform the following Newton scheme:*

$$\forall t \geq 0, \ x_{t+1} = x_t - \widetilde{\Delta}_t, \qquad \widetilde{\Delta}_t \in \mathrm{LinApprox}(\mathbf{H}_\lambda(x_t), \nabla f_\lambda(x_t), \rho).$$

*Then the scheme converges linearly, i.e.*

$$\forall t \in \mathbb{N}, \ \frac{\nu_\lambda(x_t)}{\nu_\lambda(x_0)}, \frac{\widetilde{\nu}_\lambda(x_t)}{\widetilde{\nu}_\lambda(x_0)} \leq 2^{-t}.$$

*Thus,*

- *$\forall t \in \mathbb{N}, \ x_t \in \mathsf{D}_\lambda(\mathsf{c})$.*

- *For any $\tilde{\mathsf{c}} \leq \mathsf{c}$ then $\forall t \geq \left\lceil \log_2 \frac{\mathsf{c}}{\tilde{\mathsf{c}}} \right\rceil$, $x_t \in \mathsf{D}_\lambda(\tilde{\mathsf{c}})$.*

- *For any $\varepsilon > 0$, $\forall t \geq \left\lceil \log_2 \frac{\nu_\lambda(x_0)}{\sqrt{\varepsilon}} \right\rceil$, $\nu_\lambda(x_t) \leq \sqrt{\varepsilon}$, $f_\lambda(x) - f_\lambda(x_\lambda^\star) \leq \varepsilon$*

- *If the method is performed and returns the first $x_t$ such that $x_t \cdot \widetilde{\Delta}_t \leq \frac{6}{7}\varepsilon$, then at most $2 + \left\lfloor \log_2 \left( \sqrt{\frac{4}{3}} \frac{\nu_\lambda(x_0)}{\sqrt{\varepsilon}} \right) \right\rfloor$ approximate Newton steps computations have been performed, and $\nu_\lambda(x_t) \leq \sqrt{\varepsilon}$, $f_\lambda(x) - f_\lambda(x_\lambda^\star) \leq \varepsilon$.*

*Proof.* Apply Proposition 4 with $\epsilon_0 = 0$ and $\rho = \frac{1}{7}$, since if $\rho \le \frac{1}{7}$, then a fortiori the approximation satisfies the condition for $\rho = \frac{1}{7}$. The last point clearly states that

$$\forall t \in \mathbb{N}, \ \frac{\nu_\lambda(x_t)}{\nu_\lambda(x_0)}, \frac{\widetilde{\nu}_\lambda(x_t)}{\widetilde{\nu}_\lambda(x_0)} \le \left(\frac{10}{21}\right)^t \le 2^{-t}.$$

From this, using Lemma 6 for the third point, the first three points are easily proven.

For the last point, note that since $\widetilde{\Delta}_t \in \mathrm{LinApprox}(\mathbf{H}_\lambda(x_t), \nabla f_\lambda(x_t), \rho)$, the following holds: $\nabla f_\lambda(x_t) \cdot \widetilde{\Delta}_t = \nu_\lambda(x_t)^2 + \nabla f_\lambda(x_t) \cdot \left(\widetilde{\Delta}_t - \mathbf{H}_\lambda^{-1}(x_t) \nabla f_\lambda(x_t)\right)$. Now bound

$$|\nabla f_\lambda(x_t) \cdot \left(\widetilde{\Delta}_t - \mathbf{H}_\lambda^{-1}(x_t) \nabla f_\lambda(x_t)\right)| \le \nu_\lambda(x_t) \|\widetilde{\Delta}_t - \mathbf{H}_\lambda^{-1}(x_t) \nabla f_\lambda(x_t)\|_{\mathbf{H}_\lambda(x_t)} \le \rho \nu_\lambda(x_t)^2.$$

Thus:

$$(1 - \rho)\nu_\lambda(x_t)^2 \le \nabla f_\lambda(x_t) \cdot \widetilde{\Delta}_t \le (1 + \rho)\nu_\lambda(x_t)^2.$$

Since $\rho = \frac{1}{7}$, we see that if $\nabla f_\lambda(x_t) \cdot \widetilde{\Delta}_t \le \frac{6}{7}\varepsilon$, then $\nu_\lambda(x_t)^2 \le \varepsilon$. Moreover, since we stop at the first $t$ where $\nabla f_\lambda(x_t) \cdot \widetilde{\Delta}_t \le \frac{6}{7}\varepsilon$, then if $t$ denotes the time at which we stop,

$$\frac{6}{7}\varepsilon < \nabla f_\lambda(x_{t-1}) \cdot \widetilde{\Delta}_{t-1} \le \frac{8}{7}\nu_\lambda(x_{t-1})^2 \implies \nu_\lambda(x_{t-1})^2 \ge \frac{3}{4}\varepsilon.$$

Since $\nu_\lambda(x_{t-1})^2 \le 2^{-2(t-1)}\nu_\lambda(x_0)^2$, this implies in turn that $t - 1 \le \log_2 \left(\sqrt{\frac{4}{3}} \frac{\nu_\lambda(x_0)}{\sqrt{\varepsilon}}\right)$. Thus, necessarily, $t \le 1 + \left\lfloor \log_2 \left(\sqrt{\frac{4}{3}} \frac{\nu_\lambda(x_0)}{\sqrt{\varepsilon}}\right) \right\rfloor$, and since we compute approximate Newton steps for $s = 0, ..., t$, we finally have that the number of approximate Newton steps is bounded by

$$2 + \left\lfloor \log_2 \left(\sqrt{\frac{4}{3}} \frac{\nu_\lambda(x_0)}{\sqrt{\varepsilon}}\right) \right\rfloor.$$

$\square$

Last but not least, we summarize all these theorem in the following simple result.

**Lemma 11.** *Let $\lambda > 0$, $c \le 1/7$, let $f_\lambda$ be generalized self-concordant and $x \in \mathsf{D}_\lambda(c)$. It holds: $\frac{1}{4}\nu_\lambda(x)^2 \le f_\lambda(x) - f_\lambda(x_\lambda^\star) \le \nu_\lambda(x)^2$. Moreover, the full Newton method starting from $x_0$ has quadratic convergence, i.e. if $x_t$ is obtained via $t \in \mathbb{N}$ steps of the Newton method Eq. (2), then $\nu_\lambda(x_t) \le 2^{-(2^t-1)}\nu_\lambda(x_0)$. Finally, the approximate Newton method starting from $x_0$ has linear convergence, i.e. if $x_t$ is obtained via $t \in \mathbb{N}$ steps of Eq. (3), with $\widetilde{\Delta}_t \in \mathrm{LinApprox}(\mathbf{H}_\lambda(x_t), \nabla f_\lambda(x_t), \rho)$ and $\rho \le 1/7$, then $\nu_\lambda(x_t) \le 2^{-t}\nu_\lambda(x_0)$.*

*Proof.* The three points are obtained in the following lemmas, assuming $x \in \mathsf{D}_\lambda(1/7)$.

- For $\frac{1}{4}\nu_\lambda(x)^2 \le f_\lambda(x) - f_\lambda(x_\lambda^\star) \le \nu_\lambda(x)^2$, see Lemma 6 in Appendix A.1.

- The convergence rate of the full Newton method starting in $\mathsf{D}_\lambda(1/7)$ is obtained in Lemma 10.

- The convergence rate of the approximate Newton method starting in $\mathsf{D}_\lambda(1/7)$ is obtained in Proposition 5.

$\square$

### B.4 Sketching the Hessian only once in each Dikin ellispoid

In this section, we provide a lemma which shows in essence that if we are in a small Dikin ellipsoid, then we can keep the Hessian of the starting point and compute approximations of $\mathbf{H}_\lambda^{-1}(x_0)\nabla f_\lambda(x_t)$; they will be good approximations to $\mathbf{H}_\lambda^{-1}(x_t)\nabla f_\lambda(x_t)$ as well.

**Lemma 12.** *Let* $c < 1$ *and* $x_0 \in D_\lambda(c)$ *be fixed.*

*Let* $\widetilde{\mathbf{H}}$ *be an approximation of the Hessian at* $x_0$, *approximation wich we quantify with*

$$t := \|\mathbf{H}_\lambda^{-1/2}(x_0)\left(\mathbf{H}_\lambda(x_0) - \widetilde{\mathbf{H}}\right)\mathbf{H}_\lambda^{-1/2}(x_0)\|.$$

*Assume*

$$1 + t < 2(1 - c)^2.$$

*Let* $b \in \mathcal{H}$. *If* $\widetilde{\Delta} \in \mathrm{LinApprox}(\widetilde{\mathbf{H}}_\lambda, b, \tilde{\rho})$, *then*

$$\forall x \in D_\lambda(c), \ \widetilde{\Delta} \in \mathrm{LinApprox}(\mathbf{H}_\lambda(x), b, \rho), \ \rho = \frac{(\tilde{\rho} - 1)(1 - c)^2 + (1 + t)}{2(1 - c)^2 - (1 + t)}.$$

*In particular, if* $c \leq \frac{1}{30}$, $x_0 \in D_\lambda(c)$,

$$\forall x \in D_\lambda(c), \ \forall b \in \mathcal{H}, \ \widetilde{\Delta} \in \mathrm{LinApprox}(\mathbf{H}_\lambda(x_0), b, \frac{1}{20}) \implies \widetilde{\Delta} \in \mathrm{LinApprox}(\mathbf{H}_\lambda(x), b, \frac{1}{7}).$$

*Proof.* First, start with a general theoretical result.

**1.** Let $\mathbf{A}$ and $\mathbf{B}$ be two positive semi-definite hermitian operators. Let $\lambda > 0$, $b \in \mathcal{H}$ and $\widetilde{\Delta} \in \mathrm{LinApprox}(\mathbf{B}_\lambda, b, \tilde{\rho})$. Decompose

$$\|\mathbf{A}_\lambda^{-1}b - \widetilde{\Delta}\|_{\mathbf{A}_\lambda} \leq \|\mathbf{A}_\lambda^{-1}b - \mathbf{B}_\lambda^{-1}b\|_{\mathbf{A}_\lambda} + \|\mathbf{B}_\lambda^{-1}b - \widetilde{\Delta}\|_{\mathbf{A}_\lambda}$$

$$\leq \|\mathbf{A}_\lambda^{1/2}(\mathbf{A}_\lambda^{-1} - \mathbf{B}_\lambda^{-1})\mathbf{A}_\lambda^{1/2}\| \, \|b\|_{\mathbf{A}_\lambda^{-1}} + \|\mathbf{A}_\lambda^{1/2}\mathbf{B}_\lambda^{-1/2}\| \, \|\mathbf{B}_\lambda^{-1}b - \widetilde{\Delta}\|_{\mathbf{B}_\lambda}.$$

Now using the fact that $\mathbf{A}_\lambda^{-1} - \mathbf{B}_\lambda^{-1} = \mathbf{B}_\lambda^{-1}(\mathbf{B} - \mathbf{A})\mathbf{A}_\lambda^{-1}$,

$$\|\mathbf{A}_\lambda^{1/2}(\mathbf{A}_\lambda^{-1} - \mathbf{B}_\lambda^{-1})\mathbf{A}_\lambda^{1/2}\| \leq \|\mathbf{A}_\lambda^{-1/2}(\mathbf{B} - \mathbf{A})\mathbf{A}_\lambda^{-1/2}\| \, \|\mathbf{A}_\lambda^{1/2}\mathbf{B}_\lambda^{-1}\mathbf{A}_\lambda^{1/2}\|$$

$$= \|\mathbf{A}_\lambda^{-1/2}(\mathbf{B} - \mathbf{A})\mathbf{A}_\lambda^{-1/2}\| \, \|\mathbf{A}_\lambda^{1/2}\mathbf{B}_\lambda^{-1/2}\|^2.$$

Moreover,

$$\|\mathbf{B}_\lambda^{-1}b - \widetilde{\Delta}\|_{\mathbf{B}_\lambda} \leq \tilde{\rho}\|b\|_{\mathbf{B}_\lambda^{-1}} \leq \|\mathbf{A}^{1/2}\mathbf{B}^{-1/2}\| \, \|b\|_{\mathbf{A}_\lambda^{-1}}.$$

Putting things together, and noting that from Lemma 21, $\|\mathbf{A}^{1/2}\mathbf{B}^{-1/2}\|^2 \leq \frac{1}{1 - \|\mathbf{A}_\lambda^{-1/2}(\mathbf{B}-\mathbf{A})\mathbf{A}_\lambda^{-1/2}\|}$ as soon as $\|\mathbf{A}_\lambda^{-1/2}(\mathbf{B} - \mathbf{A})\mathbf{A}_\lambda^{-1/2}\| < 1$, it holds:

$$\widetilde{\Delta} \in \mathrm{LinApprox}(\mathbf{A}_\lambda, b, \rho), \ \rho = \frac{\tilde{\rho} + \|\mathbf{A}_\lambda^{-1/2}(\mathbf{B} - \mathbf{A})\mathbf{A}_\lambda^{-1/2}\|}{1 - \|\mathbf{A}_\lambda^{-1/2}(\mathbf{B} - \mathbf{A})\mathbf{A}_\lambda^{-1/2}\|}.$$

The aim is now to apply this lemma to $\mathbf{A} = \mathbf{H}(x)$ and $\mathbf{B} = \widetilde{\mathbf{H}}$.

**2.** Let $x, x_0 \in D_\lambda(c)$. Using Lemma 22, we see that

$$1 + \|\mathbf{H}_\lambda^{-1/2}(x)(\widetilde{\mathbf{H}} - \mathbf{H}(x))\mathbf{H}_\lambda^{-1/2}(x)\| \leq (1 + t)(1 + \|\mathbf{H}_\lambda^{-1/2}(x)(\mathbf{H}(x_0) - \mathbf{H}(x))\mathbf{H}_\lambda^{-1/2}(x)\|).$$

Using Eq. (17), it holds:

$$(e^{-t(x-x_0)} - 1)\mathbf{I} \preceq \mathbf{H}_\lambda^{-1/2}(x)(\mathbf{H}(x_0) - \mathbf{H}(x))\mathbf{H}_\lambda^{-1/2}(x) \preceq (e^{t(x_0-x)} - 1)\mathbf{I}.$$

Thus,

$$\|\mathbf{H}_\lambda^{-1/2}(x)(\mathbf{H}(x_0) - \mathbf{H}(x))\mathbf{H}_\lambda^{-1/2}(x)\| \leq \max(1 - e^{-t(x-x_0)}, e^{t(x-x_0)} - 1) = e^{t(x-x_0)} - 1.$$

Finally, using the fact that $x_0, x \in D_\lambda(c)$ for $c < 1$ yields $t(x - x_0) \leq 2\log\frac{1}{1-c}$. Hence

$$1 + \|\mathbf{H}_\lambda^{-1/2}(x)(\mathbf{H}(x_0) - \mathbf{H}(x))\mathbf{H}_\lambda^{-1/2}(x)\| \leq \frac{1}{(1 - c)^2}.$$

Thus,

$$\|\mathbf{H}_\lambda^{-1/2}(x)(\widetilde{\mathbf{H}} - \mathbf{H}(x))\mathbf{H}_\lambda^{-1/2}(x)\| \leq \frac{1 + t}{(1 - c)^2} - 1.$$

The result then follows. $\qquad\square$

# C  Proof of bounds for the globalization scheme

In this section, we prove that the scheme of decreasing $\mu$ towards $\lambda$ converges.

## C.1  Main technical lemmas

**Lemma 13** (Next $\mu$). *Let $\mu > 0$, $\mathsf{c} < 1$.*

$$\nu_\mu(x) \leq \frac{\mathsf{c}}{3} \frac{\sqrt{\mu}}{R} \implies \nu_{\widetilde{\mu}}(x) \leq \mathsf{c} \frac{\sqrt{\widetilde{\mu}}}{R}, \qquad \widetilde{\mu} := q\,\mu, \qquad q \geq \frac{\frac{1}{3} + \frac{R\sqrt{\mu}\|x\|_{\mathbf{H}_\mu^{-1}(x)}}{\mathsf{c}}}{1 + \frac{R\sqrt{\mu}\|x\|_{\mathbf{H}_\mu^{-1}(x)}}{\mathsf{c}}}.$$

$$x \in \mathsf{D}_\mu\left(\frac{\mathsf{c}}{3}\right) \implies x \in \mathsf{D}_{\widetilde{\mu}}(\mathsf{c}), \qquad \widetilde{\mu} := q\,\mu, \qquad q \geq \frac{\frac{1}{3} + \frac{\mu\|x\|_{\mathbf{H}_\mu^{-1}(x)}}{\mathsf{c}\,\mathsf{r}_\mu(x)}}{1 + \frac{\mu\|x\|_{\mathbf{H}_\mu^{-1}(x)}}{\mathsf{c}\,\mathsf{r}_\mu(x)}}.$$

*Proof.* For any $\widetilde{\mu} < \mu$, note that

$$\forall x \in \mathcal{H}, \ \|\mathbf{H}_{\widetilde{\mu}}^{-1/2}(x)\mathbf{H}_\mu^{1/2}(x)\| = \sqrt{\frac{\lambda_{\min}(\mathbf{H}(x)) + \mu}{\lambda_{\min}(\mathbf{H}(x)) + \widetilde{\mu}}} \leq \sqrt{\mu/\widetilde{\mu}}.$$

This shows that $\|\cdot\|_{\mathbf{H}_{\widetilde{\mu}}^{-1}(x)} \leq \sqrt{\frac{\mu}{\widetilde{\mu}}} \|\cdot\|_{\mathbf{H}_\mu^{-1}(x)}$, and in particular that $\frac{1}{\mathsf{r}_{\widetilde{\mu}}(x)} \leq \sqrt{\mu/\widetilde{\mu}}\frac{1}{\mathsf{r}_\mu(x)}$.

Using this fact, it holds:

$$\begin{aligned}
\widetilde{\nu}_{\widetilde{\mu}}(x) &= \frac{\|\nabla f_{\widetilde{\mu}}(x)\|_{\mathbf{H}_{\widetilde{\mu}}^{-1}(x)}}{\mathsf{r}_{\widetilde{\mu}}(x)} \\
&= \frac{\|\nabla f_\mu(x) - (\mu - \widetilde{\mu})x\|_{\mathbf{H}_{\widetilde{\mu}}^{-1}(x)}}{\mathsf{r}_{\widetilde{\mu}}(x)} \\
&\leq \frac{\mu}{\widetilde{\mu}} \frac{\|\nabla f_\mu(x)\|_{\mathbf{H}_\mu^{-1}(x)}}{\mathsf{r}_\mu(x)} + \left(\frac{\mu}{\widetilde{\mu}} - 1\right) \frac{\|\mu x\|_{\mathbf{H}_\mu^{-1}(x)}}{\mathsf{r}_\mu(x)}.
\end{aligned}$$

Hence, if $\widetilde{\nu}_\mu(x) \leq \frac{\mathsf{c}}{3}$, a condition to obtain $\widetilde{\nu}_{\widetilde{\mu}}(x) \leq \mathsf{c}$ is the following:

$$\frac{\mu}{\widetilde{\mu}}\left(\frac{\mathsf{c}}{3} + t\right) \leq \mathsf{c} + t \Leftrightarrow \widetilde{\mu} \geq \mu\frac{\mathsf{c}/3 + t}{\mathsf{c} + t} \qquad t = \frac{\|\mu x\|_{\mathbf{H}_\mu^{-1}(x)}}{\mathsf{r}_\mu(x)}.$$

This yields the second point of the lemma. The analysis is completely analoguous for the first. $\qquad\square$

**Lemma 14** (Useful bounds for $q$). *Let $\mu > 0$. Then the following hold:*

$$\forall x \in \mathcal{H}, \ \frac{\mu\|x\|_{\mathbf{H}_\mu^{-1}(x)}}{\mathsf{r}_\mu(x)} \leq R\sqrt{\mu}\|x\|_{\mathbf{H}_\mu^{-1}(x)} \leq R\|x\|.$$

*Moreover, we can bound all of these quantities using $x_\mu^\star$:*

- *For any $\mathsf{c} < 1$, $x \in \mathcal{H}$, if $x \in \mathsf{D}_\mu(\mathsf{c}/3)$, then the following holds:*

$$\frac{\mu\|x\|_{\mathbf{H}_\mu^{-1}(x)}}{\mathsf{c}\,\mathsf{r}_\mu(x)} \leq \frac{1}{3}\left(1 + \frac{1}{1 - \mathsf{c}/3}\right) + \frac{1}{1 - \mathsf{c}/3}\frac{\|\mu x_\mu^\star\|_{\mathbf{H}_\mu^{-1}(x_\mu^\star)}}{\mathsf{c}\,\mathsf{r}_\mu(x_\mu^\star)}.$$

- *For any $\mathsf{c} < 1$, $x \in \mathcal{H}$, if $\frac{R\nu_\mu(x)}{\sqrt{\mu}} \leq \frac{\mathsf{c}}{3}$, then the following holds:*

$$\frac{R\sqrt{\mu}\|x\|_{\mathbf{H}_\mu^{-1}(x)}}{\mathsf{c}} \leq \left(1 + \frac{1}{1 - \mathsf{c}/3}\right)\frac{1}{3} + \sqrt{\frac{1}{1 - \mathsf{c}/3}}\frac{R\sqrt{\mu}\|x_\mu^\star\|_{\mathbf{H}_\mu^{-1}(x_\mu^\star)}}{\mathsf{c}}.$$

*Likewise, it can be shown that under the same conditions:*

$$\frac{R\|x\|}{\mathsf{c}} \leq \frac{R\|x_\mu^\star\|}{\mathsf{c}} + \frac{1}{3}\overline{\phi}(-\log(1 - \mathsf{c}/3)).$$

*Proof.* The first bound is obvious. Moreover, the fact that $\widetilde{\nu}_\mu(x) \leq \frac{c}{3}$ implies that $\mathsf{t}(x - x_\mu^\star) \leq \log \frac{1}{1-c/3}$. Thus, we get the classical bounds on the Hessian using Eq. (14):

$$e^{-\mathsf{t}(x-x_\mu^\star)}\mathbf{H}(x) \preceq \mathbf{H}(x_\mu^\star) \preceq e^{\mathsf{t}(x-x_\mu^\star)}\mathbf{H}(x).$$

**1. Bound on $\mu\|x\|_{\mathbf{H}_\mu^{-1}(x)}$.** Using Eqs. (17) and (18),

$$\mu\|x\|_{\mathbf{H}_\mu^{-1}(x)} = \|\nabla f_\mu(x) - \nabla f(x) + \nabla f(x_\mu^\star) - \nabla f(x_\mu^\star)\|_{\mathbf{H}_\mu^{-1}(x)}$$

$$\leq \nu_\mu(x) + \int_0^1 \|\mathbf{H}_\mu(x)^{-1/2}\mathbf{H}(x_t)(x - x_\mu^\star)\| \, dt + \|\nabla f(x_\mu^\star)\|_{\mathbf{H}_\mu(x)}, \; x_t = tx + (1-t)x_\mu^\star.$$

Now bound $\|\mathbf{H}_\mu(x)^{-1/2}\mathbf{H}(x_t)(x - x_\mu^\star)\| \leq \|\mathbf{H}_\mu(x)^{-1/2} \, \mathbf{H}_\mu(x_t)^{1/2}\| \, \|x - x_\mu^\star\|_{\mathbf{H}(x_t)}$ and use Eq. (17) and Eq. (14) to get:

$$\|\mathbf{H}_\mu(x)^{-1/2}\mathbf{H}(x_t)(x - x_\mu^\star)\| \leq e^{t\,\mathsf{t}(x-x_\mu^\star)}\|x - x_\mu^\star\|_{\mathbf{H}(x)}.$$

Integrating this yields:

$$\int_0^1 \|\mathbf{H}_\mu(x)^{-1/2}\mathbf{H}(x_t)(x - x_\mu^\star)\| \, dt \leq \overline{\phi}(\mathsf{t}(x - x_\mu^\star)) \, \|x - x_\mu^\star\|_{\mathbf{H}(x)} \leq e^{\mathsf{t}(x-x_\mu^\star)} \, \nu_\mu(x).$$

Where the last inequality is obtained using the bounds between gradient and hessian distance Eq. (18). Finally, using the bound on $\mathsf{t}(x - x_\mu^\star)$,

$$\mu\|x\|_{\mathbf{H}_\mu^{-1}(x)} \leq \left(1 + \frac{1}{1-c/3}\right)\nu_\mu(x) + \sqrt{\frac{1}{1-c/3}}\|\nabla f(x_\mu^\star)\|_{\mathbf{H}_\mu^{-1}(x_\mu^\star)}.$$

**2. Bound on $R\|x\|$.** Start by decomposing

$$R\|x\| \leq R\|x_\mu^\star\| + R\|x - x_\mu^\star\|.$$

Now bound

$$R\|x - x_\mu^\star\| \leq \frac{R}{\sqrt{\mu}}\|x - x_\mu^\star\|_{\mathbf{H}_\mu(x)}.$$

Using Eq. (17), $\|x - x_\mu^\star\|_{\mathbf{H}_\mu(x)} \leq \overline{\phi}(-\log(1 - c/3))\nu_\mu(x)$. Hence:

$$R\|x\| \leq R\|x_\mu^\star\| + \overline{\phi}(-\log(1 - c/3))\frac{R\nu_\mu(x)}{\sqrt{\mu}}.$$

**3. Now assume $x \in \mathsf{D}_\mu(c/3)$.** Using the bound on $\mu\|x\|_{\mathbf{H}_\mu^{-1}(x)}$, and noting that

$$\frac{1}{\mathsf{r}_\mu(x)} \leq e^{\mathsf{t}(x-x_\mu^\star)/2}\frac{1}{\mathsf{r}_\mu(x_\mu^\star)},$$

it holds:

$$\frac{\mu\|x\|_{\mathbf{H}_\mu^{-1}(x)}}{\mathsf{c}\,\mathsf{r}_\mu(x)} \leq \frac{1}{3}\left(1 + \frac{1}{1-c/3}\right) + \frac{1}{1-c/3}\frac{\|\mu x_\mu^\star\|_{\mathbf{H}_\mu^{-1}(x_\mu^\star)}}{\mathsf{c}\,\mathsf{r}_\mu(x_\mu^\star)}.$$

**4. Now assume $\frac{R\nu_\mu(x)}{\sqrt{\mu}} \leq \frac{c}{3}$.** We know that in particular, $x \in \mathsf{D}_\mu(c/3)$ and hence:

$$R\sqrt{\mu}\|x\|_{\mathbf{H}_\mu^{-1}(x)} \leq \left(1 + \frac{1}{1-c/3}\right)\frac{R\nu_\mu(x)}{\sqrt{\mu}} + \sqrt{\frac{1}{1-c/3}}\frac{R\mu\|x_\mu^\star\|_{\mathbf{H}_\mu^{-1}(x_\mu^\star)}}{\sqrt{\mu}}$$

$$\leq \left(1 + \frac{1}{1-c/3}\right)\frac{c}{3} + \sqrt{\frac{1}{1-c/3}}R\sqrt{\mu}\|x_\mu^\star\|_{\mathbf{H}_\mu^{-1}(x_\mu^\star)}.$$

Hence

$$\frac{R\sqrt{\mu}\|x\|_{\mathbf{H}_\mu^{-1}(x)}}{\mathsf{c}} \leq \left(1 + \frac{1}{1 - \mathsf{c}/3}\right)\frac{1}{3} + \sqrt{\frac{1}{1 - \mathsf{c}/3}}\frac{R\sqrt{\mu}\|x_\mu^\star\|_{\mathbf{H}_\mu^{-1}(x_\mu^\star)}}{\mathsf{c}}.$$

Likewise:

$$\frac{R\|x\|}{\mathsf{c}} \leq \frac{R\|x_\mu^\star\|}{\mathsf{c}} + \frac{1}{3}\overline{\phi}(-\log(1 - \mathsf{c}/3)).$$

$\square$

We can get the following simpler bounds.

**Corollary 3** (Application to $\mathsf{c} = \frac{1}{7}$). *Applying Lemma 14 to $\mathsf{c} = \frac{1}{7}$, we get the following bounds. Let $\mu > 0$.*

- *For any $x \in \mathcal{H}$, if $x \in \mathsf{D}_\mu(\mathsf{c}/3)$, then the following holds:*

$$\frac{7\mu\|x\|_{\mathbf{H}_\mu^{-1}(x)}}{\mathsf{r}_\mu(x)} \leq 1 + \frac{8\|\mu x_\mu^\star\|_{\mathbf{H}_\mu^{-1}(x_\mu^\star)}}{\mathsf{r}_\mu(x_\mu^\star)}.$$

- *For any $\mathsf{c} < 1$, $x \in \mathcal{H}$, if $\frac{R\nu_\mu(x)}{\sqrt{\mu}} \leq \frac{\mathsf{c}}{3}$, then the following hold:*

$$7R\sqrt{\mu}\|x\|_{\mathbf{H}_\mu^{-1}(x)} \leq 1 + 8R\sqrt{\mu}\|x_\mu^\star\|_{\mathbf{H}_\mu^{-1}(x_\mu^\star)}.$$

$$7R\|x\| \leq 7R\|x_\mu^\star\| + 1.$$

## C.2 Proof of main theorems

In this section, we bound the number of iterations of our scheme in different cases.

Recall the proposed globalization scheme in the paper, where $\mathtt{ANM}_\rho(f, x, t)$ is a method performing $t$ successive $\rho$-relative approximate Newton steps of $f$ starting at $x$.

---

**Proposed Globalization Scheme**

*Phase I: Getting in the Dikin ellispoid of $f_\lambda$*

Start with $x_0 \in \mathcal{H}, \mu_0 > 0, t, T \in \mathbb{N}$ and $(q_k)_{k\in\mathbb{N}} \in (0, 1]$.
For $k \in \mathbb{N}$
    $x_{k+1} \leftarrow \mathtt{ANM}_\rho(f_{\mu_k}, x_k, t)$
    $\mu_{k+1} \leftarrow q_{k+1}\mu_k$
Stop when $\mu_{k+1} < \lambda$ and set $x_{last} \leftarrow x_k$. $K \leftarrow k$

*Phase II: reach a certain precision starting from inside the Dikin ellipsoid*

Return $\widehat{x} \leftarrow \mathtt{ANM}_\rho(f_\lambda, x_{last}, T)$

---

Throughout this section, we will denote with $K$ the value of $k$ when the scheme stops, i.e. the first value of $k$ such that $\mu_{k+1} < \lambda$.

**Adaptive methods** We start by presenting an adaptive way to select $\mu_{k+1}$ from $\mu_k$, with theoretical guarantees. The main result is the following.

**Proposition 6** (Adaptive, simple version). *Assume that we perform phase I starting at $x_0$ such that*

$$\frac{R\nu_{\mu_0}(x_0)}{\sqrt{\mu_0}} \leq \frac{1}{7}.$$

*Assume that at each step $k$, we compute $x_{k+1}$ using $t = 2$ iterations of the $\rho$-relative approximate Newton method. Then if at each iteration, we set:*

$$\mu_{k+1} = q_{k+1}\,\mu_k, \qquad q_{k+1} := \frac{\frac{1}{3} + 7R\|x_{k+1}\|}{1 + 7R\|x_{k+1}\|}.$$

*Then the following hold:*

1. $\forall k \leq K+1, \ \frac{R\nu_{\mu_k}(x_k)}{\sqrt{\mu_k}} \leq \frac{1}{7}$ .

2. *The decreasing parameter $q_{k+1}$ is bounded above before reaching $K$:*

$$\forall k \leq K, \ q_{k+1} \leq \frac{\frac{4}{3} + 7R\|x^\star_{\mu_k}\|}{2 + 7R\|x^\star_{\mu_k}\|} \leq \frac{\frac{4}{3} + 7R\|x^\star_\lambda\|}{2 + 7R\|x^\star_\lambda\|}.$$

3. *$K$ is finite,*

$$K \leq \left\lceil \frac{\log \frac{\mu_0}{\lambda}}{\log \frac{2+7R\|x^\star_\lambda\|}{\frac{4}{3}+7R\|x^\star_\lambda\|}} \right\rceil \leq \left\lfloor (3 + 11R\|x^\star_\lambda\|) \log \frac{\mu_0}{\lambda} \right\rfloor,$$

*and $\frac{R\nu_\lambda(x_{K+1})}{\sqrt{\lambda}} \leq \frac{1}{7}$.*

*Proof.* Let us prove the three points one by one.

1. This is easily proved by induction, the keys to the induction hypothesis being:

   - Using the induction hypothesis, $x_k \in \mathsf{D}_{\mu_k}(\mathsf{c})$ and hence, using Proposition 5 shows that after two iterations of the approximate Newton scheme, $\frac{\nu_{\mu_k}(x_{k+1})}{\nu_{\mu_k}(x_k)} \leq \frac{1}{3}$ which implies $\frac{R\nu_{\mu_k}(x_{k+1})}{\sqrt{\mu_k}} \leq \frac{\mathsf{c}}{3}$.

   - Now using Lemma 13, we see that that since

   $$7R\|x_{k+1}\| = \frac{R\|x_{k+1}\|}{\mathsf{c}} \geq \frac{R\sqrt{\mu_k}\|x_{k+1}\|_{\mathbf{H}^{-1}_{\mu_k}(x_{k+1})}}{\mathsf{c}},$$

   the hypotheses to guarantee the bound for $q_{k+1}$ hold, hence

   $$\frac{R\nu_{\mu_{k+1}}(x_{k+1})}{\sqrt{\mu_{k+1}}} \leq \mathsf{c}.$$

2. Using the second bullet point of Cor. 3, we see that the previous point implies

$$\forall k \leq K, \ 7R\|x_{k+1}\| \leq 7R\|x^\star_{\mu_k}\| + 1 \implies q_{k+1} \leq \frac{4/3 + 7R\|x^\star_{\mu_k}\|}{2 + 7R\|x^\star_{\mu_k}\|}.$$

Now using the fact that for any $k \leq K$, $\mu_k > \lambda$, we can use the simple fact that $\|x^\star_\lambda\| \geq \|x^\star_{\mu_k}\|$ to get the desired bound for $q_{k+1}$.

3. Using the previous point clearly shows the following bound:

$$\forall k \leq K+1, \ \mu_k \leq \left( \frac{\frac{4}{3} + 7R\|x^\star_\lambda\|}{2 + 7R\|x^\star_\lambda\|} \right)^k \mu_0.$$

As this clearly converges to $0$ when $k$ goes to infinity, $K$ is necessarily finite. Applying this for $k = K$, we see that:

$$\lambda \leq \mu_K \leq \left( \frac{\frac{4}{3} + 7R\|x^\star_\lambda\|}{2 + 7R\|x^\star_\lambda\|} \right)^K \mu_0.$$

This shows that $K \leq \frac{\log \frac{\mu_0}{\lambda}}{\log \frac{2+7R\|x^\star_\lambda\|}{\frac{4}{3}+7R\|x^\star_\lambda\|}}$.

The final bound is obtained noting that

$$\frac{2 + 7R\|x^\star_\lambda\|}{\frac{4}{3} + 7R\|x^\star_\lambda\|} = 1 + \frac{1}{t}, \qquad t = 2 + \frac{21}{2}R\|x^\star_\lambda\|,$$

and using the classical bound:
$$\frac{1}{\log(1 + \frac{1}{t})} \leq t + 1.$$

Finally, the fact that $\frac{R\nu_\lambda(x_{K+1})}{\sqrt{\lambda}} \leq \mathsf{c}$ is just a consequence of the fact that $\mu_{K+1} \leq \lambda \leq \mu_K$ and thus that $\lambda = q\mu_K$ with $q \geq q_{K+1}$, which is shown to satisfy the condition in Lemma 13. Hence, the lemma holds not only for $\mu_{K+1}$ but also for $\lambda$. $\qquad \square$

**Remark 2** ($\mu_0$). *In the previous proposition, we assume start at $x_0, \mu_0$ such that*
$$\frac{R\nu_{\mu_0}(x_0)}{\sqrt{\mu_0}} \leq \frac{1}{7}.$$

*A simple way to have such a pair is simply to select:*
$$x_0 = 0, \ \mu_0 = 7R\|\nabla f(0)\|,$$

*since* $\frac{R\nu_{\mu_0}(x_0)}{\sqrt{\mu_0}} = \frac{R\|\nabla f(0)\|_{\mathbf{H}_{\mu_0}^{-1}(0)}}{\sqrt{\mu_0}} \leq \frac{R\|\nabla f(0)\|}{\mu_0}.$

Alternatively, if one can approximately compute $\|x\|_{\mathbf{H}_\mu^{-1}(x)}$, one can propose the following variant, whose proof is completely analogous.

**Proposition 7** (Adaptive, small variant version). *Assume that we perform phase I starting at $x_0$ such that*
$$\frac{R\nu_{\mu_0}(x)}{\sqrt{\mu_0}} \leq \frac{1}{7}.$$

*Then if at each iteration, we set:*
$$t_{k+1} = 7\sqrt{\frac{7}{6}}R\sqrt{\mu_k}\sqrt{x_{k+1} \cdot s_{k+1}}, \ s_{k+1} \in \mathrm{LinApprox}(\mathbf{H}_{\mu_k}(x_{k+1}), x_{k+1}, \frac{1}{7}),$$

*and*
$$\mu_{k+1} = q_{k+1}\,\mu_k, \qquad q_{k+1} := \frac{\frac{1}{3} + t_{k+1}}{1 + t_{k+1}}.$$

*Then the following hold:*

**1.** $\forall k \leq K, \ \frac{R\nu_{\mu_k}(x_k)}{\sqrt{\mu_k}} \leq \frac{1}{7}.$

**2.** *The decreasing parameter $q_{k+1}$ is bounded above before reaching $K$:*
$$\forall k \leq K, \ q_{k+1} \leq \sup_{\mu_0 \geq \mu \geq \lambda} \frac{\frac{7}{3} + 10R\sqrt{\mu}\|x_\mu^\star\|_{\mathbf{H}_\mu^{-1}(x_\mu^\star)}}{3 + 10R\sqrt{\mu}\|x_\mu^\star\|_{\mathbf{H}_\mu^{-1}(x_\mu^\star)}} \leq \frac{\frac{7}{3} + 10R\|x_\lambda^\star\|}{3 + 10R\|x_\lambda^\star\|}.$$

**3.** *$K$ is finite,*
$$K \leq \left(\frac{9}{2} + 15\sup_{\lambda \leq \mu \leq \mu_0} R\sqrt{\mu}\|x_\mu^\star\|_{\mathbf{H}_\mu^{-1}(x_\mu^\star)}\right)\log\frac{\mu_0}{\lambda},$$

*and* $\frac{R\nu_\lambda(x_{K+1})}{\sqrt{\lambda}} \leq \frac{1}{7}.$

*Proof.* The main thing to note is that because of the properties of $\frac{1}{7}$-approximations, if $s_{k+1} \in \mathrm{LinApprox}(\mathbf{H}_{\mu_k}(x_{k+1}), x_{k+1}, \frac{1}{7})$,
$$(1 - \frac{1}{7})\|x_{k+1}\|_{\mathbf{H}_{\mu_k}^{-1}(x_{k+1})}^2 \leq x_{k+1} \cdot s_{k+1} \leq (1 + \frac{1}{7})\|x_{k+1}\|_{\mathbf{H}_{\mu_k}^{-1}(x_{k+1})}^2.$$

Hence,
$$\|x_{k+1}\|_{\mathbf{H}_{\mu_k}^{-1}(x_{k+1})} \leq \sqrt{\frac{7}{6}}\sqrt{x_{k+1} \cdot s_{k+1}} \leq \sqrt{\frac{4}{3}}\|x_{k+1}\|_{\mathbf{H}_{\mu_k}^{-1}(x_{k+1})}.$$

Hence, $t_{k+1} \geq 7R\sqrt{\mu_k}\|x_{k+1}\|_{\mathbf{H}_{\mu_k}^{-1}(x_{k+1})}$, and we can apply Lemma 13 to get the first point. To get the second point, we bound $t_{k+1}$ above:

$$t_{k+1} \leq 7\sqrt{\frac{4}{3}}R\sqrt{\mu_k}\|x_{k+1}\|_{\mathbf{H}_{\mu_k}^{-1}(x_{k+1})}.$$

Now use Cor. 3 to find:

$$t_{k+1} \leq \sqrt{\frac{4}{3}}\left(1 + 8R\sqrt{\mu_k}\|x^\star_{\mu_k}\|_{\mathbf{H}_{\mu_k}^{-1}(x^\star_{\mu_k})}\right) \leq 2 + 10R\sqrt{\mu_k}\|x^\star_{\mu_k}\|_{\mathbf{H}_{\mu_k}^{-1}(x^\star_{\mu_k})}.$$

Thus,

$$q_{k+1} \leq \frac{\frac{7}{3} + 10R\sqrt{\mu_k}\|x^\star_{\mu_k}\|_{\mathbf{H}_{\mu_k}^{-1}(x^\star_{\mu_k})}}{3 + 10R\sqrt{\mu_k}\|x^\star_{\mu_k}\|_{\mathbf{H}_{\mu_k}^{-1}(x^\star_{\mu_k})}}.$$

Note that as long as $k \geq K$,

$$q_{k+1} \leq \sup_{\mu \geq \lambda} \frac{\frac{7}{3} + 10R\sqrt{\mu}\|x^\star_\mu\|_{\mathbf{H}_\mu^{-1}(x^\star_\mu)}}{3 + 10R\sqrt{\mu}\|x^\star_\mu\|_{\mathbf{H}_\mu^{-1}(x^\star_\mu)}} \leq \frac{\frac{7}{3} + 10R\|x^\star_\lambda\|}{3 + 10R\|x^\star_\lambda\|}.$$

This guarantees convergence.

For the last point, the proof is exactly the same as in the previous proposition.

$\square$

**General non-adaptive result.** As mentioned in the core of the article, in practice, we do not select $q_{k+1}$ at each iteration using a safe adaptative value, but rather decrease $\mu_{k+1} = q\mu_k$ with a constant $q$, which we see as a parameter to tune. The following result shows that for $q$ large enough, this is justified, and that the lower bound we get for $q$ depends on the radius of the Dikin ellipsoid $\mathsf{r}_\mu(x)$, instead of $\frac{\sqrt{\mu}}{R}$ in the previous bounds, which is somewhat finer and shows that if the data is structured such that this radius is very big, then $q$ might actually be very small.

**Proposition 8** (Fixed $q$). *Assume that we perform phase I starting at $x_0$ such that*

$$x_0 \in \mathsf{D}_{\mu_0}\left(\frac{1}{7}\right).$$

*Assume we perform the method with a fixed $q_{k+1} = q$, satisfying*

$$q \geq \sup_{\lambda \leq \mu \leq \mu_0} \frac{\frac{4}{3} + 8\frac{\mu\|x^\star_\mu\|_{\mathbf{H}_\mu^{-1}(x^\star_\mu)}}{\mathsf{r}_\mu(x^\star_\mu)}}{2 + 8\frac{\mu\|x^\star_\mu\|_{\mathbf{H}_\mu^{-1}(x^\star_\mu)}}{\mathsf{r}_\mu(x^\star_\mu)}}.$$

*Then the following hold:*

1. $\forall k \leq K+1$, $x_k \in \mathsf{D}_{\mu_k}\left(\frac{1}{7}\right)$.

2. $K$ *is finite,*

$$K \leq \frac{1}{1-q}\log\frac{\mu_0}{\lambda},$$

*and $x_{K+1} \in \mathsf{D}_\lambda\left(\frac{1}{7}\right)$.*

*Proof.* Let us prove the two points.

**1.** Let us prove the result by induction. The initialization is trivial. Now assume $x_k \in \mathsf{D}_{\mu_k}(\frac{1}{7})$. Performing two iterations of the approximate Newton method guarantees that

$$x_{k+1} \in \mathsf{D}_{\mu_k}\left(\frac{1}{21}\right),$$

as show in Proposition 5. Now using Lemma 13, we see that $x_{k+1} \in \mathsf{D}_{q\mu_k}(\frac{1}{7})$, provided that

$$q \geq \frac{\frac{1}{3} + \dfrac{7\mu_k \|x_{k+1}\|_{\mathbf{H}_{\mu_k}^{-1}(x_{k+1})}}{\mathsf{r}_{\mu_k}(x_{k+1})}}{1 + \dfrac{7\mu_k \|x_{k+1}\|_{\mathbf{H}_{\mu_k}^{-1}(x_{k+1})}}{\mathsf{r}_{\mu_k}(x_{k+1})}}.$$

Now using Cor. 3, we get that

$$\frac{7\mu_k \|x_{k+1}\|_{\mathbf{H}_{\mu_k}^{-1}(x_{k+1})}}{\mathsf{r}_{\mu_k}(x_{k+1})} \leq 1 + \frac{8\mu_k \|x_{\mu_k}^\star\|_{\mathbf{H}_{\mu_k}^{-1}(x_{\mu_k}^\star)}}{\mathsf{r}_{\mu_k}(x_{\mu_k}^\star)} \leq 1 + 8 \sup_{\lambda \leq \mu \leq \mu_0} \frac{\mu \|x_\mu^\star\|_{\mathbf{H}_\mu^{-1}(x_\mu^\star)}}{\mathsf{r}_\mu(x_\mu^\star)}.$$

Hence the result.

**2.** This point just follows, using the bound $\frac{1}{\log \frac{1}{q}} \leq \frac{1}{1-q}$.

$\square$

### C.3 Proof of Thm. 1

Using Remark 2, the fact that $x_0 = 0$ and $\mu_0 = 7R\|\nabla f(0)\|$, as well as the hypotheses of the theorem, we can apply Proposition 6, and show that the number of steps $K$ performed in the first phase is bounded:

$$K \leq \lfloor (3 + 11R\|x_\lambda^\star\|) \log(7R\|\nabla f(0)\|/\lambda) \rfloor .$$

Moreover, this proposition also shows that $R\nu_\lambda(x_{last})/\sqrt{\lambda} \leq \frac{1}{7}$. Hence, we can use Proposition 5: if

$$t \geq T = \left\lceil \log_2 \sqrt{\frac{\lambda \varepsilon^{-1}}{R^2}} \right\rceil \geq \left\lceil \log_2 \frac{\nu_\lambda(x_{last})}{\sqrt{\varepsilon}} \right\rceil,$$

then it holds $\nu_\lambda(\hat{x}) \leq \sqrt{\varepsilon}$ and $f_\lambda(\hat{x}) - f_\lambda(x_\lambda^\star) \leq \varepsilon$.

$\square$

# D Non-parametric learning with generalized self-concordant functions

In this section, the aim is to provide a fast algorithm in the case of Kernel methods which achieves the optimal statistical guarantees.

## D.1 General setting and assumptions, statistical result for regularized ERM.

In this section, we consider the supervised learning problem of learning a predictor $f : \mathcal{X} \to \mathcal{Y}$ from training samples $(x_i, y_i)_{1 \leq i \leq n}$ which we assume to be realisations from a certain random variable $Z = (X, Y) \in \mathcal{Z} = \mathcal{X} \times \mathcal{Y}$ whose distribution is $\rho$. In what follows, for simplification purposes, we assume $\mathcal{Y} = \mathbb{R}$; however, this analysis can easily be adapted (although with heavier notations) to the setting where $\mathcal{Y} = \mathbb{R}^p$. Our aim is to compute the predictor of minimal generalization error

$$\inf_{f \in \mathcal{H}} L(f) := \mathbb{E}_{z \sim \rho} \left[ \ell_z(f(x)) \right], \tag{29}$$

where $\mathcal{H}$ is a space of candidate solutions and $\ell_z : \mathbb{R} \to \mathbb{R}$ is a loss function comparing the prediction $f(x)$ to the objective $y$.

**Kernel methods.** Kernel methods consider a space of functions $\mathcal{H}_K$ implicitly constructed from a symmetric positive semi-definite Kernel $K : \mathcal{X} \times \mathcal{X} \to$ and whose basic functions are the $K_x : t \in \mathcal{X} \mapsto K(x, t)$ and the linear combinations of such functions $f = \sum_{j=1}^{m} \alpha_j K_{x_j}$.

It is endowed with a scalar product such that: $\forall x_1, x_2 \in \mathcal{X}$, $K_{x_1} \cdot K_{x_2} = K(x_1, x_2)$, and as a consequence, $\mathcal{H}_K$ satisfies the self-reprocucing property:

$$\forall x \in \mathcal{X}, \ \forall f \in \mathcal{H}, \ f(x) = \langle f, K_x \rangle_{\mathcal{H}}.$$

In order to find a good predictor for Eq. (29), the following estimator, called the regularized ERM estimator, is often computed:

$$\widehat{f}_\lambda := \arg\min_{f \in \mathcal{H}} \widehat{L}_\lambda(f) := \frac{1}{n} \sum_{i=1}^{n} \ell_{z_i}(f(x_i)) + \frac{\lambda}{2} \|f\|_{\mathcal{H}}^2.$$

The properties of this estimator have been studied in [13] for the square loss and [23] for generalized self-concordant functions. In Appendix H, we recall the full setting of [23], and extend it to include the statistical properties of the projected problem.

**Assumptions** In this section, we will make the following assumptions, which are reformulations of the assumptions of [23], which we recall in Appendix H, in order to have the statistical properties of the regularized ERM. First, we assume that the $(x_i, y_i)$ are i.i.d. samples.

**Assumption 1** (i.i.d. data). *The samples $(z_i)_{1 \leq i \leq n} = (x_i, y_i)_{1 \leq i \leq n} \in \mathcal{Z}^n$ are independently and identically distributed according to $\rho$.*

In the case where $\mathcal{Y} = \mathbb{R}$, we make the following assumptions on the loss, which leads to the self concordance of the mappings $f \mapsto \ell_z(f(x))$ and that of $L, \widehat{L}$...

**Assumption 2** (Technical assumptions). *The mapping $(z, t) \in \mathcal{Z} \times \mathbb{R} \mapsto \ell_z(t)$ is measurable. Moreover,*

- *there exists $\mathsf{R}_\ell < \infty$ such that for all $z \in \mathrm{supp}(Z)$,*

$$\forall t \in \mathbb{R}, \ |\ell_z^{(3)}(t)| \leq \mathsf{R}_\ell \ell_z''(t),$$

- *the random variables $|\ell_Z(0)|, |\ell_Z'(0)|, |\ell_Z''(0)|$ are are bounded;*

- *The kernel is bounded, i.e. $\forall x \in \mathrm{supp}(X), \ K(x, x) \leq \kappa^2$ for a certain $\kappa$.*

Using these assumptions, we see that the following properties are satisfied. Define $L_z(f) := \ell_z(f(x))$. Then the $L_z$ satisfy the following properties:

- For any $z \in \mathcal{Z}$, $(L_z, \{\mathsf{R}_\ell K_x\})$ is a generalized self-concordant function in the sense of Definition 4.
- The mapping $(z, f) \in \mathcal{Z} \times \mathcal{H} \mapsto L_z(f)$ is measurable;
- the random variables $\|L_Z(0)\|, \|\nabla L_Z(0)\|, \mathrm{Tr}(\nabla^2 L_Z(0))$ are bounded by $|\ell_Z(0)|, \kappa|\ell'_Z(0)|$, $\kappa^2|\ell''_Z(0)|$;
- $\mathcal{G} := \{\mathsf{R}_\ell K_x \ : \ z \in \mathrm{supp}(Z)\}$ is a bounded subset of $\mathcal{H}$, bounded by $R = \mathsf{R}_\ell \kappa$.

This shows that Assumption 7 and Assumption 8 are satisfied by the $L_z$ and hence, using Proposition 16 in the next appendix, $L$ is well-defined, generalized self-concordant with $\mathcal{G}$. Moreover, the empirical loss

$$\widehat{L} = \frac{1}{n} \sum_{i=1}^{n} L_{z_i},$$

is also generalized self-concordant with $\widehat{\mathcal{G}} := \{\mathsf{R}_\ell K_{x_i} \ : \ 1 \leq i \leq n\}$.

Finally, as in Appendix H, we make an assumption on the regularity of the problem; namely, we assume that a solution to the learning problem exists in $\mathcal{H}$.

**Assumption 3** (Existence of a minimizer). *There exists $f^\star \in \mathcal{H}$ such that $L(f^\star) = \inf_{f \in \mathcal{H}} L(f)$.*

We adopt all the notations from Appendix H, doing the distinction between expected an empirical problems by adding a $\widehat{\ }$ over the quantities related to the empirical problem. We continue using the standard notations for $L$: for any $f \in \mathcal{H}$ and $\lambda > 0$,

$$L_\lambda(f) = L(f) + \frac{\lambda}{2}\|f\|^2, \qquad \widehat{L}_\lambda(f) = \widehat{L}(f) + \frac{\lambda}{2}\|f\|^2$$

$$\mathbf{H}(f) = \nabla^2 L(f), \qquad \mathbf{H}_\lambda(f) = \nabla^2 L_\lambda(f) = \mathbf{H}(f) + \lambda\mathbf{I}$$

$$\widehat{\mathbf{H}}(f) = \nabla^2 \widehat{L}(f), \qquad \widehat{\mathbf{H}}_\lambda(f) = \nabla^2 \widehat{L}_\lambda(f) = \widehat{\mathbf{H}}(f) + \lambda\mathbf{I}$$

Recall that $\widehat{f}_\lambda$ is defined as the minimizer of $\widehat{L}_\lambda$.

Define the following bounds on the second order derivatives:

$$\forall f \in \mathcal{H}, \ \mathsf{b}_2(f) = \sup_{z \in \mathrm{supp}(Z)} \ell''_z(f(x)).$$

**Statistical properties of the estimator**  The statistical properties of the estimator $\widehat{f}_\lambda$ have been studied in [23] in the case of generalized self concordance, an are reported in the main lines in Appendix H. The statistical rates of this estimator and the optimal choice of $\lambda$ is determined by two parameters, defined in Proposition 17 and which we adapt to the Kernel problem here.

- the *bias* $\mathsf{b}_\lambda = \|\mathbf{H}_\lambda(f^\star)^{-1/2}\nabla L_\lambda(f^\star)\| = \lambda\|f^\star\|_{\mathbf{H}_\lambda^{-1}(f^\star)}$, which characterizes the regularity of the optimum. The faster $\mathsf{b}_\lambda$ decreases to zero, the more regular $f^\star$ is.
- the *effective dimension*

$$\mathsf{df}_\lambda = \mathbb{E}\left[\|\mathbf{H}_\lambda(f^\star)^{-1/2}\nabla L_Z(f^\star)\|^2\right]. \tag{30}$$

This quantity characterizes the size of the space $\mathcal{H}$ with respect to the problem; the slower it explodes as $\lambda$ goes to zero, the smaller the size of $\mathcal{H}$.

For more complete explanations on the meaning of these quantities, we refer to [23].

Moreover, as mentioned in Proposition 17, one can define

$$\mathsf{B}_1^\star := \sup_{z \in \mathrm{supp}(Z)} \|\nabla L_z(f^\star)\|, \ \mathsf{B}_2^\star := \sup_{z \in \mathrm{supp}(Z)} \mathrm{Tr}(\nabla^2 L_z(f^\star)), \ \mathsf{Q}^\star = \frac{\mathsf{B}_1^\star}{\sqrt{\mathsf{B}_2^\star}}, \ \mathsf{b}_2^\star = \mathsf{b}_2(f^\star). \tag{31}$$

We assume the following regularity condition on the minimizer $f^\star$, in order to get statistical bounds.

**Assumption 4** (Source condition). *There exists $r > 0$ and $g \in \mathcal{H}$ such that $f^\star = \mathbf{H}^r(f^\star)g$. This implies the following decrease rate of the bias:*

$$\mathsf{b}_\lambda \leq \mathsf{L}\lambda^{1/2+r}, \qquad \mathsf{L} = \|g\|_{\mathcal{H}}.$$

This is a stronger assumption than the existence of the minimizer as $r > 0$ is crucial for our analysis.

We also quantify the effective dimension $\mathsf{df}_\lambda$: (however, since it always holds for $\alpha = 1$, this is not, strictly speaking, an additional assumption).

**Assumption 5** (Effective dimension). *The effective dimension decreases as $\mathsf{df}_\lambda \leq \mathsf{Q}\lambda^{-1/\alpha}$.*

If these two assumptions hold, define:

$$\beta = \frac{\alpha}{1 + \alpha(1 + 2r)}, \qquad \gamma = \frac{(1 + 2r)\alpha}{1 + \alpha(1 + 2r)}.$$

Under these assumptions, one can obtain the following statistical rates (which can be found in [23] or in Cor. 4).

**Proposition 9.** *Let $\delta \in (0, 1/2]$. Under Assumptions 1 to 5, when $n \geq N$ and $\lambda = (C_0/n)^\beta$, then with probability at least $1 - 2\delta$,*

$$L(\widehat{f}_\lambda) - L(f^\star) \leq C_1 n^{-\gamma} \log \frac{2}{\delta},$$

*with $C_0 = 256(\mathsf{Q}/\mathsf{L})^2$, $C_1 = 8(256)^\gamma (\mathsf{Q}^\gamma \mathsf{L}^{1-\gamma})^2$ and $N$ defined in [23], and satisfying $N = O(\mathrm{poly}(\mathsf{B}_1^\star, \mathsf{B}_2^\star, \mathsf{L}, \mathsf{Q}, R, \log(1/\delta)))$.*

### D.2 Reducing the dimension: projecting on a subspace using Nyström sub-sampling.

**Computations** Using a representer theorem, one of the key properties of Kernel spaces is that, owing to the reproducing property,

$$\widehat{f}_\lambda \in \mathcal{H}_n := \left\{ \sum_{i=1}^n \alpha_i K_{x_i} : (\alpha_i) \in \mathbb{R}^n \right\}.$$

This means that solving the regularized empirical problem can be turned into a finite dimensional problem in $\alpha$. Indeed $\widehat{f}_\lambda = \sum_{i=1}^n \alpha_i K_{x_i}$ where $\alpha = (\alpha_i)_{1 \leq i \leq n}$ is the solution to the following problem:

$$\alpha = \arg\min_{\alpha \in \mathbb{R}^n} \frac{1}{n} \sum_{i=1}^n \ell_{z_i}(\alpha^\top \mathbf{K}_{nn} e_i) + \frac{\lambda}{2}\alpha^\top \mathbf{K}_{nn}\alpha, \qquad \mathbf{K}_{nn} = (K(x_i, x_j))_{1 \leq i,j \leq n} \in \mathbb{R}^{n \times n}.$$

The previous problem is usually too costly to solve directly for large values of $n$, both in time and memory, because of the operations involving $\mathbf{K}_{nn}$. A solution consists in looking for a solution in a smaller dimensional sub-space $\mathcal{H}_M$ constructed from sub-samples of the data $\{\tilde{x}_1, ..., \tilde{x}_M\} \subset \{x_1, ..., x_n\}$:

$$\mathcal{H}_M := \left\{ \sum_{j=1}^M \tilde{\alpha}_j K_{\tilde{x}_j} : \tilde{\alpha} \in \mathbb{R}^M \right\}.$$

In this case, the minimizer $\widehat{f}_{M,\lambda} = \arg\min_{f \in \mathcal{H}_M} \widehat{L}_\lambda(f)$ can be written $\widehat{f}_{M,\lambda} = \sum_{j=1}^M \tilde{\alpha}_j K_{\tilde{x}_j}$, where $\tilde{\alpha}$ is the solution to the following problem:

$$\tilde{\alpha} = \arg\min_{\alpha \in \mathbb{R}^M} \frac{1}{n} \sum_{i=1}^n \ell_{z_i}(\alpha^\top \mathbf{K}_{Mn} e_i) + \frac{\lambda}{2}\alpha^\top \mathbf{K}_{MM}\alpha,$$

where

$$\mathbf{K}_{nM} = (K(x_i, \tilde{x}_j))_{\substack{1 \leq i \leq n \\ 1 \leq j \leq M}}, \ \mathbf{K}_{Mn} = \mathbf{K}_{nM}^\top, \ \mathbf{K}_{MM} := (K(\tilde{x}_i, \tilde{x}_j))_{1 \leq i,j \leq M}.$$

Let $\mathbf{T}$ be an upper triangular matrix such that $\mathbf{T}^\top \mathbf{T} = \mathbf{K}_{MM}$. One can re-parametrize the previous problem in the following way. For any $\beta \in \mathbb{R}^M$, define $f_\beta = \sum_{j=1}^M [\mathbf{T}^\dagger \beta]_j \, K_{\tilde{x}_j}$. This implies in particular that $\|f_\beta\|_\mathcal{H} = \|\beta\|_{\mathbb{R}^M}$. Then $\widehat{f}_{M,\lambda} = f_{\beta_{M,\lambda}}$, where

$$\beta_{M,\lambda} = \underset{\beta \in \mathbb{R}^M}{\arg\min} \, \widehat{L}_{M,\lambda}(\beta) := \frac{1}{n} \sum_{i=1}^n \ell_{z_i}(e_i^\top \mathbf{K}_{nM} \mathbf{T}^\dagger \beta) + \frac{\lambda}{2} \|\beta\|^2.$$

Using the properties the $\ell_z$, one easily shows that $\beta \mapsto \ell_{z_i}(e_i^\top \mathbf{K}_{nM} \mathbf{T}^\dagger \beta)$ is $\{\mathsf{R}_\ell \mathbf{T}^{-\top} \mathbf{K}_{Mn} e_i\}$ generalized self-concordant, and $\|\mathsf{R}_\ell \mathbf{T}^{-\top} \mathbf{K}_{Mn} e_i\| \leq \mathsf{R}_\ell \sqrt{K(x_i, x_i)}$. Thus, $\widehat{L}_M$ is also generalized self-concordant, and the associated $\widehat{\mathcal{G}}_M$ is bounded by $R = \mathsf{R}_\ell \kappa$. It will therefore be possible to apply the second order scheme presented in this paper to approximately compute $\beta_{M,\lambda}$.

**Statistics** Let $\widehat{\nu}_{\lambda,M}(\beta)$ denote the Newton decrement of $\widehat{L}_{\lambda,M}$ at point $\beta$ and $\mathbf{P}_M$ denote the orthogonal projection on $\mathcal{H}_M$. Then the following statistical result shows that provided $\beta$ is a good enough approximation of the optimum, and provided $\mathcal{H}_M$ is large enough, then $f_\beta$ has the same generalization error as the empirical risk minimizer $\widehat{f}_\lambda$.

Recall the following result proved in Proposition 19 in Appendix H.3.

**Proposition 10** (Behavior of an approximation to the projected problem). *Suppose that Assumptions 1 to 3 are satisfied. Let $n \in \mathbb{N}$, $\delta \in (0, 1/2]$, $0 < \lambda \leq \mathsf{B}_2^\star$. Whenever*

$$n \geq \triangle_1 \frac{\mathsf{B}_2^\star}{\lambda} \log \frac{8 \square_1^2 \mathsf{B}_2^\star}{\lambda \delta}, \qquad \mathsf{C}_1 \sqrt{\frac{\mathsf{df}_\lambda \vee (\mathsf{Q}^\star)^2}{n} \, \log \frac{2}{\delta}} \leq \frac{\lambda^{1/2}}{R}, \qquad \mathsf{C}_1 \mathsf{b}_\lambda \leq \frac{\lambda^{1/2}}{R},$$

*if*

$$\|\mathbf{H}^{1/2}(f^\star)(\mathbf{I} - \mathbf{P}_M)\|^2 \leq \lambda \frac{\sqrt{2}}{480}, \quad 126 \widehat{\nu}_{M,\lambda}(\beta) \leq \frac{\lambda^{1/2}}{R},$$

*the following holds, with probability at least $1 - 2\delta$.*

$$L(f_\beta) - L(f^\star) \leq \mathsf{K}_1 \, \mathsf{b}_\lambda^2 + \mathsf{K}_2 \frac{\mathsf{df}_\lambda \vee (\mathsf{Q}^\star)^2}{n} \, \log \frac{2}{\delta} + \mathsf{K}_3 \, \widehat{\nu}_{M,\lambda}^2(\beta), \qquad R\|f_\beta - f^\star\|_\mathcal{H} \leq 10,$$

*where $\mathsf{K}_1 \leq 6.0\mathrm{e}4$, $\mathsf{K}_2 \leq 6.0\mathrm{e}6$ and $\mathsf{K}_3 \leq 810$, $\mathsf{C}_1$ is defined in Lemma 19, and the other constants are defined in Thm. 8.*

In particular, if we apply the previous result for a fixed $\lambda$, the following theorem holds (for a proof, see Appendix H.4).

**Theorem 5** (Quantitative result with source $r > 0$). *Suppose that Assumptions 1 to 5 are satisfied. Let $n \geq N$ and $\delta \in (0, \frac{1}{2}]$. If $\lambda = \left( \left( \frac{\mathsf{Q}}{\mathsf{L}} \right)^2 \frac{1}{n} \right)^{\frac{\alpha}{\alpha(1+2r)+1}}$, and if*

$$\|\mathbf{H}^{1/2}(f^\star)(\mathbf{I} - \mathbf{P}_M)\|^2 \leq \lambda \frac{\sqrt{2}}{480}, \quad \widehat{\nu}_{M,\lambda}(\beta) \leq \mathsf{Q}^\gamma \, \mathsf{L}^{1-\gamma} n^{-\gamma/2},$$

*then with probability at least $1 - 2\delta$,*

$$L(f_\beta) - L(f^\star) \leq \mathsf{K} \left( \mathsf{Q}^\gamma \, \mathsf{L}^{1-\gamma} \right)^2 \frac{1}{n^\gamma} \log \frac{2}{\delta}, \qquad R\|f_\beta - f^\star\| \leq 10,$$

*where $N$ is defined in Eq. (42) and $\mathsf{K} \leq 7.0\mathrm{e}6$.*

The proof of the previous result is quite technical and can be found in Appendix H, in Thm. 9.

### D.3 A note on sub-sampling techniques

Let $Z$ be a random variable on a Polish space $\mathcal{Z}$ and $(v_z)_{z \in \mathcal{Z}}$ be a family of vectors in $\mathcal{H}$ such that $\|v\|_{L^\infty(Z)} := \sup_{z \in \mathrm{supp}(Z)} \|v_z\| < \infty$ is bounded. Assume that $z_1, ..., z_n$ are i.i.d. samples from $Z$.

Define the following trace class Hermitian operators:

$$\mathbf{A} = \mathbb{E}\left[v_Z \otimes v_Z\right], \ \widehat{\mathbf{A}} = \frac{1}{n}\sum_{i=1}^{n} v_{z_i} \otimes v_{z_i}.$$

Define

$$\mathcal{N}^{\mathbf{A}}(\lambda) := \mathrm{Tr}(\mathbf{A}_\lambda^{-1}\mathbf{A}), \qquad \mathcal{N}_\infty^{\mathbf{A}}(\lambda) := \sup_{z\in\mathrm{supp}(Z)} \|\mathbf{A}_\lambda^{-1/2}v_z\|^2. \qquad (32)$$

We typically have:

$$\mathcal{N}^{\mathbf{A}}(\lambda) \leq \mathcal{N}_\infty^{\mathbf{A}}(\lambda) \leq \frac{\|v\|_{L^\infty(Z)}^2}{\lambda}.$$

We define the leverage scores associated to the points $z_i$ and $\mathbf{A}$:

$$\forall 1 \leq i \leq n, \ \forall t > 0, \ l_i^{\mathbf{A}}(t) = \|\widehat{\mathbf{A}}_t^{-1/2}v_{z_i}\|^2 = n\left((\mathbf{G}_{nn} + tn\mathbf{I})^{-1}\mathbf{G}_{nn}\right)_{ii}, \qquad (33)$$

where $\mathbf{G}_{nn} = (v_{z_i}\cdot v_{z_j})_{1\leq i,j\leq n}$ denotes the Gram matrix associated to the family $v_{z_i}$.

As in [28], definition 1, we give the following definition for leverage scores.

**Definition 5** ($q$-approximate leverage scores). *given $t_0$, a family $(\tilde{l}_i^{\mathbf{A}}(t))_{1\leq i\leq n}$ is said to be a family of $q$-approximate leverage scores with respect to $\mathbf{A}$ if*

$$\forall 1 \leq i \leq n, \ \forall t \geq t_0, \ \frac{1}{q} l_i^{\mathbf{A}}(t) \leq \tilde{l}_i^{\mathbf{A}}(t) \leq q\, l_i^{\mathbf{A}}(t).$$

We say that a subset of $m$ points $\{\tilde{z}_1, ..., \tilde{z}_m\} \subset \{z_i \ : \ 1 \leq i \leq n\}$ is:

- **Sampled using $q$-approximate leverage scores for** $t$ if the $\tilde{z}_j = z_{i_j}$ where the $i_j$ are $m$ i.i.d. samples from $\{1, ..., n\}$ using the probability vector $p_i = \frac{\tilde{l}_i^{\mathbf{A}}(t)}{\sum_{i=1}^n \tilde{l}_i^{\mathbf{A}}(t)}$. In that case, we define $\widehat{\mathbf{A}}_m := \frac{1}{m}\sum_{j=1}^m \frac{1}{np_{i_j}} v_{\tilde{z}_j} \otimes v_{\tilde{z}_j}$.

- **Sampled uniformly** if the $\{i_j \ : \ 1 \leq j \leq m\}$ is a uniformly chosen subset of $\{1, ..., n\}$ of size $m$. In this case, we define $\widehat{\mathbf{A}}_m := \frac{1}{m}\sum_{j=1}^m v_{\tilde{z}_j} \otimes v_{\tilde{z}_j}$.

In Appendix I.1, we present technical lemmas which allow us to show that if $m$ is large enough, the following hold:

- $\|\mathbf{A}_\eta(\mathbf{I} - \mathbf{P}_m)\|^2 \leq 3\eta$, where $\mathbf{P}_m$ is the orthogonal projection on the subspace induced by the $v_{\tilde{z}_j}$;

- $\widehat{\mathbf{A}}_{m,\lambda}$ is equivalent to $\widehat{\mathbf{A}}_\lambda$.

**Remark 3** (cost of computing $q$-approximate leverage scores). *In [30], one can show that the complexity of computing $q$-approximate leverage scores can be achieved in:* $c_{samp} = O(q^2 \mathcal{N}^{\mathbf{A}}(\lambda)^2 \min(n, 1/\lambda))$ *time (where a unit of time is a scalar product evaluation) and* $O(\mathcal{N}^{\mathbf{A}}(\lambda)^2 + n)$ *in memory.*

### D.4 Selecting the $M$ Nyström points

In order for Thm. 5 to hold, we must subsample the $M$ points such as to guarantee $\|\mathbf{H}^{1/2}(f^\star)(\mathbf{I} - \mathbf{P}_M)\|^2 \leq \frac{\sqrt{2\lambda}}{480}$.

Since we must sub-sample the $M$ points a priori, i.e. before performing the method, it is necessary to have sub-sampling schemes which do not depend heavily on the point. Define the covariance operator:

$$\mathbf{\Sigma} = \mathbb{E}\left[K_X \otimes K_X\right].$$

Since $\mathbf{H}(f^\star) = \mathbb{E}\left[\ell_Z''(f(X)) K_X \otimes K_X\right]$, it is easy to see that $\mathbf{H}(f^\star) \preceq b_2^\star \mathbf{\Sigma}$. Note that for $\mathbf{\Sigma}$, since $\widehat{\mathbf{\Sigma}} = \frac{1}{n}\sum_{i=1}^n K_{x_i} \otimes K_{x_i}$, the leverage scores have the following form:

$$\forall 1 \leq i \leq n, \ l_i^{\mathbf{\Sigma}}(t) = n\left((\mathbf{K}_{nn} + \lambda n\mathbf{I})^{-1}\mathbf{K}_{nn}\right)_{ii}.$$

**Proposition 11** (Selecting Nyström points). *Let $\delta > 0$. Let $\eta = \min(\|\boldsymbol{\Sigma}\|, \frac{\lambda\sqrt{2}}{1440(\mathsf{b}_2^\star \vee 1)})$. Assume the samples $\{\tilde{x}_1, ..., \tilde{x}_M\}$ are obtained with one of the following.*
**1.** $n \geq M \geq \left(10 + 160\mathcal{N}_\infty^{\boldsymbol{\Sigma}}(\eta)\right)\log\frac{8\kappa^2}{\eta\delta}$ *using uniform sampling;*
**2.** $M \geq \left(6 + 486q^2\mathcal{N}^{\boldsymbol{\Sigma}}(\eta)\right)\log\frac{8\kappa^2}{\eta\delta}$ *using $q$-approximate leverage scores with respect to $\boldsymbol{\Sigma}$ for $t = \eta$, $t_0 \vee \frac{19\kappa^2}{n}\log\frac{n}{2\delta} < \eta$, $n \geq 405\kappa^2 \vee 67\kappa^2\log\frac{12\kappa^2}{\delta}$.*
*Then it holds, with probability at least $1 - \delta$:*

$$\|\boldsymbol{\Sigma}_\eta^{1/2}(\mathbf{I} - \mathbf{P}_M)\| \leq 3\eta \implies \|\mathbf{H}^{1/2}(f^\star)(\mathbf{I} - \mathbf{P}_M)\|^2 \leq \lambda\frac{\sqrt{2}}{480}.$$

*Proof.* The proof is a direct application of the lemmas in Appendix I.1. Indeed, note that since $\boldsymbol{\Sigma} = \mathbb{E}[K_X \otimes K_X]$, then the results can be applied with $Z \leftarrow X$ and $v_z \leftarrow K_x$. Indeed, from Assumption 2, it holds:
$$\sup_{x \in \text{supp}(X)} \|K_x\|^2 \leq \kappa^2.$$

$\square$

We can now combine Proposition 11 and Proposition 10 to obtain the following statistical bounds for the optimizer of the projected Nyström problem $\beta_{M,\lambda}$.

**Theorem 6.** *Suppose that Assumptions 1 to 3 are satisfied. Let $n \in \mathbb{N}$, $\delta \in (0, 1/2]$, $0 < \lambda \leq \mathsf{B}_2^\star \wedge 720\sqrt{2}(\mathsf{b}_2^\star \vee 1)\|\boldsymbol{\Sigma}\|$. Assume*

$$n \geq \triangle_1\frac{\mathsf{B}_2^\star}{\lambda}\log\frac{8\square_1^2\mathsf{B}_2^\star}{\lambda\delta}, \qquad \mathsf{C}_1\sqrt{\frac{\mathsf{df}_\lambda \vee (\mathsf{Q}^\star)^2}{n}}\ \log\frac{2}{\delta} \leq \frac{\lambda^{1/2}}{R}, \qquad \mathsf{C}_1\mathsf{b}_\lambda \leq \frac{\lambda^{1/2}}{R},$$

*Let $\eta = \frac{\lambda\sqrt{2}}{1440(\mathsf{b}_2^\star \vee 1)}$. Assume the samples $\{\tilde{x}_1, ..., \tilde{x}_M\}$ are obtained with one of the following.*
**1.** $n \geq M \geq \left(10 + 160\mathcal{N}_\infty^{\boldsymbol{\Sigma}}(\eta)\right)\log\frac{8\kappa^2}{\eta\delta}$ *using uniform sampling;*
**2.** $M \geq \left(6 + 486q^2\mathcal{N}^{\boldsymbol{\Sigma}}(\eta)\right)\log\frac{8\kappa^2}{\eta\delta}$ *using $q$-approximate leverage scores with respect to $\boldsymbol{\Sigma}$ for $t = \eta$, $t_0 \vee \frac{19\kappa^2}{n}\log\frac{n}{2\delta} < \eta$, $n \geq 405\kappa^2 \vee 67\kappa^2\log\frac{12\kappa^2}{\delta}$.*
*The following holds, with probability at least $1 - 3\delta$:*

$$L(f_{\beta_{M,\lambda}}) - L(f^\star) \leq \mathsf{K}_1\ \mathsf{b}_\lambda^2 + \mathsf{K}_2\frac{\mathsf{df}_\lambda \vee (\mathsf{Q}^\star)^2}{n}\ \log\frac{2}{\delta}, \qquad R\|\beta_{M,\lambda}\| \leq R\|f^\star\| + 10,$$

*where $\mathsf{K}_1 \leq 6.0\text{e}4$, $\mathsf{K}_2 \leq 6.0\text{e}6$ and $\mathsf{K}_3 \leq 810$, $\mathsf{C}_1$ is defined in Lemma 19, and the other constants are defined in Thm. 8.*

*Proof.* This is simply a reformulation of Proposition 10, noting that $\widehat{\nu}_{M,\lambda}(\beta_{M,\lambda}) = 0$ and that Proposition 11 implies the condition on the Hessian at the optimum. $\square$

Provided source condition holds with $r > 0$, the conditions of this theorem are not void.

### D.5  Performing the globalization scheme to approximate $\beta_{M,\lambda}$

In order to apply Proposition 10, one needs to control $\widehat{\nu}_{M,\lambda}(\beta)$.

We will apply our general scheme to $\widehat{L}_{M,\lambda}$ in order to obtain such a control.

#### D.5.1  Performing approximate Newton steps

The key element in the globalization scheme is to be able to compute $\frac{1}{7}$-approximate Newton steps.

Note that at a given point $\beta$ and for a given $\mu > 0$ the Hessian is of the form:

$$\widehat{\mathbf{H}}_{M,\mu}(\beta) = \frac{1}{n}\mathbf{T}^{-\top}\mathbf{K}_{Mn}\mathbf{D}_n(\beta)\mathbf{K}_{nM}\mathbf{T}^{-1} + \mu\mathbf{I}_M,$$

where $\mathbf{D}_n(\beta) = \mathrm{diag}((d_i(\beta))_{1 \leq i \leq n})$ is a diagonal matrix whose elements are given by $d_i(\beta) = \ell''_{z_i}(e_i^\top \mathbf{K}_{nM} \mathbf{T}^{-1}\beta)$.

Note that we can always write

$$\widehat{\mathbf{H}}_{M,\mu}(\beta) = \frac{1}{n} \sum_{i=1}^{n} u_i(\beta) u_i(\beta)^\top + \mu \mathbf{I}, \qquad u_i(\beta) = \sqrt{d_i(\beta)} \mathbf{T}^{-\top} \mathbf{K}_{Mn} e_i$$

The gradient can be put in the following form:

$$\nabla \widehat{L}_{M,\mu}(\beta) = \frac{1}{n} \mathbf{T}^{-\top} \mathbf{K}_{Mn} v + \mu \beta, \qquad v = (\ell'_{z_i}(e_i^\top \mathbf{K}_{nM} \mathbf{T}^{-1}\beta))_{1 \leq i \leq n}.$$

Computing the gradient at one point therefore costs $O(nM + M^2)$, this being the cost of computing $\mathbf{K}_{nM}$ times a vector costs $O(nM)$ and computing $\mathbf{T}^{-1}$ times a vector takes $O(M^2)$ since $\mathbf{T}$ is triangular. Moreover, the cost in memory is $O(M^2 + n)$, $M^2$ being needed for the saving of $\mathbf{T}$ and $n$ for the saving of the gradient; $\mathbf{K}_{nM}$ times a vector can also be done in $O(n)$ memory, provided we compute it by blocks.

On the other hand, computing the full Hessian matrix would cost $nM^2$ operations, which is untractable. However, computing a Hessian vector product can be done in $O(nM + M^2)$ time, as for the gradient, which suggest using an iterative solver with preconditioning.

**Computing $x \in \mathrm{LinApprox}(\mathbf{A}, b, \rho)$ through pre-conditioned conjugate gradient descent.** Assume we wish to solve the problem $\mathbf{A}x = b$ where $\mathbf{A} \in \mathbb{R}^{M \times M}$ is a positive definite matrix and $b$ is a vector of $\mathbb{R}^M$. If one uses the conjugate gradient method starting from zero, then if $x_k$ denotes the $k$-the iterate of the conjugate gradient algorithm, Theorem 6.6 in [31] shows that

$$x_k \in \mathrm{LinApprox}(\mathbf{A}, b, \rho), \qquad \rho = 2 \left( \frac{\sqrt{\mathrm{Cond}(\mathbf{A})} - 1}{\sqrt{\mathrm{Cond}(\mathbf{A})} + 1} \right)^k.$$

where $\mathrm{Cond}(\mathbf{A})$ is the condition number of the matrix $\mathbf{A}$, namely the ratio $\frac{\lambda_{\max}(\mathbf{A})}{\lambda_{\min}(\mathbf{A})}$. If $\mathrm{Cond}(\mathbf{A})$ is large, this convergence can be very slow. The idea of preconditioning is to compute an approximation matrix $\widetilde{\mathbf{A}}$ such that

$$\frac{1}{2}\widetilde{\mathbf{A}} \preceq \mathbf{A} \preceq \frac{3}{2}\widetilde{\mathbf{A}}. \tag{34}$$

We then compute $\mathbf{B}$ a triangular matrix such that $\mathbf{B}^\top \mathbf{B} = \widetilde{\mathbf{A}}$ using a cholesky decomposition, which can be done in $O(M^3)$, and note that $\mathbf{B}^{-\top} \mathbf{A} \mathbf{B}^{-1}$ is very well conditioned; indeed, its condition number is bounded by 3.

Perform a conjugate gradient method to solve the pre-conditioned problem $\mathbf{B}^{-\top} \mathbf{A} \mathbf{B}^{-1} z = \mathbf{B}^{-\top} b$, and denote with $z_\tau$ the $\tau$-th iteration of this method. Then using the bound on the condition number, we find

$$z_\tau \in \mathrm{LinApprox}(\mathbf{B}^{-\top} \mathbf{A} \mathbf{B}^{-1}, \mathbf{B}^{-\top} b, \rho), \qquad \rho = 2 \left( \frac{\sqrt{3} - 1}{\sqrt{3} + 1} \right)^\tau,$$

which in turn implies that by setting $x_\tau := \mathbf{B}^{-1} z_\tau$,

$$x_\tau \in \mathrm{LinApprox}(\mathbf{A}, b, \rho), \ \rho = 2 \left( \frac{\sqrt{3} - 1}{\sqrt{3} + 1} \right)^\tau.$$

This shows that after at most $\tau = 3$ iterations, provided $\widetilde{\mathbf{A}}$ satisfies Eq. (34), $x_\tau \in \mathrm{LinApprox}(\mathbf{A}, b, \frac{1}{7})$. The cost of this method is therefore $O(M^3 + nM)$ in time, and $O(n + M^2)$ due to the computing of the preconditioner and computing matrix vector products by block. This does not include the cost of finding a suitable $\widetilde{\mathbf{A}}$.

**Computing a suitable approximation of** $\widehat{\mathbf{H}}_{M,\mu}(\beta)$    To compute a good pre-conditioner, we will subsample $Q$ points $i_1, ..., i_Q$ points from $\{1, ..., n\}$, and sketch the Hessian using these $Q$ points.

**Proposition 12** (Computing approximate newton steps). *Let $\delta > 0$. Let $\beta \in \mathbb{R}^M$ and $\mu \geq \lambda$, and assume $\frac{19\mathsf{b}_2(f_\beta)\kappa^2}{n} \log \frac{n}{2\delta} < \lambda$ and $n \geq 405\mathsf{b}_2(f_\beta)\kappa^2 \vee 67\mathsf{b}_2(f_\beta)\kappa^2 \log \frac{12\mathsf{b}_2(f_\beta)\kappa^2}{\delta}$. Let $\tilde{\mu} = \min(\mu, \|\mathbf{H}(f_\beta)\|)$. Assume one of the following properties is satisfied*

**1.** $Q \geq \left(10 + 160\mathcal{N}_\infty^{\mathbf{H}(f_\beta)}(\tilde{\mu})\right) \log \frac{8\mathsf{b}_2(f_\beta)\kappa^2}{\tilde{\mu}\delta}$ *with uniform sampling of the $\{i_1, ..., i_Q\}$. We set* $\mathbf{D}_Q = \operatorname{diag}(\ell''_{z_{i_j}}(f_\beta(x_{i_j})))_{1 \leq j \leq Q}$

**2.** $Q \geq \left(6 + 486q^2\mathcal{N}^{\mathbf{H}(f_\beta)}(\tilde{\mu})\right) \log \frac{8\mathsf{b}_2(f_\beta)\kappa^2}{\tilde{\mu}\delta}$ *using $q$-approximate leverage scores associated to* $\mathbf{H}(f_\beta)$ *for $t = \tilde{\mu}$. We set* $\mathbf{D}_Q = \operatorname{diag}\left(\frac{\ell''_{z_{i_j}}(f_\beta(x_{i_j}))}{p_{i_j}}\right)$, *where the $p_{i_j}$ are the probabilities computed from the leverage scores.*

*Assume we use a pre-conditioner $\mathbf{B}$ such that*

$$\mathbf{B}^\top \mathbf{B} = \frac{1}{Q}\mathbf{T}^{-\top}\mathbf{K}_{MQ}\mathbf{D}_Q\mathbf{K}_{QM}\mathbf{T}^{-1} + \mu\mathbf{I}_M, \qquad \mathbf{K}_{QM} = (K(x_{i_j}, \tilde{x}_k))_{\substack{1 \leq j \leq Q \\ 1 \leq k \leq M}}.$$

*If we perform $\tau = \log(\rho/2)/\log((\sqrt{3}+1)/\sqrt{3}-1)$ iterations of the conjugate gradient descent on the pre-conditioned Newton system using $\mathbf{B}$ as a preconditioner, then with probability at least $1 - \delta$, this procedure is returns $\tilde{\Delta} \in \operatorname{LinApprox}(\widehat{\mathbf{H}}_{M,\lambda}(\beta), \nabla\widehat{L}_{M,\lambda}(\beta), \rho)$, and the computational time is of order $O(\tau(Mn + M^2Q + M^3 + \mathsf{c}_{samp}))$, and the memory requirements can be reduced to $O(M^2 + n)$. Here $\mathsf{c}_{samp}$ stands for the complexity of computing Nystrom leverage scores, and using Remark 3 or [30], $\mathsf{c}_{samp} = O(1)$ if uniform sampling is used, and $\mathsf{c}_{samp} = O(\mathcal{N}^{\mathbf{H}(f_\beta)}(\tilde{\mu})^2/\lambda)$ if Nystrom sub-sampling is used. Note that for $\tau = 3$, $\rho = \frac{1}{7}$.*

*Proof.* Start by defining the following operators:

- $K_n : f \in \mathcal{H} \to (f(x_i))_{1 \leq i \leq n} \in \mathbb{R}^n$;

- $K_M : f \in \mathcal{H} \to (f(\tilde{x}_j))_{1 \leq j \leq M} \in \mathbb{R}^M$;

- $V = K_M^*\mathbf{T}^{-1}$, where $\mathbf{T}$ is an upper triangular matrix such that $\mathbf{T}^\top\mathbf{T} = \mathbf{K}_{MM} = K_M K_M^*$.

Note that $K_n V = \mathbf{K}_{nM}\mathbf{T}^{-1}$.

Now note that

$$\forall f \in \mathcal{H}, \ \mathbf{H}(f) = \mathbb{E}\left[v_z \otimes v_z\right], \qquad \widehat{\mathbf{H}}(f) = \frac{1}{n}\sum_{i=1}^n v_{z_i} \otimes v_{z_i}, \qquad v_z = \sqrt{\ell''_z(f(x))}K_x.$$

Since for any $f \in \mathcal{H}$, $\widehat{\mathbf{H}}(f) = \frac{1}{n}K_n^*\mathbf{D}_n(f)K_n$, where $\mathbf{D}_n(f) = \operatorname{diag}(\ell''_{z_i}(f(x_i)))$, we see that

$$\widehat{\mathbf{H}}_{M,\mu}(\beta) = V^*\widehat{\mathbf{H}}(f_\beta)V + \mu\mathbf{I}_M.$$

Thus, the last lemma of Appendix I.1 can be applied, using the fact that $\|v_z\|^2 \leq \mathsf{b}_2(f)\kappa^2$, to get that in both cases of the proposition, under the corresponding assumptions:

$$\frac{1}{2}\left(\frac{1}{Q}\mathbf{T}^{-\top}\mathbf{K}_{MQ}\mathbf{D}_Q\mathbf{K}_{QM}\mathbf{T}^{-1} + \mu\mathbf{I}_M\right) \preceq \widehat{\mathbf{H}}_{M,\mu}(\beta) \preceq \frac{3}{2}\left(\frac{1}{Q}\mathbf{T}^{-\top}\mathbf{K}_{MQ}\mathbf{D}_Q\mathbf{K}_{QM}\mathbf{T}^{-1} + \mu\mathbf{I}_M\right).$$

The rest of the proposition follows from the previous discussion.

$\square$

### D.5.2  Applying the globalization scheme to control $\widehat{\nu}_{M,\lambda}(\beta)$

In order to apply Proposition 12 to each point $\beta$ in our method, we need to have a globalized version of the condition of this proposition.

First, we start by localizing the different values of $\beta$ we will visit throughout the algorithm.

**Definition 6** (path of regularized solutions). *Let $\lambda > 0$, $\varepsilon > 0$. Define the path of regularized solutions*

$$\widehat{\Gamma}^M_\lambda := \{\beta_{M,\mu} \ : \ \mu \geq \lambda\}. \tag{35}$$

*And the $\varepsilon$ approximation of this path:*

$$\widehat{\Gamma}^M_{\lambda,\varepsilon} := \left\{\beta \in \mathbb{R}^M \ : \ d(\beta, \widehat{\Gamma}^M_\lambda) \leq \varepsilon\right\}. \tag{36}$$

Note that we always have $\widehat{\Gamma}^M_\lambda \subset \mathcal{B}_{\mathbb{R}^M}(\|\beta_{M,\lambda}\|)$. We now state a lemma proving that all the values visited during the algorithm will lie in an approximation of this path.

**Lemma 15.** *Define Let $\beta \in \mathbb{R}^M$ such that $\widehat{\nu}_{M,\mu}(\beta) \leq \frac{\mu^{1/2}}{7R}$ for some $\mu \geq \lambda$. Then the following holds:*

$$\beta \in \widehat{\Gamma}^M_{\lambda, \frac{1}{6R}}.$$

*Proof.* Bound

$$R\|\beta - \beta_{M,\mu}\| \leq \frac{R}{\mu^{1/2}}\|\beta - \beta_{M,\mu}\|_{\widehat{\mathbf{H}}_{M,\mu}(\beta)} \leq \frac{1}{\underline{\phi}(\mathsf{t}_M(\beta - \beta_{M,\mu}))} \frac{R\widehat{\nu}_{M,\mu}(\beta)}{\mu^{1/2}}.$$

Just apply Eq. (18) to obtain $R\|\beta - \beta_{M,\mu}\| \leq \frac{1}{6}$. $\qquad\square$

We now introduce the following quantities which will allow to control the number of sub-samples throughout the whole algorithm.

**Definition 7.** *Define*

- $\overline{\mathsf{b}}_2 := \sup_{\beta \in \widehat{\Gamma}^M_{\lambda,1/6R}} \mathsf{b}_2(f_\beta)$.

- $\overline{\mathcal{N}}^{\mathbf{H}}(\lambda) = \sup_{\beta \in \widehat{\Gamma}^M_{\lambda,1/6R}} \mathcal{N}^{\mathbf{H}(f_\beta)}(\lambda)$.

- $\overline{\mathcal{N}}^{\mathbf{H}}_\infty(\lambda) = \sup_{\beta \in \widehat{\Gamma}^M_{\lambda,1/6R}} \mathcal{N}^{\mathbf{H}(f_\beta)}_\infty(\lambda)$.

- $\overline{\|\mathbf{H}\|} = \min_{\beta \in \widehat{\Gamma}^M_{\lambda,1/6R}} \|\mathbf{H}(f_\beta)\|$.

**Proposition 13** (Performance of the globalization scheme). *Let $\varepsilon > 0$, $\delta > 0$, $\tilde{\lambda} = \min(\lambda, \overline{\|\mathbf{H}\|})$. Assume $\frac{19\overline{\mathsf{b}}_2\kappa^2}{n}\log\frac{n}{2\delta} < \tilde{\lambda}$ and $n \geq 405\overline{\mathsf{b}}_2\kappa^2 \vee 67\overline{\mathsf{b}}_2\kappa^2\log\frac{12\overline{\mathsf{b}}_2\kappa^2}{\delta}$.*

*Assume we perform the globalization scheme with the parameters in Thm. 1, where in order to compute any $\rho$ approximation of a regularized Newton step, we use a conjugate gradient descent on the pre-conditioned system, where the pre-conditioner is computed as in Proposition 12 using*

**1.** $Q \geq \left(10 + 160\overline{\mathcal{N}}^{\mathbf{H}}_\infty(\tilde{\lambda})\right)\log\frac{8\overline{\mathsf{b}}_2\kappa^2}{\tilde{\lambda}\delta}$ *if using uniform sampling*

**2.** $Q \geq \left(6 + 486q^2\overline{\mathcal{N}}^{\mathbf{H}}(\tilde{\lambda})\right)\log\frac{8\overline{\mathsf{b}}_2\kappa^2}{\tilde{\lambda}\delta}$ *if using Nyström leverage scores*

*Recall that $\mathsf{t}$ denotes the number of approximate Newton steps performed at for each $\mu$ in Phase I and $T$ denotes the number of approximate Newton steps performed in Phase II, and that using Thm. 1, $\mathsf{t} = 2$ and $T = \lceil\log_2\sqrt{1 \vee (\lambda\varepsilon^{-1}/R^2)}\rceil$. Moreover, recall that $K$ denotes the number of steps performed in Phase I. Define*

$$N_{ns} := 2\left\lfloor(3 + 11R\|\beta_{M,\lambda}\|)\log_2(7R\|\nabla\widehat{L}_M(0)\|/\lambda)\right\rfloor + \lceil\log_2\sqrt{1 \vee (\lambda\varepsilon^{-1}/R^2)}\rceil.$$

*Then with probability at least $(1 - \delta)^{N_{ns}}$:*

- *The method presented in Proposition 12 returns a $1/7$- approximate Newton step at each time it is called in the algorithm.*

- *If $\beta$ denotes the result of the method, $\widehat{\nu}_{M,\lambda}(\beta) \le \sqrt{\varepsilon}$.*

- *The number of approximate Newton steps computed during the algorithm is bounded by $N_{ns}$; the complexity of the method is therefore of order $O(N_{ns}(M^2 \max(M,Q) + nM + \mathsf{c}_{samp}(\lambda)))$ in time and $O(MQ + M^2 + n)$ in memory, where $\mathsf{c}_{samp}(\lambda)$ is a bound on the complexity associated to the computing of leverage scores (see [30] for details).*

*The algorithm is detailed in Appendix E, in algorithm 1. Note however that the notations are those of the main paper, which are slightly different from the ones used here.*

*Proof.* If we take the globalization scheme, using the parameters of Thm. 1. Assume that all previous approximate Newton steps have been computed in a good way. Then the $\beta$ at which we are belongs to $\widehat{\Gamma}^M_{\lambda,1/6R}$. Thus, the hypotheses of this proposition imply that the hypothesis of Proposition 12 are satisfied; and hence, up to a $(1-\delta)$ probability factor, we can assume that the next approximate Newton step is performed correctly, continuing the globalization scheme in the right way. Thus, the globalization scheme converges as in Thm. 1. $\qquad\square$

### D.6 Statistical properties of the algorithm

The following theorem describes the computational and statistical behavior of our algorithm.

**Proposition 14** (Behavior of an approximation to the projected problem)**.** *Suppose that Assumptions 1 to 3 are satisfied.*
*Let $n \in \mathbb{N}$, $\varepsilon > 0$, $\delta \in (0, 1/2]$, $0 < \lambda \le \mathsf{B}_2^\star$.*
*Define $\tilde{\lambda} = \min(\lambda, \overline{\|\mathbf{H}\|})$ and assume $\frac{19\overline{\mathsf{b}}_2\kappa^2}{n} \log \frac{n}{2\delta} < \tilde{\lambda}$, $n \ge 405\overline{\mathsf{b}}_2\kappa^2 \vee 67\overline{\mathsf{b}}_2\kappa^2 \log \frac{12\overline{\mathsf{b}}_2\kappa^2}{\delta}$, and $n \ge \triangle_1 \frac{\mathsf{B}_2^\star}{\lambda} \log \frac{8\square_1^2\mathsf{B}_2^\star}{\lambda\delta}$. Assume*

$$\mathsf{C}_1 \sqrt{\frac{\mathsf{df}_\lambda \vee (\mathsf{Q}^\star)^2}{n} \, \log \frac{2}{\delta}} \le \frac{\lambda^{1/2}}{R}, \qquad \mathsf{C}_1 \mathsf{b}_\lambda \le \frac{\lambda^{1/2}}{R}, \qquad 126\sqrt{\varepsilon} \le \frac{\lambda^{1/2}}{R}.$$

*Assume that the $M$ points $\tilde{x}_1, ..., \tilde{x}_M$ are obtained through Nyström sub-sampling using $\eta = \|\mathbf{\Sigma}\| \wedge \frac{\lambda\sqrt{2}}{1440(\mathsf{b}_2^\star \vee 1)}$, with either*

**1.** $M \ge \left(10 + 160\mathcal{N}^{\mathbf{\Sigma}}_\infty(\eta)\right) \log \frac{8\kappa^2}{\eta\delta}$ *if using uniform sampling;*

**2.** $M \ge \left(6 + 486q^2\mathcal{N}^{\mathbf{\Sigma}}(\eta)\right) \log \frac{8\kappa^2}{\eta\delta}$ *if using $q$-approximate leverage scores for $\eta$, associated to the co-variance operator $\mathbf{\Sigma}$.*

*Assume we perform the globalization scheme as in Proposition 13, i.e. with the parameters in Thm. 1, where in order to compute any $\rho$ approximation of a regularized Newton step, we use a conjugate gradient descent on the pre-conditioned system, where the pre-conditioner is computed as in Proposition 12 using*

**1.** $Q \ge \left(10 + 160\overline{\mathcal{N}}^{\mathbf{H}}_\infty(\tilde{\lambda})\right) \log \frac{8\overline{\mathsf{b}}_2\kappa^2}{\tilde{\lambda}\delta}$ *if using uniform sampling*

**2.** $Q \ge \left(6 + 486q^2\overline{\mathcal{N}}^{\mathbf{H}}(\tilde{\lambda})\right) \log \frac{8\overline{\mathsf{b}}_2\kappa^2}{\tilde{\lambda}\delta}$ *if using Nyström leverage scores*

*Let $N_{ns}$ be defined as in Proposition 13. Recall $N_{ns}$ is an upper bound for the number of approximate Newton steps performed in the algorithm. One can bound*

$$N_{ns} \le 2 \left\lfloor (113 + 11R\|f^\star\|) \log_2 \frac{7R\|\nabla\widehat{L}_M(0)\|}{\lambda} \right\rfloor + \left\lceil \log_2 \frac{\lambda^{1/2}}{R\varepsilon} \right\rceil.$$

*Moreover, with probability at least $1 - (N_{ns} + 2)\delta$, the following holds:*

$$L(f_\beta) - L(f^\star) \le \mathsf{K}_1 \, \mathsf{b}_\lambda^2 + \mathsf{K}_2 \, \frac{\mathsf{df}_\lambda \vee (\mathsf{Q}^\star)^2}{n} \, \log \frac{2}{\delta} + \mathsf{K}_3 \, \varepsilon.$$

*where $\mathsf{K}_1 \le 6.0\mathrm{e}4$, $\mathsf{K}_2 \le 6.0\mathrm{e}6$ and $\mathsf{K}_3 \le 810$, $\mathsf{C}_1$ is defined in Lemma 19, and the other constants are defined in Thm. 8.*

*Proof.* This is a simple combination between Propositions 10, 11 and 13. To bound the number of Newton steps $N_{ns}$, one simply uses the fact that under the conditions of the theorem, $R\|\beta_{M,\lambda}\| \leq 10 + R\|f^\star\|$. $\qquad\square$

**Remark 4** (Complexity). *Let $L = \bar{\mathsf{b}}_2 \kappa^2$. The complexity of the previous method using leverage scores computed for $\Sigma$ for the Nystrom projections and for $\mathbf{H}(f_\beta)$ for choosing the Q points at the different stages is the following. The total complexity in time will be of order:*

$$O\left(N_{ns}\left(n\mathcal{N}^{\overline{\mathbf{H}}}(\lambda)\log(L\lambda^{-1}\delta^{-1}) + \bar{\mathsf{b}}_2^3 \mathcal{N}^{\mathbf{\Sigma}}(\lambda)^3 \log^3(L\lambda^{-1}\delta^{-1}) + L/\lambda\,\bar{\mathsf{b}}_2^2 \mathcal{N}^{\mathbf{\Sigma}}(\lambda)^2\right)\right).$$

*The memory complexity can be bounded by*

$$O(\bar{\mathsf{b}}_2^2 \mathcal{N}^{\mathbf{\Sigma}}(\lambda)^2 \log^2(L\lambda^{-1}\delta^{-1}) + n).$$

*Here, we use the fact that $\mathbf{H} \leq \bar{\mathsf{b}}_2 \mathbf{\Sigma}$.*

We can now write down the previous proposition by classifying problems using Assumptions 4 and 5 and in order to get optimal rates.

**Theorem 7** (Performance of the scheme using pre-conditioning). *Let $\delta > 0$. Assume Assumptions 1 to 5 are satisfied. Let $n \geq \tilde{N}$, where $\tilde{N}$ is characterized in the proof, $\lambda = \left(\left(\frac{\mathsf{Q}}{\mathsf{L}}\right)^2 \frac{1}{n}\right)^{\frac{\alpha}{\alpha(1+2r)+1}}$.*
*Assume that the M points $\tilde{x}_1,...,\tilde{x}_M$ are obtained through Nyström sub-sampling using $\eta = \frac{\lambda\sqrt{2}}{1440(\mathsf{b}_2^\star \vee 1)}$, with either*
**1.** $M \geq \left(10 + 160\mathcal{N}_\infty^{\mathbf{\Sigma}}(\eta)\right)\log\frac{8\kappa^2}{\eta\delta}$ *if using uniform sampling;*
**2.** $M \geq \left(6 + 486q^2\mathcal{N}^{\mathbf{\Sigma}}(\eta)\right)\log\frac{8\kappa^2}{\eta\delta}$ *if using q-approximate leverage scores for $\eta$, associated to the co-variance operator $\mathbf{\Sigma}$.*

*Assume we perform the globalization scheme as in Proposition 13, i.e. with the parameters in Thm. 1, where in order to compute any $\rho$ approximation of a regularized Newton step, we use a conjugate gradient descent on the pre-conditioned system, where the pre-conditioner is computed as in Proposition 12 using*
**1.** $Q \geq \left(10 + 160\overline{\mathcal{N}}_\infty^{\mathbf{H}}(\lambda)\right)\log\frac{8\bar{\mathsf{b}}_2\kappa^2}{\lambda\delta}$ *if using uniform sampling*
**2.** $Q \geq \left(6 + 486q^2\overline{\mathcal{N}}^{\mathbf{H}}(\lambda)\right)\log\frac{8\bar{\mathsf{b}}_2\kappa^2}{\lambda\delta}$ *if using Nyström leverage scores*
*Let $N_{ns}$ be defined as in Proposition 13. Recall $N_{ns}$ is an upper bound for the number of approximate Newton steps performed in the algorithm. One can bound*

$$N_{ns} \leq (227 + 22R\|f^\star\|)\left(\left\lceil\log_2\left(7R\|\nabla\widehat{L}_M(0)\|\right)\right\rceil + \left\lceil\log_2\frac{n\mathsf{L}^2}{\mathsf{Q}^2}\right\rceil + \left\lceil\log_2\frac{1}{R\mathsf{L}}\right\rceil\right).$$

*Moreover, with probability at least $1 - (N_{ns} + 2)\delta$, the following holds:*

- *all of the approximate Newton methods yield $\frac{1}{7}$-approximate Newton steps*

- *The scheme finishes, and the number of approximate Newton steps is bounded by $N_{ns}$. The total complexity of the method is therefore*

$$O((nM + M^3 + M^2Q + \mathsf{c}_{samp})N_{ns}) \text{ in time}, \qquad O(n + M^2) \text{ in memory}.$$

- *The returned $\beta$ is statistically optimal:*

$$L(f_\beta) - L(f^\star) \leq \mathsf{K}\left(\mathsf{Q}^\gamma\,\mathsf{L}^{1-\gamma}\right)^2 \frac{1}{n^\gamma}\log\frac{2}{\delta},$$

  *where $\mathsf{K}$ is defined in Thm. 5.*

*Proof.* The proof consists mainly of combining Propositions 11 and 13 and Thm. 5.

Recall that we set $\lambda = \left(\frac{\mathsf{Q}^2}{\mathsf{L}^2}\frac{1}{n}\right)^{\frac{\alpha}{\alpha(1+2r)+1}}$.

1. Start by defining $\tilde{N}$ such that:

- $\tilde{N} \geq N$ where $N$ is defined in Thm. 5;

- $\forall n \geq \tilde{N}$, $\lambda \leq \overline{\|\mathbf{H}\|}$. This is possible as $\frac{\alpha}{\alpha(1+2r)+1}$ is a strictly positive exponent.

- $\forall n \geq \tilde{N}$, $\frac{19\overline{\mathsf{b}}_2 \vee 1 \; \kappa^2}{n} \log \frac{n}{2\delta} < \lambda$; this is possible as soon as $\frac{\alpha}{\alpha(1+2r)+1} < 1$, i.e. this is satisfied since $r > 0$;

- $\tilde{N} \geq 405\overline{\mathsf{b}}_2 \vee 1 \; \kappa^2 \vee 67\overline{\mathsf{b}}_2 \vee 1 \; \kappa^2 \log \frac{12\overline{\mathsf{b}}_2 \vee 1 \; \kappa^2}{\delta}$;

- $\forall n \geq \tilde{N}$, $\frac{\lambda\sqrt{2}}{1440(\mathsf{b}_2^\star \vee 1)} \leq \|\mathbf{\Sigma}\|$.

We see that such a $\tilde{N}$ can be defined explicitly.

2. Combining the assumptions on $\tilde{N}$ with the ones on $M$, we see that all the assumptions of Proposition 11 are satisfied and thus that with probability at least $1 - \delta$, all the hypotheses for Thm. 5 are satisfied except the bound on $\widehat{\nu}_{M,\lambda}(\beta)$.

3. Applying Proposition 13, taking $\sqrt{\varepsilon} = \mathsf{Q}^\gamma \, \mathsf{L}^{1-\gamma} n^{-\gamma/2}$ and $\lambda = \left(\frac{\mathsf{Q}^2}{\mathsf{L}^2}\frac{1}{n}\right)^{\frac{\alpha}{\alpha(1+2r)+1}}$, we see that under these hypotheses,

$$N_{ns} := 2 \left\lfloor (3 + 11R\|\beta_{M,\lambda}\|) \log_2 \left( 7R\|\nabla\widehat{L}_M(0)\| \left(\frac{n\mathsf{L}^2}{\mathsf{Q}^2}\right)^{\frac{\alpha}{\alpha(1+2r)+1}} \right) \right\rfloor + \left\lceil \log_2 \left( \frac{1}{R\mathsf{L}}\left(\frac{n\mathsf{L}^2}{\mathsf{Q}^2}\right)^{\frac{r\alpha}{\alpha(1+2r)+1}} \right) \right\rceil .$$

Now we can bound this harshly:

$$N_{ns} \leq (7 + 22R\|\beta_{M,\lambda}\|) \left( \left\lceil \log_2 \left( 7R\|\nabla\widehat{L}_M(0)\| \right) \right\rceil + \left\lceil \log_2 \frac{n\mathsf{L}^2}{\mathsf{Q}^2} \right\rceil + \left\lceil \log_2 \frac{1}{R\mathsf{L}} \right\rceil \right) .$$

Now bounding $R\|\beta_{M,\lambda}\| \leq 10 + R\|f^\star\|$, we get

$$N_{ns} \leq (227 + 22R\|f^\star\|) \left( \left\lceil \log_2 \left( 7R\|\nabla\widehat{L}_M(0)\| \right) \right\rceil + \left\lceil \log_2 \frac{n\mathsf{L}^2}{\mathsf{Q}^2} \right\rceil + \left\lceil \log_2 \frac{1}{R\mathsf{L}} \right\rceil \right) .$$

4. Finally, we use a union bound to conclude. $\square$

# E Algorithm

**Algorithm 1** Algorithm efficient non-parametric learning for generalized self-concordant losses with optimal statistical guarantees discussed in Sec. 4 of the main paper.

---

**Input:** $(x_i, y_i)_{i=1}^n$, $n \in \mathbb{N}$, $\ell$ loss function, $k$ kernel function and $\lambda > 0$.
**Return:** estimated function $\widehat{g} : \mathcal{X} \to \mathbb{R}$
Parameters: $Q, M, T \in \mathbb{N}$, $\mu_0 > 0$, $(q_k)_{k \in \mathbb{N}}$.
Fixed parameters: $t = 2$ from Thm. 1, $\tau = 3$ from Proposition 12 in Appendix D.5.1.
$(\bar{x}_j)_{j=1}^M \leftarrow$ leverage-scores-sampling$((x_i)_{i=1}^n, M, \lambda, k)$
$\mathbf{K} \leftarrow$ kernel-matrix$((\bar{x}_j)_{j=1}^M, (\bar{x}_j)_{j=1}^M)$
$\mathbf{T} \leftarrow$ cholesky-upper-triangular$(\mathbf{K})$
define the function $v(\cdot) = (k(\bar{x}_1, \cdot), \dots, k(\bar{x}_M, \cdot)) \in \mathbb{R}^M$

> **define compute-preconditioner:**
> **Input:** $\alpha \in \mathbb{R}^M, \lambda > 0$
> $c_i \leftarrow \sqrt{\ell^{(2)}(v(x_i)^\top \mathbf{T}^{-1}\alpha, y_i)}$ for all $i = 1, \dots, n$
> define the function $k'(\circ, \bullet)$ as $k'(\circ, \bullet) := c_\circ \times c_\bullet \times k(x_\circ, x_\bullet)$ for $\circ, \bullet \in \{1, \dots, n\}$
> $(h_s)_{s=1}^Q \leftarrow$ leverage-scores-sampling$((i)_{i=1}^n, Q, \lambda, k')$
> $\mathbf{G} \leftarrow$ kernel-matrix$((\bar{x}_j)_{i=1}^M, (x_{h_s})_{s=1}^Q, k)$
> $\mathbf{H} \leftarrow \mathbf{T}^{-\top} \times \mathbf{G} \times \text{diag}((c_{l_h}^2)_{h=1}^Q) \times \mathbf{G}^\top \times \mathbf{T}^{-1}$
> $\mathbf{B} \leftarrow$ cholesky-upper-triangular$(\frac{1}{Q}\mathbf{H} + \lambda I)$
> return $\mathbf{B}$

> **define preconditioned-conj-grad:**
> **Input:** $\alpha \in \mathbb{R}^M, \mu > 0, r \in \mathbb{R}^M, \tau \in \mathbb{N}, \mathbf{B} \in \mathbb{R}^{M \times M}$
> $p \leftarrow r, s_0 \leftarrow \|r\|^2, \beta \leftarrow 0$
> For $i = 1, \dots, \tau$
> $\quad z \leftarrow \mu \mathbf{B}^{-\top}\mathbf{B}^{-1}p + \frac{1}{n}\sum_{i=1}^n \ell^{(2)}(v(x_i)^\top \mathbf{T}^{-1}\alpha, y_i)\,(v(x_i)^\top \mathbf{T}^{-1}\mathbf{B}^{-1}p)\,\mathbf{B}^{-\top}\mathbf{T}^{-\top}v(x_i)$
> $\quad a \leftarrow s_0/(p^\top z)$
> $\quad \beta \leftarrow \beta + ap$
> $\quad r \leftarrow r - az, \;\; s_1 \leftarrow \|r\|^2$
> $\quad p \leftarrow r + (s_1/s_0)p$
> $\quad s_0 \leftarrow s_1$
> return $\beta$

> **define appr-linear-solver:**
> **Input:** $\alpha \in \mathbb{R}^M, \mu > 0, g \in \mathbb{R}^M$
> $\mathbf{B} \leftarrow$ compute-preconditioner$(\alpha, \mu)$
> $u \leftarrow$ preconditioned-conjugate-gradient$(\alpha, \mu, \mathbf{B}^{-\top}g, \tau = 3, \mathbf{B})$
> return $\mathbf{B}^{-1}u$

> **define approximate-Newton:**
> Input: $\alpha_0 \in \mathbb{R}^M, \mu > 0, t \in \mathbb{N}$
> For $j = 1, \dots, t$
> $\quad g \leftarrow \mu \alpha_{j-1} + \frac{1}{n}\sum_{i=1}^n \ell^{(1)}(v(x_i)^\top \mathbf{T}^{-1}\alpha_{j-1}, y_i)\,\mathbf{T}^{-\top}v(x_i)$
> $\quad \alpha_j \leftarrow \alpha_{j-1} -$ appr-linear-solver$(\alpha_{j-1}, \mu, g)$
> return $\alpha_t$

$\alpha_0 \leftarrow 0$
For $k \in \mathbb{N}$
$\quad \alpha_{k+1} \leftarrow$ approximate-Newton$(\alpha_k, \mu_k, t = 2)$
$\quad \mu_{k+1} \leftarrow q_{k+1}\mu_k$
Stop when $\mu_{k+1} < \lambda$ and set $\alpha_{last} \leftarrow \alpha_k$
$\widehat{\alpha} \leftarrow$ approximate-Newton$(\alpha_{last}, \lambda, T)$
return $\widehat{g}(\cdot) := v(\cdot)^\top \mathbf{T}^{-1}\widehat{\alpha}$

---

Let $N, M \in \mathbb{N}$ with $M \leq N$. In Alg. 1, leverage-scores-sampling$((z_i)_{i=1}^N, M, k, \lambda)$ returns a subset of $(z_i)_{i=1}^N$ of cardinality $M$ sampled by using (approximate) leverage scores at scale $\lambda > 0$ and computed using the kernel $k$. An explicit example of an algorithm computing leverage-scores-sampling is in [30]. Moreover kernel-matrix$((x_i)_{i=1}^N, (x_i')_{i=1}^M, k)$ computes the kernel matrix $K \in \mathbb{R}^{N \times M}$ where $K_{ij} = k(x_i, x_j')$, with $N, M \in \mathbb{N}$.

# F  Experiments

We present our algorithm's performance for logistic regression on two large scale data sets: Higgs and Susy. We have implemented our method using pytorch, and performed computations on one node of a Tesla P100-PCIE-16GB GPU. Recall that in the case of logistic regression, $\ell_{(x,y)}(t) = \log(1 + e^{-yt})$.

In what follows, denote with $n$ the cardinality of the data set and $d$ the number of features of this data set. The error is measured in terms of classification error for both data sets. In both cases, we pre-process the data by substracting the mean and dividing by the standard deviation for each feature. The data sets are the following.

**Susy**  ($n = 5 \times 10^6$, $d = 18$, binary classification). We always use a Gaussian Kernel with $\sigma = 5$ for logistic loss (obtained through a grid search; note that in [29], $\sigma = 4$ is used for the square loss), and will always use $10^4$ Nystrom points.

**Higgs**  ($n = 1.1 \times 10^7$, $d = 28$, binary classification). We then apply a Gaussian Kernel with $\sigma = 5$, as in [29] (we have also performed a grid search).

For these data sets, we do not have a fixed test set, and thus set apart $20\%$ of the data set at random to be the test set, and use the rest of the $80\%$ to train the classifier.

In practice, we perform our globally convergent scheme with the following parameters.

- We use $Q = M$ uniform random features to compute the pre-conditioner for each approximate Newton step;

- In the first phase, we decrease $\mu$ in a very fast way to $\lambda$ by starting at $\mu = 1$ and dividing $\mu$ by 1000 after performing only a single approximate Newton step (using 2 iterations of conjugate gradient descent);

- In the second phase, we perform 10 approximate Newton steps (each ANS is computed using 8 iterations of conjugate gradient descent).

**Selection of $\lambda$**  In the introduction, we claim that in many a learning problem, the parameter $\lambda$ obtained through cross validation is often much smaller than the ones obtained in statistical bounds which are usually of order $\frac{1}{\sqrt{n}}$. This leads to very ill conditioned problems.

For both data sets, we select $\lambda$ (and $\sigma$, but we omit the double tables from this paper) by computing the test loss and classification errors for different values of $\lambda$, and report the evolution of these losses as a function of the parameter $\lambda$ in Fig. 2 for the Higgs data set, and Fig. 3 for the Susy data set. We see that the optimal $\lambda$ yield strongly ill-conditioned problems.

Figure 2:  **(Left)** Classification error as a function of the regularization parameter and **(Right)** test loss as a function of the regularization parameter, when performing a logistic regression with $M = 2 \times 10^4$ Nyström features on the entire Higgs data set; we select $\lambda = 10^{-9}$.

Figure 3: **(Left)** Classification error as a function of the regularization parameter and **(Right)** test loss as a function of the regularization parameter, when performing a logistic regression with $M = 10^4$ Nyström features on the entire Susy data set; we select $\lambda = 10^{-10}$.

**Comparison with accelerated methods**  Given the $M$ Nystrom points, our aims to minimize $\widehat{L}_{M,\lambda}$. From an optimization point of view, i.e. from a point of view where the aim is to minimize $\widehat{L}_{M,\lambda}$, we compare our method with a large mini-batch version of Katyusha accelerated SVRG (see [4]).

Indeed, we perform this method using batch sizes of size $M$; the theoretical bounds provided in [4] show that the algorithm has linear convergence, with a time complexity of order $O(nM + M^3 + M^2\sqrt{\frac{L}{\lambda}})\log\frac{1}{\varepsilon}$ to reach precision $\varepsilon$. In the following plots, we compare both methods in terms of passes and time.

By pass, we mean the following.

- In the case of our second-order scheme, we define a pass on the data to be one step of the conjugate gradient descent used to compute approximate newton steps.

- In the case of Katyusha SVRG, we define a pass on the data to be either a full gradient computation or $n/M$ computations of the type $K_{\tau M}T^{-1}\beta$ where $T$ is an upper triangular matrix, and $K_{\tau M}$ is a $M \times M$ kernel matrix, associated to one batch gradient.

We use this notion to measure the speed of our method as they both correspond to natural $O(nM)$ operations, and incorporate the essential of the computing time. However, the second point is often much slower to compute than the first, due to the solving of the triangular system. Thus, the notion of passes is to take with precaution, as a pass for the accelerated SVRG algorithm takes much longer to run that a pass for our method. This is confirmed by the time plots (see Fig. 5 for in instance).

*Comparison between the two methods* - Due to the running time of K-SVRG, we compare both methods for $M = 10000$ Nyström points for both data sets. We compare the performance of these two algorithm with respect to the distance to the optimum in function values as well as classification error Fig. 4 for the Higgs data set, and in Fig. 5 for the Susy data set.

*Note on the need for precise optimization* - As noted in the introduction, we see in both Fig. 5 and Fig. 4 that precise optimization of the objective function is needed in order to get a good classification error. This justifies a posteriori the use of a second order method. In particular, in Fig. 5, one notes the difference in behavior between the two methods : the second order method converges linearly in a fast way while the first order method slows down because of the condition number.

*Note on ill-conditioning* - First note that in order to optimize test error, one gets very poorly conditioned problems. As predicted by the rates, we observe that K-SVRG is more sensible to ill-conditioning than our second order scheme. Indeed, in Fig. 6, we have plotted the results for Susy for a smaller condition number with $\lambda = 10^{-8}$, compared to $\lambda = 10^{-10}$ to get optimal test error in Fig. 5. We see that the difference in number of passes needed to reach a certain precision is much lower when $\lambda = 10^{-8}$ in Fig. 6, confirming that K-SVRG behaves better when the condition number is smaller.

Figure 4: **(Left)** Distance to optimum as a function of time and **(Right)** distance to optimum and classification error as a function of the number of passes on the data when performing our second order scheme and K-SVRG to minimize the train loss on Higgs, with $1.0 \times 10^4$ Nyström points and $\lambda = 10^{-9}$.

Figure 5: **(Left)** Distance to optimum as a function of time and **(Right)** distance to optimum and classification error as a function of the number of passes on the data when performing our second order scheme and K-SVRG to minimize the train loss on Susy, with $1.0 \times 10^4$ Nyström points and $\lambda = 10^{-10}$.

**Performance of our method.** In Table 1, we record the performance of the following methods, taking the $\lambda$ values we have obtained previously for the different data sets.

For FALKON (see [29]), we take the parameters suggested in the paper (except for the number of Nyström points needed for Higgs, as our computational capacity is limited).

| Method | Susy | | | Higgs | | |
|---|---|---|---|---|---|---|
| | c-error | $M$ | time(m) | c-error | $M$ | time(m) |
| Logistic regression with Katyusha SVRG | 19.64% | $10^4$ | 230 | 27.82 % | $10^4$ | 500 |
| Logistic regression with our scheme | 19.5% | $10^4$ | 15 | 26.9 % | $2.5 \times 10^4$ | 65 |
| Ridge Regression with FALKON ([29]) | 19.7% | $10^4$ | 5 | 27.16 % | $2.5 \times 10^4$ | 60 |

Table 1: Classification error of different methods

Figure 6: **(Left)** Distance to optimum as a function of time and **(Right)** distance to optimum and classification error as a function of the number of passes on the data when performing our second order scheme and K-SVRG to minimize the train loss on Susy, with $1.0 \times 10^4$ Nyström points and $\lambda = 10^{-8}$.

# G Solving a projected problem to reduce dimension

## G.1 Introduction and notations

In this section, we give ourselves a generalized self-concordant function $f$ whose associated subset we denote with $\mathcal{G}$. Once again, we will always omit the subscript $f$ in the notations associated to $f$.

The aim of this section is the following. Given $f$ and $\lambda > 0$, computing an approximate solution to

$$x_\lambda^\star = \arg\min_{x \in \mathcal{H}} f_\lambda(x),$$

is often too costly. Instead, we look for a solution in a small subset of $\mathcal{H}$ which we see as the image of a certain orthogonal projector $\mathbf{P}$ and which we denote $\mathcal{H}_{\mathbf{P}}$. Usually, this subset will be finite dimensional and admit an easy parametrization. Thus we will compare an approximation of $x_\lambda^\star$ to an approximation of

$$x_{\mathbf{P},\lambda}^* = \arg\min_{x \in \mathcal{H}_{\mathbf{P}}} f_\lambda(x) = \arg\min_{x \in \mathcal{H}} f(\mathbf{P}x) + \frac{\lambda}{2}\|x\|^2.$$

Denote with $f_{\mathbf{P}}$ the mapping $x \in \mathcal{H} \mapsto f(\mathbf{P}x)$. It is easy to see that, as $f$ is a generalized self-concordant function with $\mathcal{G}$, $f_{\mathbf{P}}$ is naturally a generalized self-concordant with $\mathcal{G}_{\mathbf{P}} := \mathbf{P}\mathcal{G} = \{\mathbf{P}g \; : \; g \in \mathcal{G}\}$. Moreover, $x_{\mathbf{P},\lambda}^* = x_{f_{\mathbf{P}},\lambda}^\star$.

We will adopt the following notations for the quantities related to the generalized self-concordant function $f_{\mathbf{P}}$. Essentially, we always replace $f_{\mathbf{P}}$ simply by $\mathbf{P}$ from our definitions in appendix.

- For the regularized function :

$$\forall x \in \mathcal{H}, \; \forall \lambda > 0, \; f_{\mathbf{P},\lambda}(x) = f_{\mathbf{P}}(x) + \frac{\lambda}{2}\|x\|^2.$$

- For the Hessians

$$\forall x \in \mathcal{H}, \; \lambda > 0, \; \mathbf{H}_{\mathbf{P},\lambda}(x) = \mathbf{H}_{f_{\mathbf{P}},\lambda}(x) = \mathbf{P}\mathbf{H}(\mathbf{P}x)\mathbf{P} + \lambda\mathbf{I}.$$

- $\forall h \in \mathcal{H}, \; \mathsf{t}_{\mathbf{P}}(h) := \mathsf{t}_{f_{\mathbf{P}}}(h) = \mathsf{t}(\mathbf{P}h).$

- For the Newton decrement:

$$\forall x \in \mathcal{H}, \; \lambda > 0, \; \nu_{\mathbf{P},\lambda}(x) = \nu_{f_{\mathbf{P}},\lambda}(x) = \|\nabla f_{\mathbf{P},\lambda}\|_{\mathbf{H}_{\mathbf{P},\lambda}^{-1}(x)} = \|\mathbf{P}\nabla f(\mathbf{P}x) + \lambda x\|_{\mathbf{H}_{\mathbf{P},\lambda}^{-1}(x)}.$$

- For the Dikin ellipsoid radius:

$$\forall \lambda > 0, \; \forall x \in \mathcal{H}, \; \mathsf{r}_{\mathbf{P},\lambda}(x) := \mathsf{r}_{f_{\mathbf{P}},\lambda}(x) = \frac{1}{\sup_{g \in \mathcal{G}} \|\mathbf{P}g\|_{\mathbf{H}_{\lambda,\mathbf{P}}^{-1}(x)}};$$

- For the Dikin ellipsoid:

$$\forall \lambda > 0, \ \forall c \geq 0, \ \mathsf{D}_{\mathbf{P},\lambda}(c) := \mathsf{D}_{f_{\mathbf{P}},\lambda}(c).$$

Note that for any $x \in \mathcal{H}_{\mathbf{P}}$, $\mathsf{r}_{\mathbf{P},\lambda}(x) \geq \mathsf{r}_\lambda(x)$.

We will now introduce the key quantities in order to compare an approximation of $x^*_{\mathbf{P},\lambda}$ to an approximation of $x^\star_\lambda$.

**Definition 8** (key quantities). *Define the following quantities*

- *For any $\lambda > 0$, the source term* $\mathsf{s}_\lambda := \lambda \|x^\star_\lambda\|_{\mathbf{H}^{-1}_\lambda(x^\star_\lambda)} = \|\nabla f(x^\star_\lambda)\|_{\mathbf{H}^{-1}_\lambda(x^\star_\lambda)}$;

- *Given an orthogonal projector $\mathbf{P}$, $\lambda > 0$, and $x \in \mathcal{H}$, the capacity of the projector* $\mathsf{C}_{\mathbf{P}}(x,\lambda) := \frac{\|\mathbf{H}(x)^{1/2}(\mathbf{I}-\mathbf{P})\|^2}{\lambda}$.

### G.2 Relating the projected to the original problem

Given $x \in \mathcal{H}_{\mathbf{P}}$, our aim is to bound $\nu_\lambda(x)$ given $\nu_{\lambda,\mathbf{P}}(x)$ and $\mathsf{s}_\lambda$.

**Proposition 15.** *Let $x \in \mathcal{H}_{\mathbf{P}}$. If*

$$\frac{\mathsf{s}_\lambda}{\mathsf{r}_\lambda(x^\star_\lambda)} \leq \frac{1}{4}, \ \mathsf{C}_{\mathbf{P}}(x^\star_\lambda, \lambda) \leq \frac{1}{120}, \ \nu_{\mathbf{P},\lambda}(x) \leq \frac{\mathsf{r}_{\mathbf{P},\lambda}(x)}{2},$$

*Then it holds:*

$$\nu_\lambda(x) \leq 3(\nu_{\mathbf{P},\lambda}(x) + \mathsf{s}_\lambda).$$

*Moreover, under these conditions,*

- $\|x - x^\star_\lambda\| \leq 7\lambda^{-1/2}(\nu_{\mathbf{P},\lambda}(x) + \mathsf{s}_\lambda)$;

- $\lambda \|x\|_{\mathbf{H}^{-1}_{\mathbf{P},\lambda}(x)} \leq 7\nu_{\mathbf{P},\lambda}(x) + 9\mathsf{s}_\lambda.$

*Proof.* In this proof, introduce the following auxiliary quantity:

$$\gamma_\lambda := \frac{\mathsf{s}_\lambda}{\mathsf{r}_\lambda(x^\star_\lambda)}.$$

**1) Start by bounding $\mathsf{t}(\mathbf{P}x^\star_\lambda - x^\star_\lambda)$.** It holds:

$$
\begin{aligned}
\mathsf{t}(\mathbf{P}x - x^\star_\lambda) &= \sup_{g \in \mathcal{G}} |g \cdot (\mathbf{I} - \mathbf{P})x^\star_\lambda| \\
&\leq \frac{1}{\mathsf{r}_\lambda(x^\star_\lambda)} \|(\mathbf{I} - \mathbf{P})x^\star_\lambda\|_{\mathbf{H}_\lambda(x^\star_\lambda)} \\
&\leq \frac{1}{\mathsf{r}_\lambda(x^\star_\lambda)} \|\mathbf{H}_\lambda(x^\star_\lambda)^{1/2}(\mathbf{I} - \mathbf{P})\mathbf{H}_\lambda(x^\star_\lambda)^{1/2}\| \ \|\mathbf{H}^{-1/2}_\lambda(x^\star_\lambda)x^\star_\lambda\| \\
&= (1 + \mathsf{C}_{\mathbf{P}}(x^\star_\lambda, \lambda)) \ \frac{\lambda \|\mathbf{H}^{-1/2}_\lambda(x^\star_\lambda)x^\star_\lambda\|}{\mathsf{r}_\lambda(x^\star_\lambda)} \\
&= (1 + \mathsf{C}_{\mathbf{P}}(x^\star_\lambda, \lambda)) \ \gamma_\lambda.
\end{aligned}
$$

**2) Then bound $\mathsf{t}(x^*_{\mathbf{P},\lambda} - \mathbf{P}x^\star_\lambda)$** First, bound $\nu_{\mathbf{P},\lambda}(\mathbf{P}x^\star_\lambda)$:

$$
\begin{aligned}
\nu_{\mathbf{P},\lambda}(\mathbf{P}x^\star_\lambda) &= \|\mathbf{P}\nabla f_\lambda(\mathbf{P}x^\star_\lambda)\|_{\mathbf{H}_{\lambda,\mathbf{P}}(\mathbf{P}x^\star_\lambda)^{-1}} \\
&\leq \|\nabla f_\lambda(\mathbf{P}x^\star_\lambda)\|_{\mathbf{H}_\lambda(\mathbf{P}x^\star_\lambda)^{-1}}.
\end{aligned}
$$

Using Eq. (17), we get $\|\nabla f_\lambda(\mathbf{P}x^\star_\lambda)\|_{\mathbf{H}_\lambda(\mathbf{P}x^\star_\lambda)^{-1}} \leq e^{\mathsf{t}((\mathbf{I}-\mathbf{P})x^\star_\lambda)/2}\nu_\lambda(\mathbf{P}x^\star_\lambda)$. Using Eq. (20), we can bound

$$\nu_\lambda(\mathbf{P}x^\star_\lambda) \leq \overline{\phi}(\mathsf{t}((\mathbf{I} - \mathbf{P})x^\star_\lambda)) \ \|(\mathbf{I} - \mathbf{P})x^\star_\lambda\|_{\mathbf{H}_\lambda(x^\star_\lambda)} \leq \overline{\phi}(\mathsf{t}((\mathbf{I} - \mathbf{P})x^\star_\lambda)) \ (1 + \mathsf{C}_{\mathbf{P}}(x^\star_\lambda, \lambda))\mathsf{s}_\lambda.$$

Putting things together,

$$\nu_{\mathbf{P},\lambda}(\mathbf{P}x_\lambda^\star) \leq e^{\mathsf{t}((\mathbf{I}-\mathbf{P})x_\lambda^\star)/2}\overline{\phi}(\mathsf{t}((\mathbf{I}-\mathbf{P})x_\lambda^\star))\,(1+\mathsf{C}_\mathbf{P}(x_\lambda^\star,\lambda))\mathsf{s}_\lambda.$$

Now

$$\frac{1}{\mathsf{r}_{\mathbf{P},\lambda}(\mathbf{P}x_\lambda^\star)} \leq \frac{1}{\mathsf{r}_\lambda(\mathbf{P}x_\lambda^\star)} \leq e^{\mathsf{t}((\mathbf{I}-\mathbf{P})x_\lambda^\star)/2}\frac{1}{\mathsf{r}_\lambda(x_\lambda^\star)}.$$

Hence,

$$\frac{\nu_{\mathbf{P},\lambda}(\mathbf{P}x_\lambda^\star)}{\mathsf{r}_{\mathbf{P},\lambda}(\mathbf{P}x_\lambda^\star)} \leq e^{\tilde{t}_\lambda}\overline{\phi}(\tilde{t}_\lambda)\,\tilde{t}_\lambda, \qquad \tilde{t}_\lambda = (1+\mathsf{C}_\mathbf{P}(x_\lambda^\star,\lambda))\gamma_\lambda.$$

Since $t \mapsto e^t\overline{\phi}(t)\,t$ is an increasing function whose value in 0 is 0, we find numerically that for $t = \frac{3}{10}$, $e^t\overline{\phi}(t)\,t \leq \frac{1}{2}$. Hence, if $(1+\mathsf{C}_\mathbf{P}(x_\lambda^\star,\lambda))\gamma_\lambda \leq \frac{3}{10}$, then $\frac{\nu_{\mathbf{P},\lambda}(\mathbf{P}x_\lambda^\star)}{\mathsf{r}_{\mathbf{P},\lambda}(\mathbf{P}x_\lambda^\star)} \leq \frac{1}{2}$. Using Lemma 5, this shows that

$$\mathsf{t}_\mathbf{P}(\mathbf{P}x_\lambda^\star - x_{\mathbf{P},\lambda}^*) = \mathsf{t}(\mathbf{P}x_\lambda^\star - x_{\mathbf{P},\lambda}^*) \leq \log 2.$$

**3) Getting a bound for** $\mathsf{t}(x - x_\lambda^\star)$**.** To do so, combine the two previous bounds with the fact that if $\nu_{\mathbf{P},\lambda}(x) \leq \frac{\mathsf{r}_{\mathbf{P},\lambda}(x)}{2}$, then using Lemma 5 with $f_\mathbf{P}$, $\mathsf{t}_\mathbf{P}(x - x_{\mathbf{P},\lambda}^*) = \mathsf{t}(x - x_{\mathbf{P},\lambda}^*) \leq \log 2$. Thus, if

$$(1+\mathsf{C}_\mathbf{P}(x_\lambda^\star,\lambda))\gamma_\lambda \leq \frac{3}{10}, \quad \nu_{\mathbf{P},\lambda}(x) \leq \frac{\mathsf{r}_{\mathbf{P},\lambda}(x)}{2},$$

then it holds

$$\mathsf{t}(x - x_\lambda^\star) \leq \frac{3}{10} + 2\log 2.$$

**4) A technical result to bound** $\|\mathbf{H}_\lambda(x)^{-1/2}\mathbf{H}_{\mathbf{P},\lambda}(x)^{1/2}\|$. Using the fact that $\mathbf{P}x = x$, and Lemma 23, applied to $\mathbf{A} = \mathbf{H}(x)$, we get

$$\|\mathbf{H}_\lambda(x)^{-1/2}\mathbf{H}_{\mathbf{P},\lambda}(x)^{1/2}\| \leq 1 + \sqrt{\mathsf{C}_\mathbf{P}(x,\lambda)}.$$

Then, one can easily bound $\mathsf{C}_\mathbf{P}(x,\lambda) \leq e^{\mathsf{t}(x-x_\lambda^\star)}\mathsf{C}_\mathbf{P}(x_\lambda^\star,\lambda)$.

**5) Let us now bound** $\nu_\lambda(x)$**.** First, decompose the term

$$\nu_\lambda(x) = \|\nabla f_\lambda(x)\|_{\mathbf{H}_\lambda^{-1}(x)} \leq \|\mathbf{P}\nabla f_\lambda(x)\|_{\mathbf{H}_\lambda^{-1}(x)} + \|(\mathbf{I}-\mathbf{P})\nabla f(x)\|_{\mathbf{H}_\lambda^{-1}(x)}.$$

Since $x \in \mathcal{H}_\mathbf{P}$, $\|\mathbf{P}\nabla f_\lambda(x)\|_{\mathbf{H}_\lambda^{-1}(x)} = \|\nabla f_{\mathbf{P},\lambda}(x)\|_{\mathbf{H}_\lambda^{-1}(x)}$, and using the previous point, we get

$$\|\mathbf{P}\nabla f_\lambda(x)\|_{\mathbf{H}_\lambda^{-1}(x)} \leq \left(1 + e^{\mathsf{t}(x-x_\lambda^\star)/2}\sqrt{\mathsf{C}_\mathbf{P}(x_\lambda^\star,\lambda)}\right)\nu_{\mathbf{P},\lambda}(x).$$

Let us now bound the second term. We divide it into two terms:

$$\|(\mathbf{I}-\mathbf{P})\nabla f(x)\|_{\mathbf{H}_\lambda^{-1}(x)} \leq \|(\mathbf{I}-\mathbf{P})\left(\nabla f(x) - \nabla f(x_\lambda^\star)\right)\|_{\mathbf{H}_\lambda^{-1}(x)} + \|(\mathbf{I}-\mathbf{P})\nabla f(x_\lambda^\star)\|_{\mathbf{H}_\lambda^{-1}(x)}.$$

The second term can be bounded in the following way:

$$\|(\mathbf{I}-\mathbf{P})\nabla f(x_\lambda^\star)\|_{\mathbf{H}_\lambda^{-1}(x)} \leq \frac{1}{\sqrt{\lambda}}\|(\mathbf{I}-\mathbf{P})\mathbf{H}_\lambda^{1/2}(x_\lambda^\star)\|\,\|\nabla f(x_\lambda^\star)\|_{\mathbf{H}_\lambda^{-1}(x_\lambda^\star)} \leq \sqrt{1+\mathsf{C}_\mathbf{P}(x_\lambda^\star,\lambda)}\,\mathsf{s}_\lambda.$$

For the first term, we proceed in the following way.

$$\begin{aligned}
\|(\mathbf{I}-\mathbf{P})\left(\nabla f(x) - \nabla f(x_\lambda^\star)\right)\|_{\mathbf{H}_\lambda^{-1}(x)} &= \|\int_0^1 \mathbf{H}_\lambda^{-1/2}(x)(\mathbf{I}-\mathbf{P})\mathbf{H}(x_t)(x - x_\lambda^\star)\,dt\| \\
&\leq \frac{1}{\sqrt{\lambda}}\int_0^1 \|(\mathbf{I}-\mathbf{P})\mathbf{H}^{1/2}(x_t)\|\,\|\mathbf{H}^{1/2}(x_t)(x - x_\lambda^\star)\|\,dt \\
&\leq \sqrt{\mathsf{C}_\mathbf{P}(x_\lambda^\star,\lambda)}\,\overline{\phi}(\mathsf{t}(x - x_\lambda^\star))\,\|x - x_\lambda^\star\|_{\mathbf{H}(x_\lambda^\star)} \\
&\leq \sqrt{\mathsf{C}_\mathbf{P}(x_\lambda^\star,\lambda)}\,e^{\mathsf{t}(x-x_\lambda^\star)}\nu_\lambda(x).
\end{aligned}$$

Hence the final bound:

$$\left(1 - \sqrt{C_{\mathbf{P}}(x_\lambda^\star, \lambda)}\, e^{\mathsf{t}(x - x_\lambda^\star)}\right) \nu_\lambda(x) \leq \left(1 + e^{\mathsf{t}(x - x_\lambda^\star)/2} \sqrt{C_{\mathbf{P}}(x_\lambda^\star, \lambda)}\right) \nu_{\mathbf{P}, \lambda}(x) + \sqrt{1 + C_{\mathbf{P}}(x_\lambda^\star, \lambda)}\, \mathsf{s}_\lambda.$$

Now if $C_{\mathbf{P}}(x_\lambda^\star, \lambda) \leq \frac{1}{120}$, we see that $\sqrt{C_{\mathbf{P}}(x_\lambda^\star, \lambda)}\, e^{\mathsf{t}(x - x_\lambda^\star)} \leq \frac{1}{2}$, and hence, using the bound on $\mathsf{t}(x - x_\lambda^\star)$,

$$\nu_\lambda(x) \leq 3(\nu_{\mathbf{P}, \lambda}(x) + \mathsf{s}_\lambda).$$

**6) Showing the last two points**  . We leverage the fact that $\nu_\lambda(x) \leq 3(\nu_{\mathbf{P}, \lambda}(x) + \mathsf{s}_\lambda)$ and $\mathsf{t}(x - x_\lambda^\star) \leq \frac{3}{10} + 2\log 2$.
To show the first bound, we plug in the previous results in the following equation:

$$\|x - x_\lambda^\star\| \leq \lambda^{-1/2} \|x - x_\lambda^\star\|_{\mathbf{H}_\lambda(x)} \leq \frac{1}{\underline{\phi}(\mathsf{t}(x - x_\lambda^\star))}\, \lambda^{-1/2} \nu_\lambda(x).$$

The last inequality is obtained using Eq. (18).

To show the second point, we use the fact that $x \in \mathcal{H}_{\mathbf{P}}$ to show that

$$\lambda \|x\|_{\mathbf{H}_{\mathbf{P}, \lambda}^{-1}(x)} \leq \lambda \|x\|_{\mathbf{H}_\lambda^{-1}(x)} \leq \lambda \|x - x_\lambda^\star\|_{\mathbf{H}_\lambda(x)} + \lambda \|x_\lambda^\star\|_{\mathbf{H}_\lambda^{-1}(x)}.$$

Then applying Eq. (17) and Eq. (18):

$$\lambda \|x\|_{\mathbf{H}_{\mathbf{P}, \lambda}^{-1}(x)} \leq \frac{1}{\underline{\phi}(\mathsf{t}(x - x_\lambda^\star))}\, \nu_\lambda(x) + e^{\mathsf{t}(x - x_\lambda^\star)/2} \mathsf{s}_\lambda.$$

We then use the previous results to conclude.  $\square$

### G.3   Finding a good projector

**Lemma 16.** *If for a certain $\eta \leq \lambda$ and for a certain constant $C$, $\|\mathbf{H}_\eta^{1/2}(x)(\mathbf{I} - \mathbf{P})\|^2 \leq C\eta$, then*

$$C_{\mathbf{P}}(x, \lambda) \leq \frac{C\eta}{\lambda}.$$

*Proof.* This is completely direct, using the fact that $\mathbf{H}^{1/2}(x) \preceq \mathbf{H}_\eta^{1/2}(x)$.  $\square$

# H   Relations between statistical problems and empirical problem.

In this section, we recall and reformulate the framework from [23].

## H.1   Statistical problem and ERM estimator

Let $\mathcal{Z}$ be a Polish space and $Z$ be a random variable on $\mathcal{Z}$ with distribution $\rho$. Let $\mathcal{H}$ be a separable Hilbert space, with norm $\|\cdot\|$, and let $(f_z)_{z\in\mathcal{Z}}$ be a family of functions on $\mathcal{H}$. Our goal is to minimize the *expected risk* with respect to $x \in \mathcal{H}$:

$$\inf_{x\in\mathcal{H}} \ f(x) := \mathbb{E}\left[f_Z(x)\right].$$

Given $(z_i)_{i=1}^n \in \mathcal{Z}^n$, we define the *empirical risk*:

$$\widehat{f}(x) := \frac{1}{n}\sum_{i=1}^n f_{z_i}(x),$$

and consider the following estimator based on regularized empirical risk minimization given $\lambda > 0$ (note that the minimizer is unique in this case):

$$\widehat{x}_\lambda^\star = \arg\min_{x\in\mathcal{H}} \widehat{f}_\lambda(x) := \widehat{f}(x) + \frac{\lambda}{2}\|x\|^2,$$

where we assume the following.

**Assumption 6** (i.i.d. data). *The samples $(z_i)_{1\le i\le n}$ are independently and identically distributed according to $\rho$.*

We make the following assumption on the family $(f_z)$ (this is a reformulation of Assumption 8 in [23])

**Assumption 7** (Generalized self-concordance). *For any $z \in \mathcal{Z}$, there exists an associated subset $\mathcal{G}_z \subset \mathcal{H}$ such that $(f_z, \mathcal{G}_z)$ is generalized self-concordant in the sense of Definition 3.*

Moreover we require the following technical assumption to guarantee that $f$ and and its derivatives are well defined for any $x \in \mathcal{H}$ (this is a reformulation of Assumptions 3 and 4 in [23], and the necessary conditions to obtain Proposition 3).

**Assumption 8** (Technical assumptions). *The mapping $(z,x) \in \mathcal{Z} \times \mathcal{H} \mapsto f_z(x)$ is measurable. Moreover,*

- *the random variables $\|f_Z(0)\|, \|\nabla f_Z(0)\|, \mathrm{Tr}(\nabla^2 f_Z(0))$ are are bounded;*

- *$\mathcal{G} := \bigcup_{z\in\mathrm{supp}(Z)} \mathcal{G}_z$ is a bounded subset of $\mathcal{H}$.*

The assumptions above are usually easy to check in practice. In particular, if the support of $\rho$ is bounded, the mappings $z \mapsto \ell_z(0), \nabla\ell_z(0), \mathrm{Tr}(\nabla^2\ell_z(0))$ are continuous, and $z \mapsto \mathcal{G}_z$ is uniformly bounded on bounded sets, then they hold.

**Proposition 16.** *Under Assumptions 7 and 8, the function $(f, \mathcal{G})$ (or simply $f$) is generalized self-concordant.*

*Moreover, under Assumption 6, define*

$$\widehat{\mathcal{G}} := \bigcup_{i=1}^n \mathcal{G}_{z_i}.$$

*Then $(\widehat{f}, \widehat{\mathcal{G}})$ (or simply $\widehat{f}$) is generalized self-concordant. Moreover, note that $\widehat{\mathcal{G}} \subset \mathcal{G}$.*

The main regularity assumption we make on our statistical problems follows (see Assumption 5 in [23]).

**Assumption 9** (Existence of a minimizer). *There exists $x^\star \in \mathcal{H}$ such that $f(x^\star) = \inf_{x\in\mathcal{H}} f(x)$.*

**Notations** We adopt all the notations from Appendix A for $f$ and $\widehat{f}$, which are generalized self-concordant functions with associated subsets given in Proposition 16 with the following conventions:

- For all quantities relating to $f$, we omit the subscript $f$ as usual;
- For all quantities relating to $\widehat{f}$, we omit the subscript $\widehat{f}$ and instead put a hat over all these quantities. For instance:

$$\widehat{\mathbf{H}}(x) := \mathbf{H}_{\widehat{f}}(x) = \frac{1}{n}\sum_{i=1}^{n}\nabla^2 f_{z_i}(x), \ \widehat{r}_\lambda(x) := r_{\widehat{f},\lambda}(x) = \frac{1}{\sup_{g\in\widehat{\mathcal{G}}}\|g\|_{\widehat{\mathbf{H}}_\lambda^{-1}(x)}}, \ \text{etc...}$$

Recall the two main quantities introduced in [23] to establish the quality of our estimator $\widehat{x}_\lambda^\star$ (in [23], this is a mix between Proposition 2 and Definition 3).

**Proposition 17** (Bias, degrees of freedom). *Suppose Assumptions 7 to 9 are satisfied. The following key quantities are well defined:*

- *the* bias $b_\lambda = \|\mathbf{H}_\lambda(x^\star)^{-1/2}\nabla f_\lambda(x^\star)\|$;

- *the* effective dimension $df_\lambda = \mathbb{E}\left[\|\mathbf{H}_\lambda(x^\star)^{-1/2}\nabla f_Z(x^\star)\|^2\right]$.

*Moreover, we also introduce the following quantities:*

$$B_1^\star := \sup_{z\in\mathrm{supp}(Z)}\|\nabla f_z(x^\star)\|, \qquad B_2^\star := \sup_{z\in\mathrm{supp}(Z)}\mathrm{Tr}(\nabla^2 f_z(x^\star)), \qquad Q^\star = \frac{B_1^\star}{\sqrt{B_2^\star}}.$$

We can now recall the main theorem of [23] (Theorem 4), which quantifies the behavior of the ERM estimator:

**Theorem 8** (Bound for the ERM estimator). *Let $n\in\mathbb{N}$, $\delta\in(0,1/2]$, $0<\lambda\leq B_2^\star$. Whenever*

$$n \geq \triangle_1\frac{B_2^\star}{\lambda}\log\frac{8\square_1^2 B_2^\star}{\lambda\delta}, \qquad \sqrt{\triangle_2\frac{df_\lambda\vee(Q^\star)^2}{n}\ \log\frac{2}{\delta}} \leq r_\lambda(x^\star), \qquad 2b_\lambda \leq r_\lambda(x^\star),$$

*then with probability at least $1-2\delta$, it holds*

$$f(\widehat{x}_\lambda^\star) - f(x^\star) \leq C_{\mathrm{bias}}\,b_\lambda^2 + C_{\mathrm{var}}\,\frac{df_\lambda\vee(Q^\star)^2}{n}\ \log\frac{2}{\delta}, \tag{37}$$

*where $C_{\mathrm{bias}}, C_{\mathrm{var}}, \square_1 \leq 414$, $\triangle_1$, $\triangle_2 \leq 5184$.*

## H.2  Link between a good approximation of $\widehat{x}_\lambda^\star$ and $x^\star$

In this paper, we provide an algorithm which can effectively compute a good approximation of $\widehat{x}_\lambda^\star$ (as it is a finite sum problem which can be solved). This algorithm will return a certain $x\in\mathcal{H}$, whose precision with respect to the empirical problem will be characterized by $\widehat{\nu}_\lambda(x)$. The aim of the following lemma is to see how this approximation $x$ behaves with respect to the statistical problem.

**Lemma 17.** *Suppose the conditions for Thm. 8 are satisfied, i.e. let $n\in\mathbb{N}$, $\delta\in(0,1/2]$, $0<\lambda\leq B_2^\star$ and suppose*

$$n \geq \triangle_1\frac{B_2^\star}{\lambda}\log\frac{8\square_1^2 B_2^\star}{\lambda\delta}, \qquad \sqrt{\triangle_2\frac{df_\lambda\vee(Q^\star)^2}{n}\ \log\frac{2}{\delta}} \leq r_\lambda(x^\star), \qquad 2b_\lambda \leq r_\lambda(x^\star).$$

*Let $x$ be an approximation of $\widehat{x}_\lambda^\star$ characterized by its Newton decrement $\widehat{\nu}_\lambda(x)$. If*

$$\widehat{\nu}_\lambda(x) \leq \frac{\widehat{r}_\lambda(x)}{2},\ \widehat{\nu}_\lambda(x) \leq \frac{r_\lambda(x^\star)}{2},$$

*then with probability at least $1-2\delta$, it holds*

$$f(x) - f(x^\star) \leq 14(f(\widehat{x}_\lambda^\star) - f(x^\star)) + 30\widehat{\nu}_\lambda(x)^2.$$

*Proof.* Using Eq. (16),

$$f(x) - f(\widehat{x}_\lambda^\star) \leq \langle\nabla f(\widehat{x}_\lambda^\star), x - \widehat{x}_\lambda^\star\rangle_{\mathcal{H}} + \psi(\mathsf{t}(x-\widehat{x}_\lambda^\star))\|x - \widehat{x}_\lambda^\star\|_{\mathbf{H}_\lambda(\widehat{x}_\lambda^\star)}^2$$

$$\leq \frac{1}{2}\|\nabla f(\widehat{x}_\lambda^\star)\|_{\mathbf{H}_\lambda^{-1}(\widehat{x}_\lambda^\star)}^2 + \left(\psi(\mathsf{t}(x-\widehat{x}_\lambda^\star)) + \frac{1}{2}\right)\|x - \widehat{x}_\lambda^\star\|_{\mathbf{H}_\lambda(\widehat{x}_\lambda^\star)}^2.$$

**1. Let us bound** $\|\nabla f(\widehat{x}_\lambda^\star)\|_{\mathbf{H}_\lambda^{-1}(\widehat{x}_\lambda^\star)}$

$$\|\nabla f(\widehat{x}_\lambda^\star)\|_{\mathbf{H}_\lambda^{-1}(\widehat{x}_\lambda^\star)} \leq \int_0^1 \|\mathbf{H}_\lambda^{-1/2}(\widehat{x}_\lambda^\star)\mathbf{H}(x_t)(\widehat{x}_\lambda^\star - x^\star)\| \, dt, \qquad\qquad x_t = (1-t)\widehat{x}_\lambda^\star + tx^\star$$

$$\leq \int_0^1 \|\mathbf{H}_\lambda^{-1/2}(\widehat{x}_\lambda^\star)\mathbf{H}^{1/2}(x_t)\| \, \|\mathbf{H}^{1/2}(x_t)(\widehat{x}_\lambda^\star - x^\star)\| \, dt.$$

Now using equation Eq. (14)

$$\mathbf{H}(x_t) \preceq e^{t\mathbf{t}(\widehat{x}_\lambda^\star - x^\star)}\mathbf{H}(\widehat{x}_\lambda^\star), \qquad \mathbf{H}(x_t) \preceq e^{(1-t)\mathbf{t}(\widehat{x}_\lambda^\star - x^\star)}.$$

Thus:

$$\|\nabla f(\widehat{x}_\lambda^\star)\|_{\mathbf{H}_\lambda^{-1}(\widehat{x}_\lambda^\star)} \leq e^{\mathbf{t}(\widehat{x}_\lambda^\star - x^\star)/2} \|\widehat{x}_\lambda^\star - x^\star\|_{\mathbf{H}(x^\star)}.$$

Finally, using equation Eq. (16)

$$\|\nabla f(\widehat{x}_\lambda^\star)\|_{\mathbf{H}_\lambda^{-1}(\widehat{x}_\lambda^\star)} \leq \frac{e^{\mathbf{t}(\widehat{x}_\lambda^\star - x^\star)/2}}{\psi(-\mathbf{t}(\widehat{x}_\lambda^\star - x^\star))^{1/2}} \left(f(\widehat{x}_\lambda^\star) - f(x^\star)\right)^{1/2}.$$

**2. Let us bound the terms involving** $\|x - \widehat{x}_\lambda^\star\|_{\mathbf{H}_\lambda(\widehat{x}_\lambda^\star)}$   Note that using Eq. (18) and Eq. (17) applied to $\widehat{f}$,

$$\|x - \widehat{x}_\lambda^\star\|_{\mathbf{H}_\lambda(\widehat{x}_\lambda^\star)} \leq \|\mathbf{H}_\lambda^{1/2}(\widehat{x}_\lambda^\star)\widehat{\mathbf{H}}_\lambda^{-1/2}(\widehat{x}_\lambda^\star)\| \frac{e^{\widehat{\mathbf{t}}(x - \widehat{x}_\lambda^\star)/2}}{\underline{\phi}(\widehat{\mathbf{t}}(x - \widehat{x}_\lambda^\star))} \widehat{\nu}_\lambda(x).$$

This also leads to:

$$\mathbf{t}(x - \widehat{x}_\lambda^\star) \leq \frac{1}{\mathsf{r}_\lambda(\widehat{x}_\lambda^\star)} \|\mathbf{H}_\lambda^{1/2}(\widehat{x}_\lambda^\star)\widehat{\mathbf{H}}_\lambda^{-1/2}(\widehat{x}_\lambda^\star)\| \, \|x - \widehat{x}_\lambda^\star\|_{\widehat{\mathbf{H}}_\lambda(\widehat{x}_\lambda^\star)}$$

$$\leq \frac{1}{\mathsf{r}_\lambda(\widehat{x}_\lambda^\star)} \|\mathbf{H}_\lambda^{1/2}(\widehat{x}_\lambda^\star)\widehat{\mathbf{H}}_\lambda^{-1/2}(\widehat{x}_\lambda^\star)\| \frac{e^{\widehat{\mathbf{t}}(x - \widehat{x}_\lambda^\star)/2}}{\underline{\phi}(\widehat{\mathbf{t}}(x - \widehat{x}_\lambda^\star))} \widehat{\nu}_\lambda(x).$$

**3. Putting things together**   In the end, we get

$$f(x) - f(x^\star) \leq \left(1 + \frac{e^{\mathbf{t}(\widehat{x}_\lambda^\star - x^\star)}}{\psi(-\mathbf{t}(\widehat{x}_\lambda^\star - x^\star))}\right)\left(f(\widehat{x}_\lambda^\star) - f(x^\star)\right)$$

$$+ \left(\psi(\mathbf{t}(x - \widehat{x}_\lambda^\star)) + \frac{1}{2}\right)\left(e^{\mathbf{t}(\widehat{x}_\lambda^\star - x^\star)/2}\|\mathbf{H}_\lambda^{1/2}(x_\lambda^\star)\widehat{\mathbf{H}}_\lambda^{-1/2}(x_\lambda^\star)\| \frac{e^{\widehat{\mathbf{t}}(x - \widehat{x}_\lambda^\star)/2}}{\underline{\phi}(\widehat{\mathbf{t}}(x - \widehat{x}_\lambda^\star))}\right)\widehat{\nu}_\lambda(x)^2.$$

Moreover, we bound

$$\mathbf{t}(x - \widehat{x}_\lambda^\star) \leq e^{(\mathbf{t}(x^\star - \widehat{x}_\lambda^\star) + \mathbf{t}(\widehat{x}_\lambda^\star - x^\star))/2} \|\mathbf{H}_\lambda^{1/2}(x_\lambda^\star)\widehat{\mathbf{H}}_\lambda^{-1/2}(x_\lambda^\star)\| \frac{e^{\widehat{\mathbf{t}}(x - \widehat{x}_\lambda^\star)/2}}{\underline{\phi}(\widehat{\mathbf{t}}(x - \widehat{x}_\lambda^\star))} \frac{\widehat{\nu}_\lambda(x)}{\mathsf{r}_\lambda(x^\star)}.$$

**4. Plugging in previous results**   Under the assumptions of this lemma, which include the assumptions of Theorem 4. in [23], we get the following bounds.

- In [23],the assumptions of Theorem 4 imply that we can use Lemma 9, which uses Lemma 8 in which we show that with probability at least $1 - \delta$,

$$\|\widehat{\mathbf{H}}_\lambda^{-1/2}(x_\lambda^\star)\mathbf{H}_\lambda(x_\lambda^\star)^{1/2}\|^2 \leq 2.$$

- Still using the assumptions of Theorem 4, we see in the proof of this theorem that the assumptions of Theorem 7 of [23] are satisfied in the case where $b_\lambda \leq \frac{r_\lambda(x^\star)}{2}$, and thus that

$$\mathsf{t}(\widehat{x}_\lambda^\star - x_\lambda^\star) \leq \log 2, \ \mathsf{t}(x_\lambda^\star - x^\star) \leq \log 2.$$

Plugging in all these bounds, we get

$$\left(1 + \frac{e^{\mathsf{t}(\widehat{x}_\lambda^\star - x^\star)}}{\psi(-\mathsf{t}(\widehat{x}_\lambda^\star - x^\star))}\right) \leq 14, \ \mathsf{t}(x - \widehat{x}_\lambda^\star) \leq 6,$$

$$\left(\psi(\mathsf{t}(x - \widehat{x}_\lambda^\star)) + \frac{1}{2}\right)\left(e^{\mathsf{t}(\widehat{x}_\lambda^\star - x_\lambda^\star)/2}\|\mathbf{H}_\lambda^{1/2}(x_\lambda^\star)\widehat{\mathbf{H}}_\lambda^{-1/2}(x_\lambda^\star)\|\frac{e^{\widehat{\mathsf{t}}(x - \widehat{x}_\lambda^\star)/2}}{\underline{\phi}(\widehat{\mathsf{t}}(x - \widehat{x}_\lambda^\star))}\right) \leq 30.$$

$\square$

### H.3 Bounds when we solve a projected empirical problem

In this section, we place ourselves in the setting of Appendix G. In this section, we had argued that for computational purposes, it was less costly to compute an approximate solution to a projected problem.

In this section, we assume that we are going to project the regularized empirical problem, that is solve approximately

$$x \approx \arg\min_{x \in \mathcal{H}} \widehat{f}_{\mathbf{P},\lambda}(x) = \widehat{f}(\mathbf{P}x) + \frac{\lambda}{2}\|x\|^2.$$

for a given orthogonal projection $\mathbf{P}$. Recall from Appendix G that there is a natural way of seeing $\widehat{f}_{\mathbf{P}}$ as a generalized self-concordant function. We import all the notations from this section, keeping a $\widehat{\cdot}$ over all notations to mark the fact that we are projecting $\widehat{f}$ and not $f$.

To quantify the quality of the approximation $x$, we will use the Newton decrement for the empirical projected problem $\widehat{\nu}_{\mathbf{P},\lambda}(x) := \nu_{\widehat{f}_{\mathbf{P},\lambda}}(x)$.

As we see in Proposition 15, under certain conditions, bounding $\widehat{\nu}_\lambda(x)$ amounts to bounding two terms:

- The empirical source $\widehat{s}_\lambda := \lambda\|\widehat{x}_\lambda^\star\|_{\widehat{\mathbf{H}}_\lambda^{-1}(\widehat{x}_\lambda^\star)}$,
- The projected empirical Newton decrement $\widehat{\nu}_{\mathbf{P},\lambda}(x)$.

**1. Bounding the empirical source term $\widehat{s}_\lambda$**  Start by bounding the source empirical source term $\widehat{s}_\lambda$ using quantities we know.

**Lemma 18** (Empirical source). *Let $n \in \mathbb{N}$, $\delta \in (0, 1/2]$, $0 < \lambda \leq \mathsf{B}_2^\star$. Whenever*

$$n \geq \triangle_1 \frac{\mathsf{B}_2^\star}{\lambda}\log\frac{8\square_1^2\mathsf{B}_2^\star}{\lambda\delta}, \qquad \sqrt{\triangle_2\frac{\mathsf{df}_\lambda \vee (\mathsf{Q}^\star)^2}{n}\ \log\frac{2}{\delta}} \leq \mathsf{r}_\lambda(x^\star), \qquad 2\mathsf{b}_\lambda \leq \mathsf{r}_\lambda(x^\star).$$

*The following holds, with probability at least $1 - 2\delta$.*

$$\widehat{s}_\lambda \leq 8\,\mathsf{b}_\lambda + 80\sqrt{\frac{\mathsf{df}_\lambda \vee (\mathsf{Q}^\star)^2\ \log\frac{2}{\delta}}{n}}.$$

*Moreover, we also have the following bound :*

$$\|\widehat{x}_\lambda^\star - x^\star\| \leq 3\,\lambda^{-1/2}\,\mathsf{b}_\lambda + 8\,\lambda^{-1/2}\sqrt{\frac{\mathsf{df}_\lambda \vee (\mathsf{Q}^\star)^2\ \log\frac{2}{\delta}}{n}}.$$

*Proof.* We first decompose the source term into two terms, and then apply different bounds from [23] to effectively bound it. We will use the following quantity:

$$\widehat{v}_\lambda := \|\mathbf{H}_\lambda^{1/2}(x_\lambda^\star)\widehat{\mathbf{H}}_\lambda^{-1/2}(x_\lambda^\star)\|^2\,\|\nabla\widehat{f}_\lambda(x_\lambda^\star)\|_{\mathbf{H}_\lambda^{-1}(x_\lambda^\star)}.$$

It is also defined in equation (23) in [23].

**1. Dividing $\widehat{s}_\lambda$ into two controllable terms** . Decompose

$$\widehat{s}_\lambda = \|\lambda\widehat{x}_\lambda^\star\|_{\widehat{\mathbf{H}}_\lambda^{-1}(\widehat{x}_\lambda^\star)} \leq \|\widehat{\mathbf{H}}_\lambda^{-1/2}(\widehat{x}_\lambda^\star)\mathbf{H}_\lambda^{1/2}(\widehat{x}_\lambda^\star)\|\,\|\lambda\widehat{x}_\lambda^\star\|_{\mathbf{H}_\lambda^{-1}(\widehat{x}_\lambda^\star)}$$
$$\leq \|\widehat{\mathbf{H}}_\lambda^{-1/2}(\widehat{x}_\lambda^\star)\mathbf{H}_\lambda^{1/2}(\widehat{x}_\lambda^\star)\|\,\left(\|\nabla f_\lambda(\widehat{x}_\lambda^\star)\|_{\mathbf{H}_\lambda^{-1}(\widehat{x}_\lambda^\star)} + \|\nabla f(\widehat{x}_\lambda^\star)\|_{\mathbf{H}_\lambda^{-1}(\widehat{x}_\lambda^\star)}\right).$$

On the one hand, from the previous proof, we get

$$\|\nabla f(\widehat{x}_\lambda^\star)\|_{\mathbf{H}_\lambda^{-1}(\widehat{x}_\lambda^\star)} \leq e^{\mathsf{t}(\widehat{x}_\lambda^\star - x^\star)/2}\,\|\widehat{x}_\lambda^\star - x^\star\|_{\mathbf{H}(x^\star)}$$
$$\leq e^{\mathsf{t}(\widehat{x}_\lambda^\star - x^\star)/2}\left(e^{\mathsf{t}(x_\lambda^\star - x^\star)}\|\widehat{x}_\lambda^\star - x_\lambda^\star\|_{\mathbf{H}_\lambda(x_\lambda^\star)} + \|x_\lambda^\star - x^\star\|_{\mathbf{H}_\lambda(x^\star)}\right)$$
$$\leq e^{\mathsf{t}(\widehat{x}_\lambda^\star - x^\star)/2}\left(\frac{e^{\mathsf{t}(x_\lambda^\star - x^\star)}}{\underline{\phi}(\mathsf{t}(\widehat{x}_\lambda^\star - x_\lambda^\star))}\widehat{v}_\lambda + \frac{1}{\underline{\phi}(\mathsf{t}(x_\lambda^\star - x^\star))}b_\lambda\right).$$

In the last line, we use the fact that $\|\widehat{x}_\lambda^\star - x_\lambda^\star\|_{\mathbf{H}_\lambda(x_\lambda^\star)} \leq \|\mathbf{H}_\lambda^{1/2}(x_\lambda^\star)\widehat{\mathbf{H}}_\lambda^{-1/2}(x_\lambda^\star)\|\,\|\widehat{x}_\lambda^\star - x_\lambda^\star\|_{\widehat{\mathbf{H}}_\lambda(x_\lambda^\star)}$ and then bound it using Eq. (18) applied to $\widehat{f}$ to get

$$\|\widehat{x}_\lambda^\star - x_\lambda^\star\|_{\widehat{\mathbf{H}}_\lambda(x_\lambda^\star)} \leq \frac{1}{\underline{\phi}(\widehat{\mathsf{t}}(x_\lambda^\star - \widehat{x}_\lambda^\star))}\|\nabla\widehat{f}_\lambda(x_\lambda^\star)\|_{\widehat{\mathbf{H}}_\lambda^{-1}(x_\lambda^\star)}$$
$$\leq \frac{1}{\underline{\phi}(\mathsf{t}(x_\lambda^\star - \widehat{x}_\lambda^\star))}\|\mathbf{H}_\lambda^{1/2}(x_\lambda^\star)\widehat{\mathbf{H}}_\lambda^{-1/2}(x_\lambda^\star)\|\,\|\nabla\widehat{f}_\lambda(x_\lambda^\star)\|_{\mathbf{H}_\lambda^{-1}(x_\lambda^\star)}.$$

On the other hand, apply successively Eq. (18) to $f$ and $\widehat{f}$ using the fact that $\widehat{\mathsf{t}} \leq \mathsf{t}$ to get

$$\|\nabla f_\lambda(\widehat{x}_\lambda^\star)\|_{\mathbf{H}_\lambda^{-1}(\widehat{x}_\lambda^\star)} = \|\nabla f_\lambda(\widehat{x}_\lambda^\star) - \nabla f_\lambda(x_\lambda^\star)\|_{\mathbf{H}_\lambda^{-1}(\widehat{x}_\lambda^\star)}$$
$$\leq e^{\mathsf{t}(\widehat{x}_\lambda^\star - x_\lambda^\star)/2}\overline{\phi}(\mathsf{t}(\widehat{x}_\lambda^\star - x_\lambda^\star))\,\|\widehat{x}_\lambda^\star - x_\lambda^\star\|_{\mathbf{H}_\lambda(x_\lambda^\star)}$$
$$\leq e^{\mathsf{t}(\widehat{x}_\lambda^\star - x_\lambda^\star)/2}\overline{\phi}(\mathsf{t}(\widehat{x}_\lambda^\star - x_\lambda^\star))\,\|\mathbf{H}_\lambda^{1/2}(x_\lambda^\star)\widehat{\mathbf{H}}_\lambda^{-1/2}(x_\lambda^\star)\|\,\|\widehat{x}_\lambda^\star - x_\lambda^\star\|_{\widehat{\mathbf{H}}_\lambda(x_\lambda^\star)}$$
$$\leq \frac{e^{\mathsf{t}(\widehat{x}_\lambda^\star - x_\lambda^\star)/2}\overline{\phi}(\mathsf{t}(\widehat{x}_\lambda^\star - x_\lambda^\star))}{\underline{\phi}(\mathsf{t}(\widehat{x}_\lambda^\star - x_\lambda^\star))}\,\|\mathbf{H}_\lambda^{1/2}(x_\lambda^\star)\widehat{\mathbf{H}}_\lambda^{-1/2}(x_\lambda^\star)\|^2\,\|\nabla\widehat{f}_\lambda(x_\lambda^\star)\|_{\mathbf{H}_\lambda(x_\lambda^\star)}$$
$$= e^{3\mathsf{t}(\widehat{x}_\lambda^\star - x_\lambda^\star)/2}\widehat{v}_\lambda.$$

Putting things together:

$$\widehat{s}_\lambda \leq \|\widehat{\mathbf{H}}_\lambda^{-1/2}(\widehat{x}_\lambda^\star)\mathbf{H}_\lambda^{1/2}(\widehat{x}_\lambda^\star)\|\,\left(e^{3\mathsf{t}(x_\lambda^\star - \widehat{x}_\lambda^\star)/2}\left(1 + \frac{1}{\underline{\phi}(\mathsf{t}(x_\lambda^\star - \widehat{x}_\lambda^\star))}\right)\widehat{v}_\lambda + \frac{e^{\mathsf{t}(x_\lambda^\star - \widehat{x}_\lambda^\star)/2}}{\underline{\phi}(\mathsf{t}(x_\lambda^\star - x^\star))}b_\lambda\right).$$

**2. We now import the results from [23]** . The following hypotheses imply those of Thms 4 and 7 in [23]:

Let $n \in \mathbb{N}$, $\delta \in (0, 1/2]$, $0 < \lambda \leq \mathsf{B}_2^\star$. Whenever

$$n \geq \triangle_1 \frac{\mathsf{B}_2^\star}{\lambda}\log\frac{8\square_1^2\mathsf{B}_2^\star}{\lambda\delta},\quad n \geq \triangle_2 \frac{\mathsf{df}_\lambda \vee (\mathsf{Q}^\star)^2}{\mathsf{r}_\lambda(x^\star)^2}\log\frac{2}{\delta}, \mathsf{b}_\lambda \leq \frac{\mathsf{r}_\lambda(x^\star)}{2}.$$

In particular, they imply that with probability at least $1 - 2\delta$:

- $\widehat{\mathsf{v}}_\lambda \le \frac{1}{2}\mathsf{b}_\lambda + 4\square_1 \sqrt{\frac{\mathsf{df}_\lambda \vee (\mathsf{Q}^\star)^2 \ \log \frac{2}{\delta}}{n}}$;

- $\|\mathbf{H}_\lambda^{1/2}(x_\lambda^\star)\widehat{\mathbf{H}}_\lambda^{-1/2}(x_\lambda^\star)\| \le \sqrt{2}$;

- $\mathsf{t}(x^\star - x_\lambda^\star) \le \log 2$;

- $\mathsf{t}(\widehat{x}_\lambda^\star - x_\lambda^\star) \le \log 2$.

Hence, plugging these bounds in the previous equation, we get

$$\widehat{s}_\lambda \le 8\mathsf{b}_\lambda + 80\sqrt{\frac{\mathsf{df}_\lambda \vee (\mathsf{Q}^\star)^2 \ \log \frac{2}{\delta}}{n}}.$$

**3.**   Note that in what has been done previously, we can bound:

$$\|\widehat{x}_\lambda^\star - x_\lambda^\star\|_{\mathbf{H}_\lambda(x_\lambda^\star)} \le \frac{1}{\underline{\phi}(\mathsf{t}(x_\lambda^\star - \widehat{x}_\lambda^\star))}\widehat{\mathsf{v}}_\lambda \le \mathsf{b}_\lambda + 8\sqrt{\frac{\mathsf{df}_\lambda \vee (\mathsf{Q}^\star)^2 \ \log \frac{2}{\delta}}{n}}.$$

Moreover,

$$\|x_\lambda^\star - x^\star\|_{\mathbf{H}_\lambda(x^\star)} \le \frac{1}{\underline{\phi}(\mathsf{t}(x_\lambda^\star - x^\star))}\|\nabla f_\lambda(x^\star)\|_{\mathbf{H}_\lambda^{-1}(x^\star)} \le 2\mathsf{b}_\lambda.$$

Hence:

$$\|\widehat{x}_\lambda^\star - x^\star\| \le 3\,\lambda^{-1/2}\,\mathsf{b}_\lambda + 8\,\lambda^{-1/2}\sqrt{\frac{\mathsf{df}_\lambda \vee (\mathsf{Q}^\star)^2 \ \log \frac{2}{\delta}}{n}}.$$

$\square$

**2. Final bound for the projected ERM approximation**   In this paragraph, denote with $\mathsf{C}_\mathbf{P}(x, \lambda)$ the quantity $\frac{\|\mathbf{H}^{1/2}(x)(\mathbf{I}-\mathbf{P})\|^2}{\lambda}$ and $\widehat{\mathsf{C}}_\mathbf{P}(x, \lambda)$ the quantity $\frac{\|\widehat{\mathbf{H}}^{1/2}(x)(\mathbf{I}-\mathbf{P})\|^2}{\lambda}$

**Lemma 19.** *Let* $n \in \mathbb{N}$, $\delta \in (0, 1/2]$, $0 < \lambda \le \mathsf{B}_2^\star$. *Whenever*

$$n \ge \triangle_1 \frac{\mathsf{B}_2^\star}{\lambda} \log \frac{8\square_1^2 \mathsf{B}_2^\star}{\lambda\delta}, \qquad \mathsf{C}_1\sqrt{\frac{\mathsf{df}_\lambda \vee (\mathsf{Q}^\star)^2}{n} \ \log \frac{2}{\delta}} \le \mathsf{r}_\lambda(x^\star), \qquad \mathsf{C}_1\mathsf{b}_\lambda \le \mathsf{r}_\lambda(x^\star),$$

*if*

$$\mathsf{C}_\mathbf{P}(x^\star, \lambda) \le \frac{\sqrt{2}}{480}, \ \widehat{\nu}_{\mathbf{P},\lambda}(x) \le \frac{\widehat{\mathsf{r}}_{\mathbf{P},\lambda}(x)}{2} \ \wedge \ \frac{\mathsf{r}_\lambda(x^\star)}{126},$$

*the following holds, with probability at least* $1 - 2\delta$.

$$\widehat{\nu}_\lambda(x) \le \frac{\widehat{\mathsf{r}}_\lambda(x)}{2}, \ \widehat{\nu}_\lambda(x) \le \frac{\mathsf{r}_\lambda(x^\star)}{2}.$$

*Here,* $\mathsf{C}_1 = 1008$.

*Proof.* Proceed in the following way.

**1.**   It is easy to see that the conditions of this lemma imply the conditions of Thm. 8. Hence, as in the previous proofs, the following hold:

- $\|\mathbf{H}_\lambda^{1/2}(x_\lambda^\star)\widehat{\mathbf{H}}_\lambda^{-1/2}(x_\lambda^\star)\| \le \sqrt{2}$;

- $\mathsf{t}(x^\star - x_\lambda^\star) \le \log 2$;

- $\mathsf{t}(\widehat{x}_\lambda^\star - x_\lambda^\star) \le \log 2$.

**2.** Let us now apply Proposition 15 to $\widehat{f}$. If

$$\frac{\widehat{s}_\lambda}{\widehat{r}_\lambda(\widehat{x}_\lambda^\star)} \leq \frac{1}{4}, \quad \widehat{\mathsf{C}}_{\mathbf{P}}(\widehat{x}_\lambda^\star, \lambda) \leq \frac{1}{120}, \quad \widehat{\nu}_{\mathbf{P},\lambda}(x) \leq \frac{\widehat{r}_{\mathbf{P},\lambda}(x)}{2},$$

Then it holds:

$$\widehat{\nu}_\lambda(x) \leq 3(\widehat{\nu}_{\mathbf{P},\lambda}(x) + \widehat{s}_\lambda), \qquad \widehat{\mathsf{t}}(x - \widehat{x}_\lambda^\star) \leq \frac{3}{10} + 2\log 2. \tag{38}$$

where the second bound is obtained in the proof of this proposition. Now since

$$\frac{1}{\widehat{r}_\lambda(\widehat{x}_\lambda^\star)} \leq e^{\widehat{\mathsf{t}}(\widehat{x}_\lambda^\star - x_\lambda^\star)/2} \frac{1}{\widehat{r}_\lambda(x_\lambda^\star)} \qquad\qquad Eq.\ (17)$$

$$\leq e^{\widehat{\mathsf{t}}(\widehat{x}_\lambda^\star - x_\lambda^\star)/2} \|\mathbf{H}_\lambda^{1/2}(x_\lambda^\star)\widehat{\mathbf{H}}_\lambda^{-1/2}(x_\lambda^\star)\| \sup_{g \in \widehat{\mathcal{G}}} \|g\|_{\mathbf{H}_\lambda^{-1}(x_\lambda^\star)} \qquad\qquad \text{Def}$$

$$\leq e^{\mathsf{t}(\widehat{x}_\lambda^\star - x_\lambda^\star)/2} \|\mathbf{H}_\lambda^{1/2}(x_\lambda^\star)\widehat{\mathbf{H}}_\lambda^{-1/2}(x_\lambda^\star)\| \sup_{g \in \mathcal{G}} \|g\|_{\mathbf{H}_\lambda^{-1}(x_\lambda^\star)} \qquad\qquad \widehat{\mathcal{G}} \subset \mathcal{G}$$

$$= e^{\mathsf{t}(\widehat{x}_\lambda^\star - x_\lambda^\star)/2} \|\mathbf{H}_\lambda^{1/2}(x_\lambda^\star)\widehat{\mathbf{H}}_\lambda^{-1/2}(x_\lambda^\star)\| \frac{1}{r_\lambda(x_\lambda^\star)} \qquad\qquad \text{Def}$$

$$\leq e^{(\mathsf{t}(\widehat{x}_\lambda^\star - x_\lambda^\star) + \mathsf{t}(x_\lambda^\star - x^\star))/2} \|\mathbf{H}_\lambda^{1/2}(x_\lambda^\star)\widehat{\mathbf{H}}_\lambda^{-1/2}(x_\lambda^\star)\| \frac{1}{r_\lambda(x^\star)} \qquad\qquad Eq.\ (17)$$

$$\leq \frac{2\sqrt{2}}{r_\lambda(x^\star)}. \qquad\qquad \text{previous bounds}$$

In a similar way, we get $\widehat{\mathsf{C}}_{\mathbf{P}}(\widehat{x}_\lambda^\star, \lambda) \leq 2\sqrt{2}\mathsf{C}_{\mathbf{P}}(x^\star, \lambda)$. Thus, the conditions above are satisfied if the following conditions are satisfied:

$$\frac{\widehat{s}_\lambda}{r_\lambda(x^\star)} \leq \frac{\sqrt{2}}{16}, \quad \mathsf{C}_{\mathbf{P}}(x^\star, \lambda) \leq \frac{\sqrt{2}}{480}, \quad \widehat{\nu}_{\mathbf{P},\lambda}(x) \leq \frac{\widehat{r}_{\mathbf{P},\lambda}(x)}{2}.$$

Finally, note that under these conditions,

$$\frac{1}{\widehat{r}_\lambda(x)} \leq \frac{e^{\widehat{\mathsf{t}}(x - \widehat{x}_\lambda^\star)/2}}{\widehat{r}_\lambda(x)} \leq \frac{7}{r_\lambda(x^\star)}. \tag{39}$$

using the previous bound and the bound on $\widehat{\mathsf{t}}(x - \widehat{x}_\lambda^\star)$.

**3.** Let us assume

$$\frac{\widehat{s}_\lambda}{r_\lambda(x^\star)} \leq \frac{\sqrt{2}}{16}, \quad \mathsf{C}_{\mathbf{P}}(x^\star, \lambda) \leq \frac{\sqrt{2}}{480}, \quad \widehat{\nu}_{\mathbf{P},\lambda}(x) \leq \frac{\widehat{r}_{\mathbf{P},\lambda}(x)}{2}.$$

According to Eq. (39), and to Eq. (38), if

$$\widehat{\nu}_{\mathbf{P},\lambda}(x) + \widehat{s}_\lambda \leq \frac{r_\lambda(x^\star)}{42},$$

then it holds

$$\widehat{\nu}_\lambda(x) \leq \frac{\widehat{r}_\lambda(x)}{2}, \quad \widehat{\nu}_\lambda(x) \leq \frac{r_\lambda(x^\star)}{2}.$$

We simplify this condition as:

$$\widehat{\nu}_{\mathbf{P},\lambda}(x) \leq \frac{r_\lambda(x^\star)}{126}, \qquad \widehat{s}_\lambda \leq \frac{2r_\lambda(x^\star)}{126}.$$

**4.** Now using the fact that under the conditions of this lemma, those of Lemma 18 are satisfied:

$$\widehat{s}_\lambda \leq 8\,\mathsf{b}_\lambda + 80\sqrt{\frac{\mathsf{df}_\lambda \vee (\mathsf{Q}^\star)^2 \,\log\frac{2}{\delta}}{n}}.$$

Thus, $\widehat{s}_\lambda \leq \frac{2r_\lambda(x^\star)}{126}$ holds, provided

$$\mathsf{b}_\lambda \leq \frac{r_\lambda(x^\star)}{\mathsf{C}_1}, \quad n \geq \mathsf{C}_1^2 \frac{\mathsf{df}_\lambda \vee (\mathsf{Q}^\star)^2 \,\log\frac{2}{\delta}}{r_\lambda(x^\star)^2},$$

where $\mathsf{C}_1 = 1008$. $\qquad\qquad\qquad\qquad\qquad\qquad\qquad\qquad\qquad\qquad\qquad\qquad\qquad\qquad\square$

**Proposition 18** (Behavior of an approximation to the projected problem). *Let* $n \in \mathbb{N}$, $\delta \in (0, 1/2]$, $0 < \lambda \leq \mathsf{B}_2^\star$. *Let* $x \in \mathcal{H}_\mathbf{P}$. *Whenever*

$$n \geq \triangle_1 \frac{\mathsf{B}_2^\star}{\lambda} \log \frac{8\square_1^2 \mathsf{B}_2^\star}{\lambda \delta}, \qquad \mathsf{C}_1 \sqrt{\frac{\mathsf{df}_\lambda \vee (\mathsf{Q}^\star)^2}{n} \, \log \frac{2}{\delta}} \leq \mathsf{r}_\lambda(x^\star), \qquad \mathsf{C}_1 \mathsf{b}_\lambda \leq \mathsf{r}_\lambda(x^\star),$$

*if*

$$\mathsf{C}_\mathbf{P}(x^\star, \lambda) \leq \frac{\sqrt{2}}{480}, \; \widehat{\nu}_{\mathbf{P},\lambda}(x) \leq \frac{\widehat{\mathsf{r}}_{\mathbf{P},\lambda}(x)}{2} \wedge \frac{\mathsf{r}_\lambda(x^\star)}{126}.$$

*The following holds, with probability at least* $1 - 2\delta$.

$$f(x) - f(x^\star) \leq \mathsf{K}_1 \, \mathsf{b}_\lambda^2 + \mathsf{K}_2 \, \frac{\mathsf{df}_\lambda \vee (\mathsf{Q}^\star)^2}{n} \, \log \frac{2}{\delta} + \mathsf{K}_3 \, \widehat{\nu}_{\mathbf{P},\lambda}^2(x),$$

*where* $\mathsf{K}_1 \leq 6.0\mathrm{e}4$, $\mathsf{K}_2 \leq 6.0\mathrm{e}6$ *and* $\mathsf{K}_3 \leq 810$, $\mathsf{C}_1$ *are defined in Lemma 19, and the other constants are defined in Thm. 8.*

**Remark 5** (Constants). *In this result, absolutely huge constants are obtained. They are (of course) totally sub-optimal. Indeed, this analysis has been simplified by dividing the bound into blocks: error of the empirical risk minimization with regularization, error of the projection compared to this empirical risk minimizer. Going back and forth from empirical to statistical, from projected to non projected induces exponential explosion of the constants. There is a way of doing the analysis directly by projecting the statistical problem. However, in order to relate to our previous work [23] and avoid re-doing all of our work we discarded this. If we were to perform this more direct analysis, we could keep the constants to a reasonable level, of order $10^2$.*

*Proof.* We apply Lemma 17, using the previous lemma to guarantee the conditions.

**1.** Under the conditions of this proposition, applying Lemma 19, the conditions of Lemma 17 are satisfied. Thus,
$$f(x) - f(x^\star) \leq 14(f(\widehat{x}_\lambda^\star) - f(x^\star)) + 30\widehat{\nu}_\lambda(x)^2.$$
Moreover, from the previous proof,
$$\widehat{\nu}_\lambda(x) \leq 3(\widehat{\nu}_{\mathbf{P},\lambda}(x) + \widehat{s}_\lambda),$$
and seeing as Lemma 18 is satisfied,

$$\widehat{s}_\lambda \leq 8\, \mathsf{b}_\lambda + 80 \sqrt{\frac{\mathsf{df}_\lambda \vee (\mathsf{Q}^\star)^2 \, \log \frac{2}{\delta}}{n}}.$$

This therefore yields:

$$\widehat{\nu}_\lambda(x)^2 \leq 27\widehat{\nu}_{\mathbf{P},\lambda}(x)^2 + 1726\mathsf{b}_\lambda^2 + 172600 \frac{\mathsf{df}_\lambda \vee (\mathsf{Q}^\star)^2 \, \log \frac{2}{\delta}}{n}.$$

**2.** Moreover, from Thm. 8, it holds:

$$f(\widehat{x}_\lambda^\star) - f(x^\star) \leq 414\, \mathsf{b}_\lambda^2 + 414 \, \frac{\mathsf{df}_\lambda \vee (\mathsf{Q}^\star)^2}{n} \, \log \frac{2}{\delta}.$$

**3.** Putting things together:

$$f(x) - f(x^\star) \leq \mathsf{K}_1 \, \mathsf{b}_\lambda^2 + \mathsf{K}_2 \, \frac{\mathsf{df}_\lambda \vee (\mathsf{Q}^\star)^2}{n} \, \log \frac{2}{\delta} + \mathsf{K}_3 \, \widehat{\nu}_{\mathbf{P},\lambda}^2(x).$$

We bound the constants in the theorem.

$\square$

**Lemma 20.** *Under the conditions of the previous theorem, the following hold:*

- $\frac{1}{\widehat{\mathsf{r}}_{\mathbf{P},\lambda}(x)} \leq \frac{8}{\mathsf{r}_\lambda(x^\star)};$

- $\lambda^{1/2}\|x - x^\star\| \leq 7\widehat{\nu}_{\mathbf{P},\lambda}(x) + 59\mathsf{b}_\lambda + 568\sqrt{\dfrac{\mathsf{df}_\lambda \vee (\mathsf{Q}^\star)^2 \ \log\frac{2}{\delta}}{n}}$;

- $\lambda\|x\|_{\widehat{\mathbf{H}}_{\mathbf{P},\lambda}^{-1}(x)} \leq 7\widehat{\nu}_{\mathbf{P},\lambda}(x) + 72\mathsf{b}_\lambda + 720\sqrt{\dfrac{\mathsf{df}_\lambda \vee (\mathsf{Q}^\star)^2 \ \log\frac{2}{\delta}}{n}}$.

*In particular,* $\dfrac{\lambda\|x\|_{\widehat{\mathbf{H}}_{\mathbf{P},\lambda}^{-1}(x)}}{\widehat{\mathsf{r}}_{\mathbf{P},\lambda}(x)} \leq 11$.

*Proof.* Let us prove the three statements.

**1.** Write $\dfrac{1}{\widehat{\mathsf{r}}_{\mathbf{P},\lambda}(x)} = \sup_{g \in \widehat{\mathcal{G}}} \|\mathbf{P}g\|_{\widehat{\mathbf{H}}_{\mathbf{P},\lambda}^{-1}(x)}$. Now

$$\sup_{g \in \widehat{\mathcal{G}}} \|\mathbf{P}g\|_{\widehat{\mathbf{H}}_{\mathbf{P},\lambda}^{-1}(x)} \leq \sup_{g \in \widehat{\mathcal{G}}} \|g\|_{\widehat{\mathbf{H}}_\lambda^{-1}(x)} \leq e^{\widehat{\mathsf{t}}(x - \widehat{x}_\lambda^\star)/2} \sup_{g \in \widehat{\mathcal{G}}} \|g\|_{\widehat{\mathbf{H}}_\lambda^{-1}(\widehat{x}_\lambda^\star)}.$$

Now bound

$$\sup_{g \in \widehat{\mathcal{G}}} \|g\|_{\widehat{\mathbf{H}}_\lambda^{-1}(\widehat{x}_\lambda^\star)} \leq e^{\widehat{\mathsf{t}}(x_\lambda^\star - \widehat{x}_\lambda^\star)/2} \sup_{g \in \widehat{\mathcal{G}}} \|g\|_{\widehat{\mathbf{H}}_\lambda^{-1}(x_\lambda^\star)} \leq e^{\widehat{\mathsf{t}}(x_\lambda^\star - \widehat{x}_\lambda^\star)/2} \|\mathbf{H}_\lambda^{1/2}(x_\lambda^\star)\widehat{\mathbf{H}}_\lambda^{-1/2}(x_\lambda^\star)\| \sup_{g \in \widehat{\mathcal{G}}} \|g\|_{\mathbf{H}_\lambda^{-1}(x_\lambda^\star)}.$$

Finally bound

$$\sup_{g \in \widehat{\mathcal{G}}} \|g\|_{\mathbf{H}_\lambda^{-1}(x_\lambda^\star)} \leq e^{\mathsf{t}(x^\star - x_\lambda^\star)/2} \frac{1}{\mathsf{r}_\lambda(x^\star)}.$$

Now using the fact that under the previous assumptions $\mathsf{t}(x^\star - x_\lambda^\star), \mathsf{t}(x_\lambda^\star - \widehat{x}_\lambda^\star) \leq \log 2, \widehat{\mathsf{t}}(x - \widehat{x}_\lambda^\star) \leq \frac{3}{10} + 2\log 2$ and $\|\mathbf{H}_\lambda^{1/2}(x_\lambda^\star)\widehat{\mathbf{H}}_\lambda^{-1/2}(x_\lambda^\star)\| \leq \sqrt{2}$, we get the first equation.

**2.** In order to bound $\lambda^{1/2}\|x - x^\star\|$, decompose

$$\lambda^{1/2}\|x - x^\star\| \leq \lambda^{1/2}\|x - \widehat{x}_\lambda^\star\| + \lambda^{1/2}\|\widehat{x}_\lambda^\star - x^\star\|.$$

Now use Proposition 15 to bound $\lambda^{1/2}\|x - \widehat{x}_\lambda^\star\| \leq 7(\widehat{\nu}_{\mathbf{P},\lambda}(x) + \widehat{s}_\lambda)$. Using Lemma 18, under the conditions above,

$$\widehat{s}_\lambda \leq 8\,\mathsf{b}_\lambda + 80\sqrt{\dfrac{\mathsf{df}_\lambda \vee (\mathsf{Q}^\star)^2 \ \log\frac{2}{\delta}}{n}}.$$

Hence

$$\lambda^{1/2}\|x - \widehat{x}_\lambda^\star\| \leq 7\widehat{\nu}_{\mathbf{P},\lambda}(x) + 56\mathsf{b}_\lambda + 560\sqrt{\dfrac{\mathsf{df}_\lambda \vee (\mathsf{Q}^\star)^2 \ \log\frac{2}{\delta}}{n}}.$$

Moreover, using again Lemma 18

$$\lambda^{1/2}\|\widehat{x}_\lambda^\star - x^\star\| \leq 3\,\mathsf{b}_\lambda + 8\sqrt{\dfrac{\mathsf{df}_\lambda \vee (\mathsf{Q}^\star)^2 \ \log\frac{2}{\delta}}{n}}.$$

Combining these two inequalities, we get:

$$\lambda^{1/2}\|x - x^\star\| \leq 7\widehat{\nu}_{\mathbf{P},\lambda}(x) + 59\mathsf{b}_\lambda + 568\sqrt{\dfrac{\mathsf{df}_\lambda \vee (\mathsf{Q}^\star)^2 \ \log\frac{2}{\delta}}{n}}.$$

**3.** In order to bound $\lambda\|x\|_{\widehat{\mathbf{H}}_{\mathbf{P},\lambda}^{-1}(x)}$, use Proposition 15 to get $\lambda\|x\|_{\widehat{\mathbf{H}}_{\mathbf{P},\lambda}^{-1}(x)} \leq 7\widehat{\nu}_{\mathbf{P},\lambda}(x) + 9\widehat{s}_\lambda$.

Now using Lemma 18, the following bound holds:

$$\lambda\|x\|_{\widehat{\mathbf{H}}_{\mathbf{P},\lambda}^{-1}(x)} \leq 7\widehat{\nu}_{\mathbf{P},\lambda}(x) + 72\mathsf{b}_\lambda + 720\sqrt{\frac{\mathsf{df}_\lambda \vee (\mathsf{Q}^\star)^2 \, \log\frac{2}{\delta}}{n}}.$$

$\square$

**Proposition 19** (Simplification). *Let $n \in \mathbb{N}$, $\delta \in (0, 1/2]$, $0 < \lambda \leq \mathsf{B}_2^\star$. Let $x \in \mathcal{H}_{\mathbf{P}}$. Whenever*

$$n \geq \triangle_1 \frac{\mathsf{B}_2^\star}{\lambda} \log \frac{8\square_1^2 \mathsf{B}_2^\star}{\lambda\delta}, \qquad \mathsf{C}_1\sqrt{\frac{\mathsf{df}_\lambda \vee (\mathsf{Q}^\star)^2}{n} \, \log\frac{2}{\delta}} \leq \frac{\sqrt{\lambda}}{R}, \qquad \mathsf{C}_1\mathsf{b}_\lambda \leq \frac{\sqrt{\lambda}}{R},$$

*if*

$$\mathsf{C}_{\mathbf{P}}(x^\star, \lambda) \leq \frac{\sqrt{2}}{480}, \; \widehat{\nu}_{\mathbf{P},\lambda}(x) \leq \frac{\sqrt{\lambda}}{126R},$$

*then the following holds, with probability at least $1 - 2\delta$.*

$$f(x) - f(x^\star) \leq \mathsf{K}_1 \, \mathsf{b}_\lambda^2 + \mathsf{K}_2 \frac{\mathsf{df}_\lambda \vee (\mathsf{Q}^\star)^2}{n} \, \log\frac{2}{\delta} + \mathsf{K}_3 \, \widehat{\nu}_{\mathbf{P},\lambda}^2(x),$$

*where $\mathsf{K}_1 \leq 6.0\mathrm{e}4$, $\mathsf{K}_2 \leq 6.0\mathrm{e}6$ and $\mathsf{K}_3 \leq 810$, $\mathsf{C}_1$ are defined in Lemma 19, and the other constants are defined in Thm. 8.*

*Moreover, in that case, $R\|x - x^\star\| \leq 10$.*

### H.4 Optimal choice of $\lambda$, specific source conditions

In this part, we continue to assume Assumptions 6 to 9. We present a classification of distributions $\rho$ and show that we can achieve better rates than the classical slow rates, as presented in Appendix F of [23].

#### H.4.1 Classification of distributions and statistical bounds for the ERM

We use the following classification for distributions.

**Definition 9** (class of distributions). *Let $\alpha \in [1, +\infty]$ and $r \in [0, 1/2]$.*
*We denote with $\mathcal{P}_{\alpha,r}$ the set of probability distributions $\rho$ such that there exists $\mathsf{L}, \mathsf{Q} \geq 0$,*

- $\mathsf{b}_\lambda \leq \mathsf{L} \, \lambda^{\frac{1+2r}{2}}$;

- $\mathsf{df}_\lambda \leq \mathsf{Q}^2 \, \lambda^{-1/\alpha}$;

*where this holds for any $0 < \lambda \leq 1$. For simplicity, if $\alpha = +\infty$, we assume that $\mathsf{Q} \geq \mathsf{Q}^\star$.*

Note that given our assumptions, we always have

$$\rho \in \mathcal{P}_{1,0}, \quad \mathsf{L} = \|x^\star\|, \; \mathsf{Q} = \mathsf{B}_1^\star. \tag{40}$$

We also define

$$\lambda_1 = \left(\frac{\mathsf{Q}}{\mathsf{Q}^\star}\right)^{2\alpha} \wedge 1, \tag{41}$$

such that

$$\forall \lambda \leq \lambda_1, \; \mathsf{df}_\lambda \vee (\mathsf{Q}^\star)^2 \leq \frac{\mathsf{Q}^2}{\lambda^{1/\alpha}}.$$

**Interpretation of the classes**

- The bias term $b_\lambda$ characterizes the regularity of the objective $x^\star$. In a sense, if $r$ is big, then this means $x^\star$ is very regular and will be easier to estimate. The following results reformulates this intuition.

  **Remark 6** (source condition). *Assume there exists $0 \le r \le 1/2$ and $v \in \mathcal{H}$ such that*
  $$\mathbf{P}_{\mathbf{H}(x^\star)} x^\star = \mathbf{H}(x^\star)^r v.$$
  *Then it holds:*
  $$\forall \lambda > 0, \ b_\lambda \le \mathsf{L}\, \lambda^{\frac{1+2r}{2}}, \quad \mathsf{L} = \|\mathbf{H}(x^\star)^{-r} x^\star\|.$$

- The effective dimension $\mathsf{df}_\lambda$ characterizes the size of the space $\mathcal{H}$ with respect to the problem. The higher $\alpha$, the smaller the space. If $\mathcal{H}$ is finite dimensional for instance, $\alpha = +\infty$.

In this section, for any given pair $(\alpha, r)$ characterizing the regularity and size of the problem, we associate
$$\beta = \frac{1}{1 + 2r + 1/\alpha}, \quad \gamma = \frac{\alpha(1+2r)}{\alpha(1+2r)+1}.$$

In [23] (see corollary 3), explicit bounds are given for the performance of the regularized expected risk minimizer $\widehat{x}^\star_\lambda$ depending on which class $\rho$ belongs to, i.e., as a function of $\alpha, r$.

**Corollary 4.** *Let $\delta \in (0, 1/2]$. Under Assumptions 6 to 9, if $\rho \in \mathcal{P}_{\alpha,r}$ with $r > 0$, when $n \ge N$ and $\lambda = (C_0/n)^\beta$, then with probability at least $1 - 2\delta$,*
$$f(\widehat{x}^\star_\lambda) - f(x^\star) \le C_1 n^{-\gamma} \log \frac{2}{\delta},$$
*with $C_0 = 256(\mathsf{Q}/\mathsf{L})^2$, $C_1 = 8(256)^\gamma\,(\mathsf{Q}^\gamma\,\mathsf{L}^{1-\gamma})^2$ and $N$ defined in [23], and satisfying $N = O(\mathrm{poly}(\mathsf{B}^\star_1, \mathsf{B}^\star_2, \mathsf{L}, \mathsf{Q}, R, \log(1/\delta)))$.*

### H.4.2 Quantitative bounds for the projected problem

In this part, the aim is to show that if we approximately solve the projected problem up to a certain precision, then this approximation has the same statistical rates as the regularized ERM with the good choice of $\lambda$. For the sake of simplicity, we will assume that $r > 0$.

In what follows, we define

$$N = \frac{\mathsf{Q}^2}{\mathsf{L}^2}\,(\mathsf{B}^\star_2 \wedge \lambda_0 \wedge \lambda_1)^{-1/\beta} \quad \vee \quad \left(2.1e4\frac{1}{1-\beta}A\log\left(1.4e6\frac{1}{1-\beta}A^2\frac{1}{\delta}\right)\right)^{1/(1-\beta)}, \qquad (42)$$

where $A = \frac{\mathsf{B}^\star_2 \mathsf{L}^{2\beta}}{\mathsf{Q}^{2\beta}}$, $\lambda_0 = (\mathsf{C}_1 \mathsf{L} R \log \frac{2}{\delta})^{-1/r} \wedge 1$ and $\lambda_1 = \frac{\mathsf{Q}^{2\alpha}}{(\mathsf{Q}^\star)^{2\alpha}}$.

**Theorem 9** (Quantitative result with source $r > 0$). *Let $\rho \in \mathcal{P}_{\alpha,r}$ and assume $r > 0$. Let $\delta \in (0, \frac{1}{2}]$. Let $\mathbf{P}$ be an orthogonal projection, $x \in \mathcal{H}$. If*

$$n \ge N, \quad \lambda = \left(\left(\frac{\mathsf{Q}}{\mathsf{L}}\right)^2 \frac{1}{n}\right)^\beta, \quad \mathsf{C}_\mathbf{P}(x^\star, \lambda) \le \frac{\sqrt{2}}{480}, \quad \widehat{\nu}_{\mathbf{P},\lambda}(x) \le \mathsf{Q}^\gamma\,\mathsf{L}^{1-\gamma} n^{-\gamma/2}$$

*then with probability at least $1 - 2\delta$,*

$$f(x) - f(x^\star) \le \mathsf{K}\left(\mathsf{Q}^\gamma\,\mathsf{L}^{1-\gamma}\right)^2\,\frac{1}{n^\gamma}\log\frac{2}{\delta},$$

*where $N$ is defined in Eq. (42) and $\mathsf{K} \le 7.0e6$. Moreover, $R\|x - x^\star\| \le 10$.*

*Proof.* Using the definition of $\lambda_1$, as soon as $\lambda \le \lambda_1$, it holds: $\mathsf{df}_\lambda \vee (\mathsf{Q}^\star)^2 \le \mathsf{Q}^2 \lambda^{-1/\alpha}$.

Let us formulate Proposition 19 using the fact that $\rho \in \mathcal{P}_{\alpha,r}$.

Let $n \in \mathbb{N}$, $\delta \in (0, 1/2]$, $0 < \lambda \le \mathsf{B}_2^\star$, $x \in \mathcal{H}_{\mathbf{P}}$. Whenever

$$n \ge \triangle_1 \frac{\mathsf{B}_2^\star}{\lambda} \log \frac{8\square_1^2 \mathsf{B}_2^\star}{\lambda \delta}, \quad \mathsf{C}_1 \sqrt{\frac{\mathsf{Q}^2}{\lambda^{1/\alpha} n} \log \frac{2}{\delta}} \le \frac{\lambda^{1/2}}{R}, \mathsf{C}_1 \, \mathsf{L} \lambda^{1/2+r} \le \frac{\lambda^{1/2}}{R},$$

if

$$\mathsf{C}_{\mathbf{P}}(x^\star, \lambda) \le \frac{\sqrt{2}}{480}, \widehat{\nu}_{\mathbf{P},\lambda}(x) \le \mathsf{L}\lambda^{1/2+r},$$

The following holds, with probability at least $1 - 2\delta$.

$$f(x) - f(x^\star) \le (\mathsf{K}_1 + \mathsf{K}_3)\mathsf{L}^2 \lambda^{1+2r} + \mathsf{K}_2 \frac{\mathsf{Q}^2}{\lambda^{1/\alpha} n} \log \frac{2}{\delta}, \qquad R\|x - x^\star\| \le 10,$$

where all constants are defined in Proposition 19.

**Assume that** $r > 0$ . Define

$$\lambda_0 = (\mathsf{C}_1 \mathsf{L} R \log \frac{2}{\delta})^{-1/r} \wedge 1.$$

Then for any $\lambda \le \lambda_0$:

$$\mathsf{L}\lambda^{1/2+r} \le \frac{1}{\mathsf{C}_1} \frac{\sqrt{\lambda}}{R}.$$

**1)** First, we find a simple condition to guarantee

$$\mathsf{r}_\lambda(x^\star)^2 \lambda^{1/\alpha} \ge \mathsf{C}_2 \, \mathsf{Q}^2 \frac{1}{n} \, \log \frac{2}{\delta}.$$

We see that if $\lambda \le \lambda_0$, then $\mathsf{r}_\lambda \ge \mathsf{C}_1 \mathsf{L} \lambda^{1/2+r} \log \frac{2}{\delta}$. Hence, this condition is satisfied if

$$\lambda \le \lambda_0, \quad \mathsf{C}_1^2 \mathsf{L}^2 \lambda^{1+2r+1/\alpha} \ge \mathsf{C}_2 \, \mathsf{Q}^2 \frac{1}{n}.$$

Using the fact that $\mathsf{C}_2 = \mathsf{C}_1^2$, we reformulate:

$$\lambda \le \lambda_0, \quad \mathsf{L}^2 \lambda^{1+2r+1/\alpha} \ge \mathsf{Q}^2 \frac{1}{n}.$$

**2)** Now fix

$$\lambda^{1+2r+1/\alpha} = \frac{\mathsf{Q}^2}{\mathsf{L}^2} \frac{1}{n} \iff \lambda = \left(\frac{\mathsf{Q}^2}{\mathsf{L}^2} \frac{1}{n}\right)^{\beta}.$$

where $\beta = 1/(1 + 2r + 1/\lambda) \in [1/2, 1)$.

Using our restatement of Proposition 18, with probability at least $1 - 2\delta$,

$$L(x) - L(x^\star) \le \left(\mathsf{K}_1 + \mathsf{K}_3 + \mathsf{K}_2 \log \frac{2}{\delta}\right) \mathsf{L}^2 \lambda^{1+2r} \le \mathsf{K} \log \frac{2}{\delta} \mathsf{L}^2 \lambda^{1+2r},$$

where $\mathsf{K} = \mathsf{K}_1 + \mathsf{K}_3 + \mathsf{K}_2 \le 7.0e6$ (see Proposition 18).
This result holds provided

$$0 < \lambda \le \mathsf{B}_2^\star \wedge \lambda_0 \wedge \lambda_1, \ n \ge \triangle_1 \frac{\mathsf{B}_2^\star}{\lambda} \log \frac{8\square_1^2 \mathsf{B}_2^\star}{\lambda \delta}. \tag{43}$$

Indeed, it is shown in the previous point that the other conditions are satisfied.

**3)** Let us now work to guarantee the conditions in Eq. (43).
First, to guarantee $n \ge \triangle_1 \frac{\mathsf{B}_2^\star}{\lambda} \log \frac{8\square_1^2 \mathsf{B}_2^\star}{\lambda \delta}$, bound

$$\frac{\mathsf{B}_2^\star}{\lambda} = \frac{\mathsf{B}_2^\star \mathsf{L}^{2\beta} n^\beta}{\mathsf{Q}^{2\beta} \log^\beta \frac{2}{\delta}} \le 2 \frac{\mathsf{B}_2^\star \mathsf{L}^{2\beta}}{\mathsf{Q}^{2\beta}} n^\beta.$$

Then apply lemma 15 from [23] with $a_1 = 2\triangle_1$, $a_2 = 16\square_1^2$, $A = \frac{\mathsf{B}_2^\star \mathsf{L}^{2\beta}}{\mathsf{Q}^{2\beta}}$. Since $\beta \geq 1/2$, using the bounds in Thm. 8, we find $a_1 \leq 10400$ and $a_2 \leq 64$, hence the following sufficient condition:

$$n \geq \left( 2.1e4 \frac{1}{1-\beta} A \log \left( 1.4e6 \frac{1}{1-\beta} A^2 \frac{1}{\delta} \right) \right)^{1/(1-\beta)} .$$

Then, to guarantee the condition

$$\lambda \leq \mathsf{B}_2^\star \wedge \lambda_0 \wedge \lambda_1,$$

we simply need

$$n \geq \frac{\mathsf{Q}^2}{\mathsf{L}^2} \left( \mathsf{B}_2^\star \wedge \lambda_0 \wedge \lambda_1 \right)^{-1/\beta} .$$

Hence, defining

$$N = \frac{\mathsf{Q}^2}{\mathsf{L}^2} \left( \mathsf{B}_2^\star \wedge \lambda_0 \wedge \lambda_1 \right)^{-1/\beta} \quad \vee \quad \left( 2.1e4 \frac{1}{1-\beta} A \log \left( 1.4e6 \frac{1}{1-\beta} A^2 \frac{1}{\delta} \right) \right)^{1/(1-\beta)} ,$$

we see that as soon as $n \geq N$, Eq. (43) holds.

$\square$

# I  Multiplicative approximations for Hermitian operators

In this section, we put together useful tools for approximating linear operators and solving linear systems with regularization.

In this section, $\mathbf{A}$ and $\mathbf{B}$ will always denote positive semi-definite Hermitian operators on a Hilbert space $\mathcal{H}$, and $\mathbf{P}$ will denote an orthogonal projection operator. Moreover, given a positive semi-definite operator $\mathbf{A}$, and $\lambda > 0$, $\mathbf{A}_\lambda$ will stand for the regularized operator $\mathbf{A} + \lambda \mathbf{I}$.

**Lemma 21** (Equivalence of Hermitian operators). *Let $\mathbf{A}$ and $\mathbf{B}$ be two semi-definite Hermitian operators. Let $\lambda > 0$. Assume you have access to*

$$t := \|\mathbf{A}_\lambda^{-1/2}(\mathbf{B} - \mathbf{A})\mathbf{A}_\lambda^{-1/2}\|.$$

*It holds:*

$$\|\mathbf{A}_\lambda^{-1/2}\mathbf{B}_\lambda^{1/2}\|^2 \leq 1 + t \Leftrightarrow \mathbf{B}_\lambda \preceq (1 + t)\mathbf{A}_\lambda.$$

*Moreover, if $t < 1$,*

$$\|\mathbf{B}_\lambda^{-1/2}\mathbf{A}_\lambda^{1/2}\|^2 \leq \frac{1}{1 - t} \Leftrightarrow (1 - t)\mathbf{A}_\lambda \preceq \mathbf{B}_\lambda.$$

*Proof.* For the first point, simply note that:

$$\|\mathbf{A}_\lambda^{-1/2}\mathbf{B}_\lambda^{1/2}\|^2 = \|\mathbf{A}_\lambda^{-1/2}\mathbf{B}_\lambda\mathbf{A}_\lambda^{-1/2}\| = \|\mathbf{I} + \mathbf{A}_\lambda^{-1/2}(\mathbf{B} - \mathbf{A})\mathbf{A}_\lambda^{-1/2}\| \leq 1 + t.$$

For the second point,

$$\|\mathbf{B}_\lambda^{-1/2}\mathbf{A}_\lambda^{1/2}\|^2 = \|\left(\mathbf{A}_\lambda^{-1/2}\mathbf{B}_\lambda\mathbf{A}_\lambda^{-1/2}\right)^{-1}\| = \|\left(\mathbf{I} + \mathbf{A}_\lambda^{-1/2}(\mathbf{B} - \mathbf{A})\mathbf{A}_\lambda^{-1/2}\right)^{-1}\|.$$

Moreover, we know that if $\|\mathbf{H}\| < 1$ with $\mathbf{H}$ a Hermitian operator, then $\|(\mathbf{I} + \mathbf{H})^{-1}\| \leq \frac{1}{1 - \|\mathbf{H}\|}$. The result follows. $\square$

We will now state a technical lemma which describes how combining approximation behaves.

**Lemma 22** (Combination of approximations). *Let $N \geq 1$. Let $(\mathbf{A}_i)_{1 \leq i \leq N+1}$ be a sequence of positive semi-definite Hermitian operators. Define*

$$t_i := \|\mathbf{A}_{i,\lambda}^{-1/2}(\mathbf{A}_{i+1} - \mathbf{A}_i)\mathbf{A}_{i,\lambda}^{-1/2}\|.$$

*For any $1 \leq i, j \leq N + 1$, define*

$$t_{i:j} := \|\mathbf{A}_{i,\lambda}^{-1/2}(\mathbf{A}_j - \mathbf{A}_i)\mathbf{A}_{i,\lambda}^{-1/2}\|.$$

*In particular, $t_i = t_{i:i+1}$. Then the following holds:*

$$\forall 1 \leq i \leq j \leq N,\ 1 + t_{i:j} \leq \prod_{k=i}^{j-1}(1 + t_k)$$

*Moreover, if $t_i < 1$, then it holds:*

$$\|\mathbf{A}_{i+1,\lambda}^{-1/2}(\mathbf{A}_i - \mathbf{A}_{i+1})\mathbf{A}_{i+1,\lambda}^{-1/2}\| \leq \frac{t_i}{1 - t_i}$$

*Hence, in that case*

$$\forall 1 \leq j \leq i \leq N,\ 1 + t_{j:i} \leq \prod_{k=i}^{j-1}\frac{1}{1 - t_k}$$

*Proof.* Let us prove everything for a sequence of three operators; the rest follows by induction. Let $\mathbf{A}_1, \mathbf{A}_2, \mathbf{A}_3$ be three positive semi-definite operators.

1. **Bound**

$$t_{1:3} = \|\mathbf{A}_{1,\lambda}^{-1/2}(\mathbf{A}_1 - \mathbf{A}_3)\mathbf{A}_{1,\lambda}^{-1/2}\|$$
$$\leq \|\mathbf{A}_{1,\lambda}^{-1/2}(\mathbf{A}_1 - \mathbf{A}_2)\mathbf{A}_{1,\lambda}^{-1/2}\| + \|\mathbf{A}_{1,\lambda}^{-1/2}(\mathbf{A}_2 - \mathbf{A}_3)\mathbf{A}_{1,\lambda}^{-1/2}\|$$
$$\leq t_{1:2} + \|\mathbf{A}_{1,\lambda}^{-1/2}\mathbf{A}_{2,\lambda}^{1/2}\|^2 t_{2:3}$$
$$\leq t_{1:2} + (1 + t_{1:2})t_{2:3}.$$

The last line comes from Lemma 21. Thus

$$1 + t_{1:3} \leq 1 + t_{1:2} + t_{2:3} + t_{1:2}t_{2:3} = (1 + t_{1:2})(1 + t_{2:3}).$$

2. Let us now bound $t_{2:1}$ knowing $t_{1:2}$. This will imply the rest of the lemma.

Indeed, simply note that

$$t_{2:1} = \|\mathbf{A}_{2,\lambda}^{-1/2}(\mathbf{A}_2 - \mathbf{A}_1)\mathbf{A}_{2,\lambda}^{-1/2}\| \leq \|\mathbf{A}_{2,\lambda}^{-1/2}\mathbf{A}_{1,\lambda}^{1/2}\|^2 \, t_{1:2}.$$

Using Lemma 21, if $t_{1:2} < 1$, $\|\mathbf{A}_{2,\lambda}^{-1/2}\mathbf{A}_{1,\lambda}^{1/2}\|^2 \leq \frac{1}{1-t_{1:2}}$, hence

$$t_{2:1} \leq \frac{t_{1:2}}{1 - t_{1:2}}.$$

$\square$

**Lemma 23** (Projection of Hermitian operators)**.** *For any Hermitian operator* $\mathbf{A}$ *and orthogonal projection* $\mathbf{P}$*, the following holds:*

$$\|\mathbf{A}_\lambda^{-1/2}(\mathbf{A} - \mathbf{P}\mathbf{A}\mathbf{P})\mathbf{A}_\lambda^{-1/2}\| \leq \left(1 + \frac{\|\mathbf{A}^{1/2}(\mathbf{I} - \mathbf{P})\|}{\sqrt{\lambda}}\right)^2 - 1.$$

*In particular,*

$$\|\mathbf{A}_\lambda^{-1/2}(\mathbf{P}\mathbf{A}\mathbf{P} + \lambda\mathbf{I})^{1/2}\| \leq 1 + \frac{\|\mathbf{A}^{1/2}(\mathbf{I} - \mathbf{P})\|}{\sqrt{\lambda}}.$$

*Moreover, if*

$$\frac{\|\mathbf{A}^{1/2}(\mathbf{I} - \mathbf{P})\|}{\sqrt{\lambda}} < \sqrt{2} - 1,$$

*then it holds*

$$\|\mathbf{A}_\lambda^{1/2}(\mathbf{P}\mathbf{A}\mathbf{P} + \lambda\mathbf{I})^{-1/2}\|^2 \leq \frac{1}{2 - \left(1 + \frac{\|\mathbf{A}^{1/2}(\mathbf{I}-\mathbf{P})\|}{\sqrt{\lambda}}\right)^2}.$$

*We also always have:*

$$\|(\mathbf{P}\mathbf{A}\mathbf{P} + \lambda\mathbf{I})^{-1/2}\mathbf{P}\mathbf{A}_\lambda^{1/2}\|^2 \leq 1.$$

*Proof.* For any Hermitian operator $\mathbf{A}$, the following computation holds:

$$\|\mathbf{A}_\lambda^{-1/2}(\mathbf{A} - \mathbf{P}\mathbf{A}\mathbf{P})\mathbf{A}_\lambda^{-1/2}\| = \|\mathbf{A}_\lambda^{-1/2}(\mathbf{A} - (\mathbf{I} - (\mathbf{I} - \mathbf{P}))\mathbf{A}(\mathbf{I} - (\mathbf{I} - \mathbf{P}))\mathbf{A}_\lambda^{-1/2}\|$$
$$\leq 2\|\mathbf{A}_\lambda^{-1/2}(\mathbf{I} - \mathbf{P})\mathbf{A}\mathbf{A}_\lambda^{-1/2}\| + \|\mathbf{A}_\lambda^{-1/2}(\mathbf{I} - \mathbf{P})\mathbf{A}(\mathbf{I} - \mathbf{P})\mathbf{A}_\lambda^{-1/2}\|$$
$$\leq \frac{2\|\mathbf{A}^{1/2}(\mathbf{I} - \mathbf{P})\|}{\sqrt{\lambda}} + \frac{\|\mathbf{A}^{1/2}(\mathbf{I} - \mathbf{P})\|^2}{\lambda}$$
$$= \left(1 + \frac{\|\mathbf{A}^{1/2}(\mathbf{I} - \mathbf{P})\|}{\sqrt{\lambda}}\right)^2 - 1.$$

$\square$

**Lemma 24** (Relationship between approximations)**.** *Let* $\mathbf{A}$ *and* $\mathbf{B}$ *be two positive semi-definite hermitian operators. Let* $\lambda > 0$, $b \in \mathcal{H}$ *and* $\rho > 0$*. If*

$$\|\mathbf{A}_\lambda^{-1/2}(\mathbf{B} - \mathbf{A})\mathbf{A}_\lambda^{-1/2}\| \leq \frac{1}{2} \wedge \frac{\rho}{4}, \qquad \widetilde{\Delta} \in \mathrm{LinApprox}(\mathbf{B}_\lambda, b, \rho/4),$$

*then it holds:*

$$\widetilde{\Delta} \in \mathrm{LinApprox}(\mathbf{A}_\lambda, b, \rho).$$

*Proof.* Assume $\widetilde{\Delta} \in \mathrm{LinApprox}(\mathbf{B}_\lambda, b, \tilde{\rho}/4)$ for a certain $\tilde{\rho}$. Decompose

$$\|\mathbf{A}_\lambda^{-1}b - \widetilde{\Delta}\|_{\mathbf{A}_\lambda} \leq \|\mathbf{A}_\lambda^{-1}b - \mathbf{B}_\lambda^{-1}b\|_{\mathbf{A}_\lambda} + \|\mathbf{B}_\lambda^{-1}b - \widetilde{\Delta}\|_{\mathbf{A}_\lambda}$$
$$\leq \|\mathbf{A}_\lambda^{1/2}(\mathbf{A}_\lambda^{-1} - \mathbf{B}_\lambda^{-1})\mathbf{A}_\lambda^{1/2}\| \, \|b\|_{\mathbf{A}_\lambda^{-1}} + \|\mathbf{A}_\lambda^{1/2}\mathbf{B}_\lambda^{-1/2}\| \, \|\mathbf{B}_\lambda^{-1}b - \widetilde{\Delta}\|_{\mathbf{B}_\lambda}.$$

Now using the fact that $\mathbf{A}_\lambda^{-1} - \mathbf{B}_\lambda^{-1} = \mathbf{B}_\lambda^{-1}(\mathbf{B} - \mathbf{A})\mathbf{A}_\lambda^{-1}$,

$$\|\mathbf{A}_\lambda^{1/2}(\mathbf{A}_\lambda^{-1} - \mathbf{B}_\lambda^{-1})\mathbf{A}_\lambda^{1/2}\| \leq \|\mathbf{A}_\lambda^{-1/2}(\mathbf{B} - \mathbf{A})\mathbf{A}_\lambda^{-1/2}\| \, \|\mathbf{A}_\lambda^{1/2}\mathbf{B}_\lambda^{-1}\mathbf{A}_\lambda^{1/2}\|$$
$$= \|\mathbf{A}_\lambda^{-1/2}(\mathbf{B} - \mathbf{A})\mathbf{A}_\lambda^{-1/2}\| \, \|\mathbf{A}_\lambda^{1/2}\mathbf{B}_\lambda^{-1/2}\|^2.$$

Moreover,
$$\|\mathbf{B}_\lambda^{-1}b - \widetilde{\Delta}\|_{\mathbf{B}_\lambda} \leq \tilde{\rho}\|b\|_{\mathbf{B}_\lambda^{-1}} \leq \|\mathbf{A}^{1/2}\mathbf{B}^{-1/2}\| \, \|b\|_{\mathbf{A}_\lambda^{-1}}.$$

Putting things together, and noting that from Lemma 21, $\|\mathbf{A}^{1/2}\mathbf{B}^{-1/2}\|^2 \leq \frac{1}{1 - \|\mathbf{A}_\lambda^{-1/2}(\mathbf{B}-\mathbf{A})\mathbf{A}_\lambda^{-1/2}\|}$ as soon as $\|\mathbf{A}_\lambda^{-1/2}(\mathbf{B} - \mathbf{A})\mathbf{A}_\lambda^{-1/2}\| < 1$, it holds:

$$\widetilde{\Delta} \in \mathrm{LinApprox}(\mathbf{A}_\lambda, b, \rho), \ \rho = \frac{\tilde{\rho} + \|\mathbf{A}_\lambda^{-1/2}(\mathbf{B} - \mathbf{A})\mathbf{A}_\lambda^{-1/2}\|}{1 - \|\mathbf{A}_\lambda^{-1/2}(\mathbf{B} - \mathbf{A})\mathbf{A}_\lambda^{-1/2}\|}.$$

Choosing the right values for $\tilde{\rho}$ and $\|\mathbf{A}_\lambda^{-1/2}(\mathbf{B} - \mathbf{A})\mathbf{A}_\lambda^{-1/2}\|$ yields the result. $\qquad\square$

## I.1   Results for Nystrom sub-sampling

Recall the notations from Appendix D.

We write without proof the following lemmas, which are just restatements of lemmas 9 and 10 of [29].

**Lemma 25** (Uniform sampling). *Let $\delta > 0$. If $\{\tilde{z}_1, ..., \tilde{z}_m\}$ are sampled uniformly, then if $0 < \lambda < \|\mathbf{A}\|$, $m \leq n$ and*

$$m \geq \left(10 + 160\mathcal{N}_\infty^{\mathbf{A}}(\lambda)\right) \log \frac{8\|v\|_{L^\infty(Z)}^2}{\lambda\delta}.$$

*Then it holds, with probability at least $1 - \delta$:*

$$\|\mathbf{A}_\lambda^{-1/2}(\widehat{\mathbf{A}} - \mathbf{A})\mathbf{A}_\lambda^{-1/2}\| \leq \frac{1}{2}, \qquad \|\widehat{\mathbf{A}}_{m,\lambda}^{-1/2}(\widehat{\mathbf{A}} - \widehat{\mathbf{A}}_m)\widehat{\mathbf{A}}_{m,\lambda}^{-1/2}\| \leq \frac{1}{2}.$$

**Lemma 26** (Nystrom sampling). *Let $\delta > 0$. If $\{\tilde{z}_1, ..., \tilde{z}_m\}$ are sampled using $q$-approximate leverage scores for $t = \lambda$, then if $t_0 \vee \frac{19\|v\|_{L^\infty(Z)}^2}{n} \log \frac{n}{2\delta} < \lambda < \|\mathbf{A}\|$, and $n \geq 405\|v\|_{L^\infty(Z)}^2 \vee 67\|v\|_{L^\infty(Z)}^2 \log \frac{12\|v\|_{L^\infty(Z)}^2}{\delta}$, if*

$$m \geq \left(6 + 486q^2\mathcal{N}^{\mathbf{A}}(\lambda)\right) \log \frac{8\|v\|_{L^\infty(Z)}^2}{\lambda\delta}.$$

*Then it holds, with probability at least $1 - \delta$:*

$$\|\mathbf{A}_\lambda^{-1/2}(\widehat{\mathbf{A}} - \mathbf{A})\mathbf{A}_\lambda^{-1/2}\| \leq \frac{1}{2}, \qquad \|\widehat{\mathbf{A}}_{m,\lambda}^{-1/2}(\widehat{\mathbf{A}} - \widehat{\mathbf{A}}_m)\widehat{\mathbf{A}}_{m,\lambda}^{-1/2}\| \leq \frac{1}{2}.$$

**Lemma 27.** *Let $\lambda > 0$. Assume:*

$$\|\mathbf{A}_\lambda^{-1/2}(\widehat{\mathbf{A}} - \mathbf{A})\mathbf{A}_\lambda^{-1/2}\| \leq \frac{1}{2}, \qquad \|\widehat{\mathbf{A}}_{m,\lambda}^{-1/2}(\widehat{\mathbf{A}} - \widehat{\mathbf{A}}_m)\widehat{\mathbf{A}}_{m,\lambda}^{-1/2}\| \leq \frac{1}{2}.$$

*Denote with $P_m$ the projection on $\mathrm{span}(v_{\tilde{z}_j})_{1 \leq j \leq m}$. Then the following holds:*

$$\|\mathbf{A}_\lambda^{1/2}(\mathbf{I} - P_m)\|^2 \leq 3\lambda,$$

*and for any partial isometry $V$,*

$$\frac{1}{2}\left(V^*\widehat{\mathbf{A}}_m V + \lambda\mathbf{I}\right) \preceq V^*\widehat{\mathbf{A}}V + \lambda\mathbf{I} \preceq \frac{3}{2}\left(V^*\widehat{\mathbf{A}}_m V + \lambda\mathbf{I}\right).$$

*Proof.* For the first point, use the well known fact that

$$\mathbf{I} - \mathbf{P}_m \leq \lambda \widehat{\mathbf{A}}_{m,\lambda}^{-1},$$

since the range of $\mathbf{P}_m$ contains that of $\widehat{\mathbf{A}}_m$. Thus,

$$\|\mathbf{A}_\lambda^{1/2}(\mathbf{I} - \mathbf{P}_m)\|^2 \leq \lambda \|\mathbf{A}_\lambda^{1/2} \widehat{\mathbf{A}}_{m,\lambda}^{-1/2}\|^2.$$

Now using Lemma 22,

$$\|\mathbf{A}_\lambda^{-1/2}(\widehat{\mathbf{A}} - \mathbf{A})\mathbf{A}_\lambda^{-1/2}\| \leq \frac{1}{2} \implies \|\widehat{\mathbf{A}}_\lambda^{-1/2}(\widehat{\mathbf{A}} - \mathbf{A})\widehat{\mathbf{A}}_\lambda^{-1/2}\| \leq 1.$$

Hence, again using Lemma 22,

$$\|\widehat{\mathbf{A}}_{m,\lambda}^{-1/2}(\widehat{\mathbf{A}}_m - \mathbf{A})\widehat{\mathbf{A}}_{m,\lambda}^{-1/2}\| \leq 2,$$

and therefore, using Lemma 21,

$$\|\mathbf{A}_\lambda^{1/2} \widehat{\mathbf{A}}_{m,\lambda}^{-1/2}\|^2 \leq 3.$$

For the second point, this is only a consequence of Lemma 21. $\qquad\square$

Now state two results which show that

**Lemma 28** (Uniform sampling yielding $\rho$-approximation). *Let $0 < \rho \leq 1$ and $\delta > 0$. Let $b \in \mathcal{H}$. If $\{\tilde{z}_1, ..., \tilde{z}_m\}$ are sampled uniformly, $0 < \lambda < \|\mathbf{A}\|$, $m \leq n$ and*

$$m \geq \left(2 + \frac{48}{\rho} + \frac{5000}{\rho^2}\mathcal{N}_\infty^{\mathbf{A}}(\lambda)\right) \log \frac{8\|v\|_{L^\infty(Z)}^2}{\lambda\delta}.$$

*Then it holds, with probability at least $1 - \delta$:*

$$x \in \mathrm{LinApprox}(\widehat{\mathbf{A}}_{m,\lambda}, b, \rho/4) \implies x \in \mathrm{LinApprox}(\mathbf{A}_\lambda, b, \rho).$$

*In particular, with probability $1 - \delta$,*

$$\widehat{\mathbf{A}}_{m,\lambda}^{-1}b \in \mathrm{LinApprox}(\mathbf{A}_\lambda, b, \rho).$$

*Proof.* Apply Lemma 9 from [29] with $\eta = \frac{\rho}{12} < \frac{1}{2}$. We find that under the conditions above, with probability at least $1 - \delta$,

$$\|\mathbf{A}_\lambda^{-1/2}(\widehat{\mathbf{A}} - \mathbf{A})\mathbf{A}_\lambda^{-1/2}\| \leq \eta, \qquad \|\widehat{\mathbf{A}}_{m,\lambda}^{-1/2}(\widehat{\mathbf{A}} - \widehat{\mathbf{A}}_m)\widehat{\mathbf{A}}_{m,\lambda}^{-1/2}\| \leq \eta.$$

Now use Lemma 22 to see that

$$\|\mathbf{A}_\lambda^{-1/2}(\widehat{\mathbf{A}}_m - \mathbf{A})\mathbf{A}_\lambda^{-1/2}\| \leq (1 + \eta^2) - 1 \leq 3\eta \leq \rho/4.$$

Thus, we can apply Lemma 24 to get the desired result. $\qquad\square$

**Lemma 29** (Leverage scores Nystrom sampling yielding $\rho$-approximation). *Let $\delta > 0$. If $\{\tilde{z}_1, ..., \tilde{z}_m\}$ are sampled using $q$-approximate leverage scores for $t = \lambda$, then if $t_0 \vee \frac{19\|v\|_{L^\infty(Z)}^2}{n} \log \frac{n}{2\delta} < \lambda < \|\mathbf{A}\|$, and $n \geq 405\|v\|_{L^\infty(Z)}^2 \vee 67\|v\|_{L^\infty(Z)}^2 \log \frac{12\|v\|_{L^\infty(Z)}^2}{\delta}$, if*

$$m \geq \left(2 + \frac{24}{\rho} + \frac{13000q^2}{\rho^2}\mathcal{N}^{\mathbf{A}}(\lambda)\right) \log \frac{8\|v\|_{L^\infty(Z)}^2}{\lambda\delta}.$$

*Then it holds, with probability at least $1 - \delta$:*

$$x \in \mathrm{LinApprox}(\widehat{\mathbf{A}}_{m,\lambda}, b, \rho/4) \implies x \in \mathrm{LinApprox}(\mathbf{A}_\lambda, b, \rho).$$

*In particular, with probability $1 - \delta$,*

$$\widehat{\mathbf{A}}_{m,\lambda}^{-1}b \in \mathrm{LinApprox}(\mathbf{A}_\lambda, b, \rho).$$

*Proof.* The proof is exactly the same as that of the previous lemma, using Lemma 10 instead of Lemma 9 in [29]. $\qquad\square$