[Reviews · NeurIPS 2019]

Reviewer 1



This paper proposes an approximate Newton algorithm (using e.g. sketching or subsampling) for minimizing generalized self-concordant convex functions for supervised learning problems. The main novelty of the algorithm itself appears to be that, instead of finding the region of fast convergence with a linear method, they use their own algorithm, with a decreasing sequence of regularization parameters. The primary benefit of their approach is that it extends earlier convergence convergence rates that are indepentent of the condition number (in the self-concordant setting) to the generalized-self-concordant setting (although it apparently isn't quite so simple---see Remark 1 of the paper). Writing Quality --------------- First, the paper is overall well-written, and fairly comprehensive (i.e. I think that the authors considered everything that needed to be considered, although one might argue about how much attention was paid to various portions). With that said, there are a few things that confused me, and might be possible to clarify: 1) In definition 1, what does the bracket notation after \nabla^3 and \nabla^2 mean? Also, a discussion of how, exactly, this definition differs from that of [6], and why these differences matter, would be appreciated. Going further, it might be nice to discuss how it differs from regular self-concordance. 2) The "contributions" paragraph of the introduction is valuable, but I think it would be more useful (at least to a reviewer, and we're presumably the intended audience of this paragraph) if it not only said *what* was new, but more explicitly *how it improved* on the prior state-of-the art. This is obviously intended to be fleshed-out in Section 2, but even there, the differences between the proposal and the references are not explicit. For example, I'm not sure how this paper differs from prior generalized-self-concordant work (e.g. [6,7]). Is the main difference in the method by which you find the region of fast convergence? That they didn't use approximate Newton (or used a "different" approximate Newton)? That they did not pay as much attention to condition numbers? Something else? 3) Following-up on this, could some of the information in Section 2 be summarized in a Table? With corresponding cuts in the section itself, this could wind up being more space-efficient, in addition to being easier to use as a reference when reading later in the paper. 4) A lot of notation is introduced, and it's hard for a reader to keep track of all of it (Q, for example, stymied me for a while, in the discussion after Theorem 1). A table of notation in the appendix might mitigate this. Finally, as a minor quibble, late in Section 3 the expression "way smaller" is used repeatedly. I'm not complaining about the informality, but the repetition is jarring. Contribution ------------ I'm not familiar enough with the background literature to judge the magnitude of the contribution of the paper. I assume that that the idea of using an approximate Newton method based on relative approximations as in Definition 2 is *not* new---is this correct? If so, then it appears to me as if the main contribution is in using their own algorithm (instead of a first-order algorithm), with a decreasing schedule of regularization parameters, to find the region of fast convergence. Assuming that this really is a new approach (is it?), and that it is what's mostly responsible for reducing/removing their dependence on the condition number (is this true?), then it seems to me to be a significant contribution. More to the point, I think it would be of sufficient magnitude to warrant publication in NeurIPS. But, as I said, I don't have enough familiarity with the background literature to really know. With that said, there are two obvious weaknesses: (1) this work only applies to certain particular convex problems (Example 1 in the paper discusses the main instances), and (2) many machine learning practitioners have developed a well-earned skepticism of the applicability of higher-order methods (or really, of anything with a high per-iteration cost) to ML problems. These, particularly point (2), might seem unfair, and to some extent they are (past failures may indicate future failures---hence "well-earned"---but they don't require them), but thankfully they could still be addressed with sufficiently comprehensive experiments. The experiments are decent. They use large real-world datasets on realistic problems, and achieve impressive-seeming results. What they were not---and for the reasons mentioned above I think this is important---was comprehensive. For one, both were in the kernel setting, and for another, both used logistic regression. There was also only one baseline (KSVRG). Where were the other settings mentioned in Example 1, and where were the numerous potentional baselines mentioned in Section 2? More importantly, these are machine learning applications, so "distance to optimum" doesn't strike me as being the correct metric. What one is really interested in is "accuracy on a test set", which I suspect plateaus at a far less precise solution than is plotted in Figure 1 (i.e. I suspect that KSVRG, and other alternatives, will appear far more competitive measured in terms of test accuracy). ==== AFTER AUTHOR RESPONSE ==== Thank you for responding so thoroughly to my review. The changes that you suggested address nearly all of my concerns, and I'm happy to see that your approach does indeed outperform KSVRG in terms of classification error. I'm raising my score to 6.

Reviewer 2



Originality ------------ The paper present a very novel Newton method algorithm adapted to large-scale ill-conditioned problems. The authors clearly state how their algorithm differ and relates to the literature. Quality --------- The authors present a very complete study of an optimization algorithm and provide a. Due to the (over)length of the appendix, it is difficult to carefully check all the proofs. Maybe a journal would be a more appropriate submission track. Nevertheless all claims in the paper are justified and the proofs look technically sound. For completeness and transparency, the experiment against FALKON should be included in the main paper, as it is a directly related work. Clarity -------- The paper is very dense in information, but well written and organized. All the elements required for the understanding of the paper are provided. There is some formatting issue line 211: the box is too big. No conclusion is provided. Significance --------------- Even tough a detailed pseudo-code is given in the appendix, the impact and significance of the paper could be improved a lot by providing a functional piece of code, as this algorithm could interest and impact a large community. I have read the author response and my opinion remains the same. I have read the author response and my opinion remains the same.

Reviewer 3



I thank the authors for their response, and would like to maintain my overall evaluation. ===== The paper is generally well-written and addresses a fundamental problem. The proposed algorithm seems sound, although I will let other reviewers comment on its novelty as I may be not familiar with the most recent developments in second-order methods. The appendix is quite comprehensive, but I have not gone through to verify the proof details. The experiments (Section F in the Appendix provides more details) appear to confirm the theoretical claims, although it may be helpful to discuss or compare with the Newton sketch of [24] (same as [25]). A couple questions regarding related work: - I'm still unclear how the proposed method compares to the Newton sketch of [24]/[25] both theoretically and empirically, in terms of accuracy and computational complexity. - As the authors noted in Section 1.1, methods based on sketching and sub-sampling are essentially independent of the condition number. In this case, what are the advantages of the proposed approach?

[Author Response · NeurIPS 2019]

We thank reviewers for the time they took to read the paper and provide comments and suggestions.

**Reviewer 1.** As the reviewer correctly points out one crucial aspect of the proposed algorithm is exactly the inde-
pendence from the condition number of the problem. More generally, note that the paper provides an optimization
algorithm that at the same time is: (a) independent (except for logarithmic terms) from the condition number of the
problem, (b) globally convergent (unlike [6,7], the method will reach the optimum point from zero), (c) efficient from a
computational viewpoint. Essentially all the known optimization algorithms nowadays satisfy only a subset of the points
above. Indeed first order methods don't satisfy (a); Second order methods for general losses only (a); Second order with
approximation methods as in Def. 2 satisfy (a)+(c). However the fact that they are not globally convergent make them
not usable on real problems, since a first order method is needed to arrive very close to the optimum, losing (a). Finally
there exists the very peculiar class of self-concordant function for which it has been proven that it is possible to achieve
(a)+(b)+(c). However such result is more of theoretical interest in ML since very few objective functions there are self
concordant. The novel contribution of the paper is a globally convergent optimization scheme that allows to achieve
(a)+(b)+(c) for a larger class of functions i.e. *generalized self concordant* functions (GSC), which instead contains many
losses of interest for regression, multivalued regression, robust regression, multiclass classification and multilabeling.

*Relevance of GSC for ML and kernel methods.* We would like to stress the significance of our result for part of ML
and related fields. Indeed, while GSC is a subset of convex functions, it still covers many widely used approaches in
machine learning, model estimation, statistics and finance, which are still framed in terms of large scale generalized
linear models, like logistic and soft-max regression. More specifically, kernel regression and classification methods
have the peculiar property to lead to large scale convex problem even if they are learning non-linear functions. In
the community of kernel methods it is still an open problem how to find efficient methods to solve such big convex
problems, without depending on the condition number, and our result is can constitute a key step in this direction.

*Second order skepticism.* We also understand the skepticism towards second order methods. Note however that (1) they
become crucial to solve severely ill-conditioned problems, where the condition number is $\gg n$ i.e. $10^{10}$, since first
order methods can't avoid the dependence on the condition number. (2) the fact that second order methods have been
labeled as inefficient in the '90 exactly because proper approximation techniques *and* globalization schemes where
missing. In the past few years many approximation schemes have been developed as discussed in the introduction using
novel results from randomized linear algebra. Nowadays as discussed in Section 2, the cost per iteration (epoch) of
one approximated second order method is essentially the same as gradient descent (or stochastic gradient descent).
Crucially the globalization scheme provided in this paper guarantees that the number or required iterations does not
depend polynomially on the condition number (as in gradient descent or the stochastic methods where it is linear, or a
square root with acceleration), but only logarithmically. So we need way less iterations than a first order method.

In particular we will make the introduction more clear putting elements of the discussion above; we will add a paragraph
after the definition of GSC function to clarify their role; we will summarize the computational costs of the various
algorithms of Section 2 via an additional table and moved some of the crucial aspects on how we improved over the
state of the art from Section 2 to the contribution paragraph in the spirit of the discussion above.

*Experiments.* The proposed paper is clearly of theoretical flavour. We provide a new algorithm, and crucial part of the
contribution is the development of the mathematical theory in terms of tech-
niques, theorems and proofs to guarantee the correct behaviour of the algorithm.
Additionally to show the practical validity of the approach, we performed a large
scale experiment in the field of kernel methods (see above). In particular, in the
experiments we compared our approach with KSVRG that is nowadays one of
the fastest accelerated first order stochastic methods and is widely used to train
kernel methods. We will add this plot showing the classification error for the
experiment, which shows interestingly that it is still hard for first order methods,
to achieve good performance in classification, for ill conditioned problems.

**Reviewer 2.** We thank Reviewer 2 the very thorough reading of the paper and for pointing out many typos which we
will correct. We agree that FALKON is a very related algorithm; Indeed, the aim of the paper is to extend FALKON
to more general losses. In particular the proposed approach recovers it in the case of squared loss. We are currently
working on a minimal usable implementation for python and multicore-GPU, we will make the code open source and
available on github this fall. Finally we will correct the format of the equation and the typos as suggested.

**Reviewer 3.** We thank Reviewer 3 for the very thorough understanding of the paper. We will add a note about
the difference between GSC [6] and self concordant functions [24] after the definition of GSC functions. The
difference with Newton sketch and our paper is that Newton sketch's globally convergent scheme applies to self-
concordant functions while ours applies to GSC functions. Moreover as suggested, in Section 1.1 we will emphasize
the advantages/disadvantages of our method. Indeed, its aim is to extend the performance of quadratic solvers to GSC
functions, up to a log factor (only quadratic solvers are "condition number independent").

[Meta-Review · NeurIPS 2019]

The paper studies large-scale convex optimization algorithms based on the Newton method applied to regularized generalized self-concordant losses, in particular in ill-conditioned settings, providing new optimal generalization bounds and proofs of convergence. The reviewers found the contributions of high quality and were satisfied with the clarifications provided by the author response.